



# Retrieving 3D distributions of atmospheric particles using Atmospheric Tomography with 3D Radiative Transfer – Part 1: Model description and Jacobian calculation.

Jesse Loveridge[1], Aviad Levis[2], Larry Di Girolamo[1], Vadim Holodovsky[3], Linda Forster[4], Anthony B. Davis[4], Yoav Y. Schechner[3]

[1]Department of Atmospheric Sciences, University of Illinois, Urbana, 61801, USA
[2]Computer and Mathematical Sciences Department, California Institute of Technology, Pasadena, 91125, USA
[3]Viterbi Faculty of Electrical and Computer Engineering, Technion – Israel Institute of Technology, Haifa 3200003, Israel
[4]Jet Propulsion Laboratory, California Institute of Technology, Pasadena, 91109, USA

*Correspondence to*: Jesse Loveridge (jesserl2@illinois.edu)

**Abstract.** Our global understanding of clouds and aerosols relies on the remote sensing of their optical, microphysical, and macrophysical properties using, in part, scattered solar radiation. These retrievals assume clouds and aerosols form plane-parallel, homogeneous layers and utilize 1D radiative transfer (RT) models, limiting the detail that can be retrieved about the 3D variability of cloud and aerosol fields and inducing biases in the retrieved properties for highly heterogeneous structures such as cumulus clouds and smoke plumes. To overcome these limitations, we introduce and validate an algorithm for retrieving the 3D optical or microphysical properties of atmospheric particles using multi-angle, multi-pixel radiances and a 3D RT model. The retrieval software, which we have made publicly available, is called Atmospheric Tomography with 3D Radiative Transfer (AT3D). It uses an iterative, local optimization technique to solve a generalized least-squares problem and thereby find a best-fitting atmospheric state. The iterative retrieval uses a fast, approximate Jacobian calculation, which we have extended from Levis et al. (2020) to accommodate open as well as periodic horizontal boundary conditions (BC) and an improved treatment of non-black surfaces.

We validated the accuracy of the approximate Jacobian calculation for derivatives with respect to both the 3D volume extinction coefficient and the parameters controlling the open horizontal boundary conditions across media with a range of optical depths and single scattering properties and find that it is highly accurate for a majority of cloud and aerosol fields over oceanic surfaces. Relative root-mean-square errors in the approximate Jacobian for 3D volume extinction coefficient in media with cloud-like single scattering properties increase from 2% to 12% as the Maximum Optical Depths (MOD) of the medium increases from 0.2 to 100.0 over surfaces with Lambertian albedos < 0.2. Over surfaces with albedos of 0.7, these errors increase to 20%. Errors in the approximate Jacobian for the optimization of open horizontal boundary conditions exceed 50% unless the plane-parallel media providing the boundary conditions are very optically thin (~0.1).

We use the theory of linear inverse RT to provide insight into the physical processes that control the cloud tomography problem and identify its limitations, supported by numerical experiments. We show that the Jacobian matrix becomes increasing ill-posed as the optical size of the medium increases and the forward scattering peak of the phase function decreases. This suggests



that tomographic retrievals of clouds will become increasingly difficult as clouds becoming optically thicker. Retrievals of asymptotically thick clouds will likely require other sources of information to be successful.

In Part 2 of this study, we examine how the accuracy of the retrieved 3D volume extinction coefficient varies as the optical size of the target medium increases using synthetic data. We do this to explore how the increasing error in the approximate Jacobian and increasingly ill-posed nature of the inversion in the optically thick limit affects the retrieval. We develop a method to improve retrieval accuracy in the optically thick limit. We also assess the accuracy of retrieved optical depths and surface irradiances and compare them to retrievals using 1D radiative transfer.

## 1 Introduction

Cloud and aerosol properties retrieved from the inversion of remote sensing measurements (Stephens and Kummerow, 2007; Dubovik et al., 2011) are a critical source of information for understanding and testing the closure of the Earth's radiation budget (Raschke et al., 2005; McFarlane et al., 2016; Zhou et al., 2016), validating dynamical atmospheric models of varying complexity (Hack et al., 2006; Endo et al., 2015; Bodas-Salcedo et al., 2016) and developing parameterizations in large-scale

models (Hill et al., 2012; Xie and Zhang, 2015). Cloud radiative feedbacks and Aerosol-Cloud Interactions (ACI) (Bellouin et al., 2020), particularly in cumuliform clouds (Sherwood et al., 2014; Vial et al., 2018), are key sources of uncertainty in projections of future climate (Sherwood et al., 2020), and the forecasting of weather (Van Weverberg et al., 2018) and solar energy (Jimenez et al., 2016).

As dynamical modelling of the atmosphere and climate becomes more and more complex, there is a greater demand for high quality observations to constrain the uncertain processes within the models and inform model development (Morrison et al., 2020). New observational techniques are required that can provide robust statistics of small-scale, spatially resolved, cloud and aerosol microphysical parameters, so that their controlling processes can be constrained in both high- and low-resolution modelling. We describe a novel remote sensing retrieval technique with the potential to meet these needs by providing 3D

instantaneous snapshots of volumetric properties of the atmosphere at the resolution of the sensors.

Scattered solar radiation is one of the best candidates for providing high-resolution constraints on aerosol and cloud microphysics due to its well documented sensitivity to the properties of particles in those size ranges (Dubovik et al., 2002; King and Vaughan, 2012; Dzambo et al., 2021; Ewald et al., 2021) and our ability to design narrowband sensors that can cost

efficiently reach high spatial resolution (10s of meters) from space. Typical cloud and aerosol remote sensing retrieval algorithms do not utilize realistic 3D radiative transfer (RT) models to interpret measured scattered solar radiation, e.g., (Dubovik et al., 2011; Grosvenor et al., 2018). Instead, they make use of two key simplifying assumptions when interpreting radiance measurements. Firstly, they assume that the media (e.g., clouds) form horizontally homogeneous, plane-parallel layers within the field of view of each radiance measurement. Secondly, they assume that there is no radiative interaction between



the regions within the field-of-view of each radiance measurements, in what is known as the Independent Pixel Approximation (IPA).

These assumptions dramatically reduce the computational complexity of the retrieval process but they also compromise retrieval accuracy (Marshak et al., 2006; Zhang et al., 2012; Kato and Marshak, 2009) leading to errors in, for example, cloud

optical depth that can have domain biases of ~35% in cumuliform clouds (Seethala, 2012), and large inter-instrument inconsistencies in retrievals (Di Girolamo et al., 2010; Lebsock and Su, 2014; Ahn et al., 2018; Fu et al., 2019; Painemal et al., 2021). These errors arise because clouds and aerosols can have strong horizontal gradients in their physical and optical properties (Marshak et al., 1997; Gerber et al., 2001; Zhao and Di Girolamo, 2007; Kahn et al., 2007) which breaks the assumption of the IPA and necessitates the use of 3D RT to accurately model the transport of radiation (Davies, 1978; Cahalan

et al., 2005).

These errors alias into our climate records and also affect retrievals made using a combination of passive and active sensors (Saito et al., 2019). The retrieval assumptions also limit the amount of detail about the cloud and aerosol fields that can be retrieved using passive sensing as, once the IPA is imposed, there is no way to identify the vertical geometric variability of

cloud and aerosol microphysics without active instruments and, in the case of cloud radar, strong assumptions about the shape of the particle size distribution. An algorithmic advance in operational retrievals is required to better extract the information about 3D variability of cloud and aerosol microphysics contained within high-resolution solar radiance measurements to provide the novel observations required for advancing cloud and aerosol science.

Many algorithms have been demonstrated that utilize 3D RT to improve remote sensing retrievals of cloud properties, but have been limited to using radiometric information from a single mono-angle imager at a time (Marshak et al., 1998a; Marchand and Ackerman, 2004; Zinner et al., 2006; Cornet and Davies, 2008) or a single zenith radiance measurement (Fielding et al., 2014). This restriction limits the amount of information obtainable from the radiance field so that only column-integrated or horizontally averaged properties of the atmosphere can be inferred rather than fully three-dimensional variability. As a result,

several of these methods have relied on external sources of information such as scanning radar or in-situ data (Marchand and Ackerman, 2004; Fielding et al., 2014), or strong assumptions that are not generally applicable (Marshak et al., 1998a; Zinner et al., 2006; Cornet and Davies, 2008). It is clear that a new source of information is required to relax the strong assumptions required by these retrieval algorithms to enable the retrieval of 3D spatial structure.

Multi-angle imagery is a promising source of information to constrain the 3D structure of the atmosphere. Inverse problems where multi-angle boundary measurements are used to infer internal structure are commonly known as tomography. In atmospheric science, tomographic methods have been applied to retrieve cloud properties using non-scattering, multi-angle microwave emission (Huang et al., 2008, 2010), water vapor using microwave attenuation (Jiang et al., 2022), and aerosol



using scattering measurements (Garay et al., 2016; Zawada et al., 2017, 2018). Multi-angle imaging has been utilized to
systematically retrieve geometric properties of clouds (Muller et al., 2002, 2007) and aerosols (Kahn et al., 2007) using
stereoscopic methods.

In the field of medical imaging, multiple detectors are routinely utilized to retrieve spatially varying optical properties of the
human body from multiply scattered near-infrared radiation in a process known as Diffuse Optical Tomography (Arridge and
Schotland, 2009; Bal, 2009). Taking this as inspiration, a similar tomographic approach has been proposed and formalized for
the retrieval of spatially varying atmospheric constituents from multi-angle imagery of multiply scattered solar radiation in the
atmospheric context (Martin et al., 2014). These methods must make use of 3D RT models as 1D RT models are unable to
reproduce the angular variations of the observed radiation field (Di Girolamo et al., 2010). Tomographic methods have the
potential to provide remote sensing retrievals of volumetric cloud and aerosol properties such as the 3D distribution of volume
extinction coefficient, possibly even microphysical quantities.

Tomography problems are commonly solved using iterative, physics-based optimization procedures similar to state of the art
methods in aerosol remote sensing (Xu et al., 2019; Gao et al., 2021). They can also be solved using statistical methods (Zhang
and Zhang, 2019; Ronen et al., 2022) or heuristic methods, which have been explored recently in the atmospheric science
context (Alexandrov et al., 2021). Tomographic methods in Diffuse Optical Tomography typically make use of
computationally efficient forward-adjoint methods to linearize a 3D RT model and calculate cost function gradients (Arridge
and Schotland, 2009). Similar methods have been employed in plane-parallel retrievals of aerosol properties (Hasekamp and
Landgraf, 2005). So far, tomographic retrievals utilizing forward-adjoint methods have only been demonstrated in atmospheric
sciences in 2D (Martin and Hasekamp, 2018).


Similar tools have been developed elsewhere. The Monte Carlo 3D RT equation (RTE) solver McArtim (Deutschmann et al.,
2011) is specialized for radiance derivative calculations using forward-adjoint methods, but uses a backward Monte Carlo
technique, similar to other work (Loeub et al., 2020), with importance sampling, which will not scale well to the multi-angle
imagery required for tomography. Other Monte Carlo techniques for derivative calculations typically use forward methods
with path-recycling (Langmore et al., 2013; Yao et al., 2018; Czerninski and Schechner, 2021), however most implementations
of such models lack the variance reduction methods required for efficient modelling of radiances with sharply peaked phase
functions (Buras and Mayer, 2011; Wang et al., 2017) and have not been benchmarked on atmospheric problems.

Of the available 3D RTE solvers that are benchmarked on atmospheric scattering problems (Cahalan et al., 2005), the
deterministic (a.k.a explicit) Spherical Harmonics Discrete Ordinates Method (SHDOM) (Evans, 1998) is the most
computationally efficient for tomography. This is due to the need to simulate many radiometric quantities. SHDOM is almost
two orders of magnitude more computationally efficient than Monte Carlo on CPU for multi-angle imagery (Pincus and Evans,



2009). Monte Carlo solvers specialized for 3D atmospheric scattering problems have been slow to adopt GPU-based computation, which is anticipated to give a reduction in wall-time of between one and two orders of magnitude (Efremenko et

al., 2014; Ramon et al., 2019; Wang et al., 2021; Lee et al., 2022) and thereby making Monte Carlo competitive against SHDOM in the future.

As of writing, there is no publicly available adjoint to a deterministic (a.k.a. explicit) 3D RTE solver appropriate to the atmospheric context like SHDOM. A forward-adjoint linearization of the SHDOM method has been developed (Doicu and

Efremenko, 2019) and an SHDOM solver has been extended so that general adjoints appropriate for tomography can be computed (Doicu et al., 2022b). Unfortunately, these developments are not publicly available, which makes us unable to build upon these advances. Fortunately, a computationally efficient approximation to the adjoint of SHDOM has been developed and used to demonstrate the success of fully 3D retrievals of the volume extinction coefficient of clouds using multi-angle, mono-spectral imagery in a first for atmospheric remote sensing (Levis et al., 2015). This method of approximate linearization

has been extended to utilize multi-spectral (Levis et al., 2017) and polarized (Levis et al., 2020) observations. Interestingly, the forward-adjoint method of cloud tomography using SHDOM suffered from slow convergence and the authors only found success in their synthetic tomographic retrievals when utilizing the approximate linearization of Levis et al. (2020) (Doicu et al., 2022a) in combination with their adjoint method (Doicu et al., 2022b).

The method of Levis et al. (2020) is the most mature and successful remote sensing retrieval using solar radiances and 3D RT available in the atmospheric sciences. The method is still restricted in that its implementation is limited to isolated 3D domains and Lambertian surfaces, and the approximate linearization is a poor approximation for non-black surfaces. Despite these limitations, the method's maturity makes it the ideal starting point for developing retrievals of 3D volumetric microphysical parameters at the scale of LES resolution using real measurements. The future space-borne CloudCT mission (Schilling et al.,

2019) will provide the required simultaneous multi-angle imagery for tomographic retrievals. Existing airborne instruments such as AirMSPI (Diner et al., 2013) and AirHARP (McBride et al., 2020) and the space-borne MISR and MAIA instruments also have potential for tomographic retrievals, though they must additionally deal with the effects of cloud evolution (Ronen et al., 2021) as they do not acquire their observations simultaneously. The availability of these measurements makes the continued development of tomographic algorithms especially timely. Retrievals of this sort have the potential to provide the

robust statistics of small-scale cloud and aerosol properties required for constraining cloud processes (Morrison et al., 2020), especially when extended to include information from other instruments such as cloud radar.

In this two-part series of papers, we present and validate an extension to the retrieval framework of Levis et al. (2020), which we have implemented and made publicly available in the software package Atmospheric Tomography with 3D Radiative

Transfer (AT3D) (Loveridge et al., 2022). This paper, which is Part 1, is devoted to the description of the retrieval methodology



and the underlying theory of the retrieval, along with supporting numerical evidence. Part 2 of this study is devoted to tomographic retrievals on synthetic data to validate the method.

In Section 2, we describe the retrieval software AT3D which is quite general in that it is designed for retrieving the 3D microphysical properties of external mixtures of atmospheric particles using multi-angle, multi-pixel and possibly multi-spectral, polarized radiances. In Section 3, we describe extensions to the method of Levis et al. (2020) to include an improved treatment of non-black surfaces and the retrieval of a plane-parallel medium in which the 3D domain is embedded, thereby improving the realism of the method. The appendices document the discrete implementation of the algorithm and model verification.


Despite the successful demonstrations of the tomographic retrieval (Levis et al., 2015, 2017; Martin and Hasekamp, 2018; Levis et al., 2020; Doicu et al., 2022a, b), it is still unclear how the effectiveness of tomographic techniques will vary with scattering regime. Previous studies have shown that success is not uniform, with poorer performance in optically thick clouds (Levis et al., 2015). It is not clear whether this is a result of a limitation in the approximate linearization method or a physical

limitation. In Section 4, we present the theory of linear inverse transport problems and use it to explain the limitations of tomography in general, to provide insight into the physical processes that control the cloud tomography problem. This theory is drawn from both the wider literature on inverse problems (Bal and Jollivet, 2008; Bal, 2009; Chen et al., 2018; Zhao and Zhong, 2019) as well as relevant literature in the atmospheric sciences that have studied the loss of information about spatial detail of cloud properties in multi-pixel radiances due to multiple scattering (Marshak et al., 1995, 1998b; Davis et al., 1997;

Forster et al., 2020).

In Section 5, we perform a detailed quantitative validation of the approximate linearization to SHDOM that is utilized in AT3D, following Levis et al. (2020). Despite the success of the method in several test cases in Levis et al. (2015, 2017, 2020), no validation of the approximation itself has yet been performed, which has made it as yet unclear how the method will

generalize to the wider variety of scattering regimes present in the cloudy atmosphere. We present an alternative derivation of the approximation that places it in the context of the forward-adjoint formalism developed by Martin et al. (2014). We can then explain the success of the approximate method using the theory presented in Section 4. In Section 6, we briefly quantitatively contrast stratiform and cumulus cloud geometries in terms of the well-posedness of the tomography problem from a linear perspective using the approximate Jacobian. We summarize our results in Section 7. In Part 2 of this study we

explore how these issues affect the fully nonlinear retrieval problem and demonstrate the effectiveness of the method described here.



## 2 Atmospheric Tomography with 3D Radiative Transfer (AT3D)

Atmospheric Tomography with 3D Radiative Transfer (AT3D) is a software package designed to perform tomographic retrievals of atmospheric properties. It poses the inverse problem as a nonlinear, generalized least-squares problem that is
solved using iterative local optimization techniques. The solution procedure is physics-based and uses the 3D RT model SHDOM (Evans 1998) as its forward model to connect retrieved quantities to measured radiance. SHDOM is an explicit solver of the polarized 3D RTE that is well established in atmospheric science. During the Intercomparison of 3D Radiative Transfer Codes (I3RC, Cahalan et al. 2005), SHDOM was well within the consensus results. This is also true of intercomparisons of polarized RT (Emde et al., 2015, 2018).


In brief, SHDOM solves the integral form of the monochromatic vector RTE on a Cartesian grid using a fixed-point iteration scheme for collimated solar or thermal emission sources of radiation. A spherical harmonic expansion representation of the radiation field is used for computing the source function of the RTE while a discrete ordinate representation is used for the streaming of radiation. At each iteration the source function is transformed to discrete ordinates, new radiances are computed
at each grid point using a short characteristic scheme, and a new spherical harmonic representation of the source function is computed. An adaptive spatial grid is employed so that grid cells with a variation of the source function larger than a threshold are split in half, generating new grid points. The number of spherical harmonics kept at each grid point is also adaptively truncated.

AT3D builds off of the software implemented by Levis et al. (2020), which itself builds upon the work of Evans (1998) on the publicly-available Fortran implementation of SHDOM. AT3Dis also a python wrapper for the SHDOM RTE solver developed using the f2py tool (Peterson, 2009), this enables easy interfacing with external optimization libraries from SciPy (Virtanen et al., 2020). Use of Python also enables interactivity, even in High Performance Computing (HPC) environments, which accelerates data exploration and code prototyping. The key features of the SHDOM software are preserved in AT3D with the
only notable exception being that AT3D does not yet implement the MPI-based parallelization of SHDOM so it is not yet able to efficiently utilize HPC resources to solve large-scale forward or inverse problems.

AT3D's strength is as a provider of a physics-based, and therefore flexible, method for solving the inverse problem of atmospheric tomography. It is therefore perfectly suited for performing sensitivity tests to changes in the measuring
instrument's configuration (e.g., number of view angles and sensor resolution). AT3D supports the retrieval of multiple external mixtures of 3D distributions of scattering particles with solar and thermal sources using arbitrary combinations of possibly polarized, multi-wavelength, monochromatic radiances. Each unknown can be retrieved on a 3D grid or on a user-specified simplified spatial basis (e.g., column averages). AT3D does not yet support non-simultaneous measurements and corresponding retrieval of a time-varying cloud (Ronen et al., 2021). Flexibility with the configuration of any retrieval problem


is supported using object-oriented and functional programming in the python wrapper. Currently, AT3D includes just the Rayleigh and Mie scattering particle models for homogeneous spheres distributed with SHDOM (https://nit.coloradolinux.com/shdom.html) (Evans 1998). The parameterization of the size distribution and accompanying selection of unknowns (e.g. droplet number concentration or liquid water content) for retrievals is flexible.

We note that the software is far from a black-box tool and operates more as a library of high-level objects and functions that can be combined in short python scripts according to user specifications. This level of flexibility is good for a research tool that is under active development, though it can lead to a steeper learning curve than a well-defined executable with a fixed set of options common in other RT software packages. The code is well documented and several tutorials are included with the code to mitigate this. Users or potential developers are welcome to contact the authors to discuss their potential use case.


Just like the SHDOM software, AT3D is not a complete RT package, and does not include detailed spectroscopic or particle scattering data that can be found elsewhere (Emde et al., 2016; Gordon et al., 2022; Saito et al., 2021). Interfacing with these packages is relatively simple as data is represented in AT3D using the xarray package (Hoyer and Hamman, 2017), which supports a variety of file formats such as netCDF.


AT3D supports all surface Bidirectional Reflectance Distribution Functions (BRDF) available in SHDOM but does not yet include linearization with respect to the parameters describing surface BRDFs. AT3D supports inversion of optical or microphysical properties with thermal or combined sources, a helpful extension from Levis et al. (2020) for the far SWIR and IR, but also does not yet include linearization with respect to atmospheric temperature. These aspects are under development.

The radiance calculations in AT3D have been generalized from SHDOM to support more realistic sensor geometries and sensor spatial response functions. However, as noted above, observables are currently monochromatic and some small extension to the software would be required to accommodate observables requiring multiple monochromatic RTE solutions during inversion. The representation of phase functions within the SHDOM solver has been modified so that the SHDOM model is differentiable (Appendix A). The package also includes several useful tools including a stochastic generator for

making synthetic clouds (Section 5.1), a space-carving algorithm (Lee et al. 2018) for performing volume cloud masking for retrieval initialization (Section 5.3) and several basic regularization schemes.

While many practical extensions have been made to the retrieval software since Levis et al. (2020), including a comprehensive verification (Appendices B & D), which is critical to establish the veracity of the scientific results (Kanewala and Bieman,

2014). The novel elements of the retrieval software are the extensions of the linearization to non-homogeneous surfaces and open BCs, which are documented in the following section.



# 3 Retrieval Methodology

We now present an overview of the iterative retrieval process. There are many mathematical terms introduced in this section. A glossary is provided in the Appendix for reference and a flowchart of the retrieval process is shown in Fig. 1. The solution of the tomography problem involves the selection of a state vector ($\boldsymbol{a}$) that parameterizes a discrete representation of the atmospheric optical or physical properties and best fits the available measurements ($\boldsymbol{y}$) and any prior knowledge of the unknown state. The total size of the state vector ($\boldsymbol{a}$) depends on the domain size and discretization scheme, but even for the smallest problems with a 3D gridded representation it will typically range upwards of 10,000. We note that the state vector does not necessarily need to consist of physical variables and may instead consist of, for example, linear combinations of physical variables. This is a generalization that is preferable to ensure the associated optimization problem is well-scaled (Nocedal and Wright, 2006).

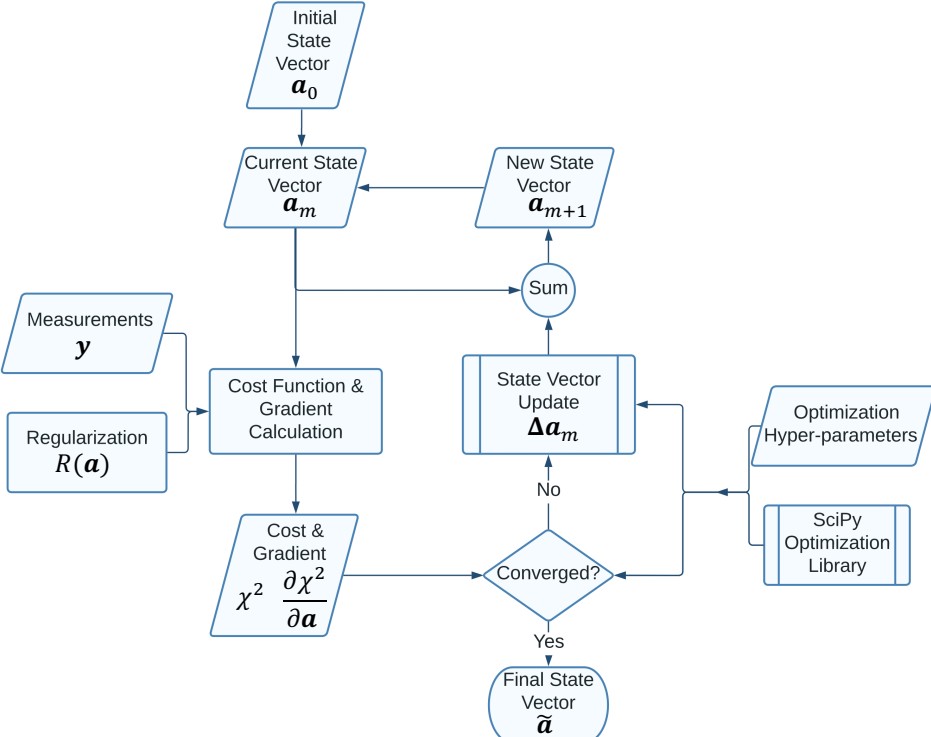

**Figure 1: A flowchart depicting the overall iterative retrieval methodology of AT3D.**

The measurement vector ($\boldsymbol{y}$) contains multi-angle, multi-pixel radiances, which may also be multi-spectral and polarized. The number of observations will also typically range above 10,000 and exceed the dimension of the state vector to avoid ill-posedness. The forward model $\boldsymbol{F}(\boldsymbol{a})$ provides the mapping from the state vector to the measurement space by producing synthetic measurements equivalent to $\boldsymbol{y}$ based on the unknown state vector ($\boldsymbol{a}$) and any fixed ancillary data. The forward model consists of multiple components: the mapping from the state vector ($\boldsymbol{a}$) to the 3D optical properties at all required wavelengths,



the solution of the required 3D RT problems and, finally, the sampling of the radiance field at the required positions, angles,
wavelengths and polarization states to produce synthetic measurements. We describe each of these components of the forward
model in the following subsection.

We select a best-fitting state by minimizing a scalar cost function. The scalar cost function $\chi^2$ is chosen to penalize misfit from
the measurements in a generalized, least-squares sense:

$$\chi^2 = \big(\boldsymbol{y} - \boldsymbol{F}(\boldsymbol{a})\big)^T \boldsymbol{S}_\epsilon^{-1}\big(\boldsymbol{y} - \boldsymbol{F}(\boldsymbol{a})\big) + R(\boldsymbol{a}), \tag{1}$$

where the error covariance matrix of the residual between the measurements ($\boldsymbol{y}$) and the forward model $\boldsymbol{F}(\boldsymbol{a})$ is denoted by
$\boldsymbol{S}_\epsilon$, and accounts for both measurement uncertainty and forward model uncertainty. AT3D currently supports a block diagonal
$\boldsymbol{S}_\epsilon$, where error-correlations are allowed between different Stokes components measured at each pixel, which supports
inclusion of certain types of forward modelling error and instrumental noise (Harten et al., 2018). It does not yet support
systematic error-correlations between pixels due to, for example, uncertainties in flat-fielding, camera-to-camera
intercalibration, band-to-band calibration error or absolute calibration error. $R(\boldsymbol{a})$ is a differentiable regularization term that
reflects prior knowledge about the structure of the unknown state. Note that there is no requirement here that the regularization
term takes the form of an a priori distribution. As such, the formulation in AT3D is more general than the optimal estimation
frameworks utilized widely in atmospheric remote sensing (Sourdeval et al., 2013; Wang et al., 2016).

The solution to the inverse problem is given by the minimizer of the cost function subject to box constraints described by
vectors of lower bounds ($\boldsymbol{l}$) and upper bounds ($\boldsymbol{u}$) on each element of the state vector. Formally,

$$\tilde{\boldsymbol{a}} = \underset{\boldsymbol{a}}{\arg\min}\,\chi^2, \quad \text{s.t.} \quad \boldsymbol{l} \le \boldsymbol{a} \le \boldsymbol{u}. \tag{2}$$

For example, these bounds can be used to ensure positivity of liquid water content or volume extinction coefficient. This
optimization problem can be solved efficiently using a local minimization technique such as the Limited memory-Broyden-
Fletcher-Goldfarb-Shannon method for Bounded minimization (L-BFGS-B) (Byrd et al., 1995) when the cost function is
differentiable. The local minimization proceeds through the selection of an initial guess $\boldsymbol{a_0}$, and iteratively updating the state
through repeated evaluation of the cost function and its gradient. At the $m^{\text{th}}$ iteration we then have

$$\boldsymbol{a}_{m+1} = \boldsymbol{a}_m + \Delta\boldsymbol{a}_m. \tag{3}$$

The retrieval method requires the selection of an initial guess, which can have a significant influence on the optimization. In
Part 2 of this study, we describe some of the ways that initial guesses can be generated in AT3D.

The L-BFGS-B method is a quasi-Newton method which selects the update to the state vector ($\Delta\boldsymbol{a}_m$) by first selecting a search
direction through the minimization of an approximate, local, quadratic model of the cost function. The quadratic model uses





an approximation to the Hessian of the cost function which is formed by analysing how the cost function gradient changes over the most recent $M$ iterations. $M$ is a hyperparameter of the optimization. In essence, the approximation to the Hessian of the cost function is formed from finite-differencing of successive gradient vectors. As such, it only reflects curvature information along the directions that the optimization trajectory is currently exploring. After the selection of a search direction, an inexact line search obeying the Wolfe-Armijo conditions, which ensure stability and convergence, is then used to select the

final update $\Delta a_m$ along the search direction. The implementation of L-BFGS-B from the SciPy library is used (Virtanen et al., 2020). The L-BFGS-B algorithm has a better convergence rate than simple gradient descent but still only requires the evaluation of the cost function and its gradient. The computational cost of the update is modest once the gradient vector is computed. The storage requirement of L-BFGS-B is also modest, being limited to storing the state updates and gradient changes over the past $M$ iterations, so it scales linearly with the size of the state-vector. This makes the method appropriate for

the large-scale optimization problem of cloud tomography.

The expression for the gradient of the data fit term required by the L-BFGS-B algorithm is

$$\frac{\partial \chi^2}{\partial a} = 2\big(y - F(a)\big)^T S_\epsilon^{-1} K, \tag{4}$$

where $K$ is the Jacobian matrix containing the partial derivatives of the $i^{th}$ output of the forward model with respect to the $j^{th}$

component of the state vector:

$$K_{ij} = \frac{\partial F_i(a)}{\partial a_j}. \tag{5}$$

It is in our interest to understand the information content of our measurements and the factors that control the convergence rate of the retrieval so that these can be maximized. The linear information content in the measurements in the vicinity of a cost function minimum can be determined using the Fisher information matrix, which in the case of Gaussian errors takes the

simple form $K^T S_\epsilon^{-1} K$. The singular-value spectrum of this matrix describes the magnitude of retrieval uncertainties across different directions in state space and this distribution is largely controlled by the singular-value spectrum of $K$. The largest physical values of the singular value spectrum of $K$ are physically-bounded (Chen et al., 2018) and linearizations of RT tends to have rapidly (i.e., exponentially) decaying singular values (Culver et al., 2001). As such, we summarize the singular value spectrum using just the condition number of the Jacobian matrix $\kappa(K)$, which is the ratio of its largest ($s_{largest}$) and smallest

($s_{smallest}$) singular values:

$$\kappa(K) = \frac{s_{largest}}{s_{smallest}}. \tag{6}$$

The L-BFGS-B method can suffer in systems where the condition number becomes large, where poor accuracy and slow convergence can occur (Zhu et al., 1997) even for quadratic cost functions. A slow convergence rate is particularly troubling as it will determine the computational feasibility of performing cloud tomography. In Section 4, we relate the structure of the



Jacobian matrix, specifically the condition number, to the properties of the medium through the principles of RT theory and support these principles with numerical simulations. We then examine the ability of the approximate Jacobian to accurately represent the true Jacobian matrix and to correctly capture the information content of the measurements about the state vector. Before presenting these results, we must first introduce the formulation of the forward model and the exact calculation of the Jacobian matrix.

## 3.1 Forward Model Description


We now describe the formulation of the forward model and calculation of its Jacobian matrix. The theory for such calculations has already been presented for general 3D problems (Martin et al., 2014), and also in the specific context of the SHDOM model with periodic BCs (Doicu and Efremenko, 2019). In the work of Levis et al. (2015, 2017, 2020), open BCs were used without any incoming radiance at the domain edges (i.e., vacuum BCs). Neither of these configurations fully represents the

realistic case of retrieving a heterogeneous 3D domain embedded within a horizontally infinite medium such as might be performed when we retrieve a field of cumulus clouds embedded in a cloud-free atmosphere. In this section, we describe a forward model for this scenario and its linearization, focusing on the specific context of the SHDOM solver. We describe the linearization of the forward model with respect to the parameters that control both the 3D domain and the embedding horizontally infinite medium, as implemented in AT3D.


Note that the principles of the forward model itself remain unchanged from the implementation of open BCs in the original SHDOM code. In SHDOM, when open horizontal BCs are selected, incoming radiances may be prescribed at the horizontal boundaries of the primary domain of 3D RT through the solution of auxiliary RT problems that describe the radiance field in the embedding medium. These auxiliary RT problems are 2D and 1D, describing the plane-parallel embedding medium. The

coupling of the BCs between the RT problems, and the fact that a sampled radiance measurement contains contributions from both the primary 3D RTE solution and also the auxiliary RTE problems, introduces some complexity in the mathematical formulation of this model and its linearization. Our description below builds upon the formalism presented by Martin et al. (2014) and simply applies it to describe existing behaviour of the SHDOM model. This detailed description is necessary so that we can differentiate between the exact calculation of the Jacobian matrix and the approximations used in AT3D in Sections

3.2 and 3.3. Section 3.1.1 presents the definitions and geometry of the model. Section 3.1.2 describes the RT solution procedure. Section 3.1.3 describes the radiance calculation.

### 3.1.1 Problem Setting

We begin by first defining the spatial domain of interest on which the monochromatic vector 3D RTE will be solved. The SHDOM model adopts a Cartesian geometry and the physical domain $D \subset \mathbb{R}^3$ is the horizontally infinite slab of thickness $L_z$

where $z \in [0, L_z]$. This domain is broken up into nine cuboids, in each of which a different RT problem is solved. A top-down



view of the arrangement of the domains is shown in Fig. 2. The primary physical domain of 3D RT is the cuboid $D_1$ described by the position vector $\boldsymbol{r}$ with smooth boundary $\partial D_1$:

$$D_1 = \left\{ \boldsymbol{r} \in \mathbb{R}^3 : 0 < x < L_x, 0 < y < L_y, 0 < z < L_z \right\}. \tag{7}$$

Around this primary domain are the auxiliary RTE domains $D_2$ through $D_9$ which we may consider to also be cuboids but they have horizontal boundaries at some very large distance from the primary domain. We will denote the position of these horizontal boundaries as being at infinity, for example, $x = -\infty$. Practically speaking, the absolute extent of these auxiliary cuboids only needs to be greater than the position of any considered sensor. The absolute size only comes into play during the calculation of radiances in SHDOM, not the solution of the RTE itself, as will soon become apparent. In our description, we only consider a subset of the auxiliary domains and their coupling to the primary domain as the rest can be treated similarly,

following symmetry. $D_2$ and $D_4$ are examples of corner auxiliary domains, which share 1-dimensional edges with the primary domain ($D_1$), while $D_3$ is an example of a side auxiliary domain, which shares a horizontal plane with the primary domain ($D_1$). Specifically,

$$D_2 = \left\{ \boldsymbol{r} \in \mathbb{R}^3 : -\infty < x < 0, \ y > L_y, \qquad 0 < z < L_z \right\}, \tag{8}$$

$$D_3 = \left\{ \boldsymbol{r} \in \mathbb{R}^3 : -\infty < x < 0, \ 0 < y < L_y, 0 < z < L_z \right\}, \tag{9}$$

$$D_4 = \left\{ \boldsymbol{r} \in \mathbb{R}^3 : -\infty < x < 0, \ y < 0, \qquad 0 < z < L_z \right\}. \tag{10}$$

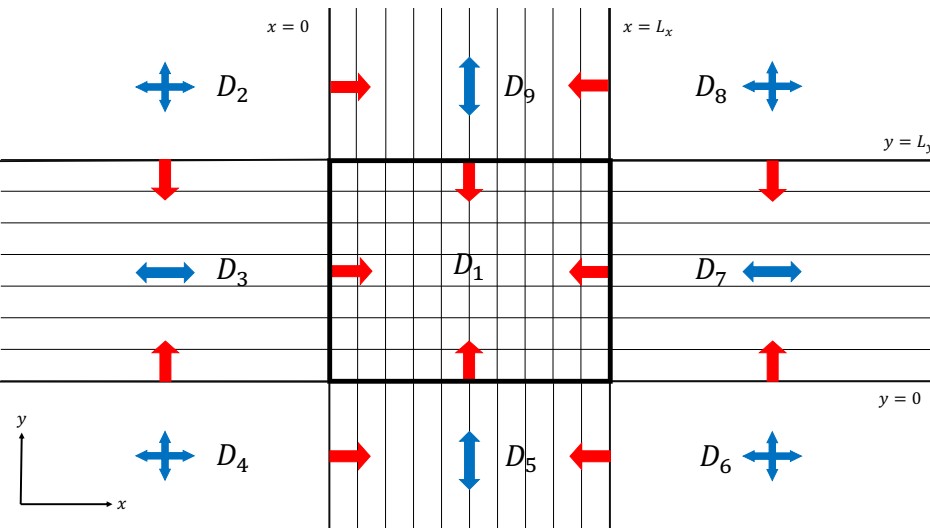

**Figure 2: Top-down view of the geometry of the system of RT problems. $D_1$ denotes the domain of 3D RT. The blue arrows denote the directions in each domain for which there are periodic horizontal BCs. The red arrows denote which domains supply incoming radiances at each boundary. For example, the RTE problem solved on $D_3$ supplies the horizontal radiance BC to the 3D RTE**

**problem solved on $D_1$ at the common plane shared between them.**





Optical properties are defined at all positions in $D$, and are allowed to vary in 3D within $D_1$. Within the auxiliary domains, the optical properties are simplified depending on whether the auxiliary domain shares a corner (e.g., $D_2$ and $D_4$) or a horizontal side (e.g., $D_3$) with the primary domain $D_1$. Optical properties are homogeneous in the $x$ and $y$ in the corner domains and are homogeneous in the direction normal to the boundary of the side domain that is shared with the primary domain. In the case

of $D_3$, this is the $x$ direction.

To describe the RT problems and their coupling we need to distinguish between the domains and their boundaries, for which we must define some relevant sets. These definitions are also illustrated in Fig. 3. The set of directions for the RTE is the unit sphere $\mathbb{S}^2$ scanned by the propagation direction $\boldsymbol{\Omega}$. For any domain $D_n$, if we define the unit outward normal to the boundary

of domain $D_n$ as $\boldsymbol{n}_n(\boldsymbol{r})$, we can define the incoming $\Gamma_n^-$ and outgoing sets $\Gamma_n^+$ on the boundary of the domain as

$$\Gamma_n^\pm = \{(\boldsymbol{r}, \boldsymbol{\Omega}) \in \partial D_n \times \mathbb{S}^2 : \pm \boldsymbol{n}_n(\boldsymbol{r}) \cdot \boldsymbol{\Omega} > 0\}. \tag{11}$$

These two sets allow the separation of radiance at the boundary of the domain into the sets of directions that are entering the domain ($\Omega^-$) and those directions where radiance is leaving the domain ($\Omega^+$).

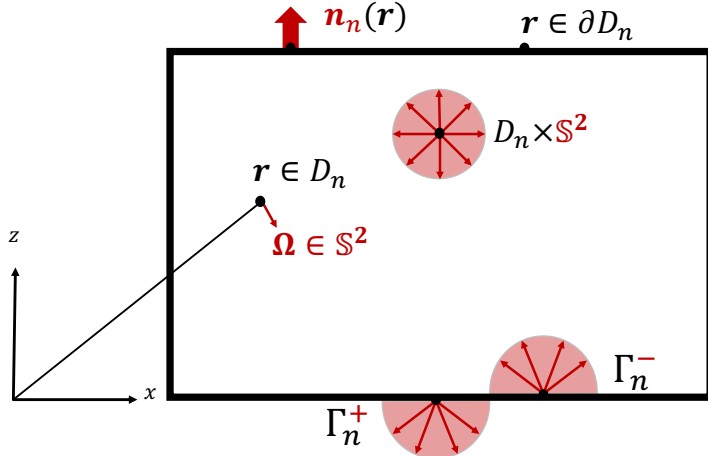

**Figure 3: A side view of the n^th RT domain illustrating several key definitions such as the internal, incoming and outgoing sets and the normal vector. Directional quantities are shown in red, while positional quantities are shown in black. See Eq. 11 and associated discussion in the main text for more details.**

For each domain ($n = 1, \cdots, 9$), we solve the monochromatic vector RTE for the polarized radiance field (a.k.a. Stokes vector) $\boldsymbol{I}_n(\boldsymbol{r}, \boldsymbol{\Omega}) = [I, Q, U, V]_n^{\mathrm{T}}$ on the internal set $D_n \times \mathbb{S}^2$. In the following, we omit the domain subscript unless relevant, i.e., when

considering the coupling between domains. The union of the internal set and boundary is written as $D_n \times \mathbb{S}^2 \oplus \Gamma_n^\pm$. The RTE can be written in terms of the transport operator

$$\mathcal{L}[\boldsymbol{I}(\boldsymbol{r}, \boldsymbol{\Omega})] = \boldsymbol{\Omega} \cdot \nabla \boldsymbol{I}(\boldsymbol{r}, \boldsymbol{\Omega}) + \boldsymbol{I}(\boldsymbol{r}, \boldsymbol{\Omega})\sigma(\boldsymbol{r}) - \omega(\boldsymbol{r})\sigma(\boldsymbol{r}) \int_{\mathbb{S}^2} \boldsymbol{Z}(\boldsymbol{r}, \boldsymbol{\Omega}, \boldsymbol{\Omega}') \boldsymbol{I}(\boldsymbol{r}, \boldsymbol{\Omega}') \mathrm{d}\boldsymbol{\Omega}', \tag{12}$$





where $\sigma(r)$ is the volume extinction coefficient, $\omega(r)$ is the single scattering albedo, and $Z(r, \Omega, \Omega')$ is the phase matrix. The RTE is then simply

$$\mathcal{L}[I(r, \Omega)] = f(r, \Omega) \qquad \text{on } D_n \times \mathbb{S}^2, \tag{13}$$

where $f(r, \Omega)$ is the volume source vector. For example, thermal emission is an isotropic unpolarized volume source that reads as $f(r, \Omega) = (1 - \omega(r))\sigma(r)B(r)$, where $B(r) = [B_\lambda(T), 0, 0, 0]^{\mathrm{T}}$ with $B_\lambda(T)$ being the Planck black-body radiance function. Equation (13) is paired with appropriate BCs which constrain the solution through enforced continuity of the radiance field at the boundary. The BCs of the primary, side and corner domains are different and this introduces significant complexity in the formulation of the forward model. Corner domains (e.g., $D_2$ and $D_4$) have periodic BCs in both $x$ and $y$ while side domains (e.g., $D_3$) have periodic boundaries only in the direction normal to the boundary of the primary domain. For example, in $D_3$, a side domain, we have in the $x$ direction that

$$I_3(x = -\infty, y, z, \Omega)|_{\Gamma_3^+} = I_3(x = 0, y, z, \Omega)|_{\Gamma_3^-}. \tag{14}$$

Directions of periodicity in the RT in each domain are denoted by the blue arrows in Fig. 2. On the horizontal directions that do not have periodicity, we have open BCs with incoming radiance prescribed by the boundary source vector $g(r, \Omega)$ and reflection of outgoing radiance by the reflection operator $\mathcal{R}$:

$$I(r, \Omega)|_{\Gamma^-} - \mathcal{R}[I(r, \Omega)|_{\Gamma^+}] = g(r, \Omega)|_{\Gamma^-} \qquad \text{on } \Gamma^-. \tag{15}$$

The reflection operator $\mathcal{R}$ is defined as the integral over the hemisphere of incoming directions weighted by $\mathbf{R}(r, \Omega, \Omega')$, which is a polarized Bidirectional Reflectance Distribution Function (BRDF),

$$\mathcal{R}[I(r, \Omega)|_{\Gamma^+}](r, \Omega) = \int_{\Omega^+} |n \cdot \Omega'| \, \mathbf{R}(r, \Omega, \Omega') I(r, \Omega') \mathrm{d}\Omega', \qquad \text{on } \Gamma^+. \tag{16}$$

We note that, in the implementation of SHDOM, the BRDF function $\mathbf{R}(r, \Omega, \Omega')$ is non-vanishing only on the lower boundary of all the domains ($z = 0$). Within SHDOM, the boundary source vector $g(r, \Omega)$ is also restricted to being composed of four components. The first three are thermal emission from the surface $g^{\mathrm{BOT}}(r, \Omega)$, horizontally homogeneous, isotropic emission from the domain top $g^{\mathrm{TOP}}(r, \Omega)$, and a unidirectional collimated source due to solar illumination, $g_\odot(r, \Omega)$, with intensity $F_0$ incident on the domain top:

$$g_\odot(r, \Omega) = F_0 \delta(\Omega - \Omega_{\mathrm{sun}}) \delta(z - L_z). \tag{17}$$

The fourth component is the most complex and is only defined on the horizontal sides which we will denote by $g^{\mathrm{SIDE}}(r, \Omega)$. This boundary source vector on the sides is the solution of the neighbouring auxiliary RTE problem, and thus represents the incoming light into each domain due to the embedding medium. Taking the side domain $D_3$ as an example, the incoming boundary source at each of the boundaries shared with corner domains $D_2$ and $D_4$ are simply the outgoing radiance fields in those corner domains:



$$g_3^{\text{SIDE}}\left(x, y = L_y, z, \boldsymbol{\Omega}\right)\Big|_{\Gamma_3^-} = \boldsymbol{I}_2\left(x, y = L_y, z, \boldsymbol{\Omega}\right)\Big|_{\Gamma_2^+}; \tag{18}$$

$$g_3^{\text{SIDE}}\left(x, y = 0, z, \boldsymbol{\Omega}\right)\Big|_{\Gamma_3^-} = \boldsymbol{I}_4\left(x, y = 0, z, \boldsymbol{\Omega}\right)\Big|_{\Gamma_4^+}. \tag{19}$$

Similarly, for the primary domain, $D_1$, the incoming radiance at all horizontal sides will be prescribed by the four auxiliary,

side RT solutions. For example, for the side shared between $D_1$ and $D_3$ we have

$$g_1^{\text{SIDE}}\left(x = 0, y, z, \boldsymbol{\Omega}\right)\Big|_{\Gamma_1^-} = \boldsymbol{I}_3\left(x = 0, y, z, \boldsymbol{\Omega}\right)\Big|_{\Gamma_3^+}. \tag{20}$$

This flow of incoming radiance is denoted by the red arrows in Fig. 2. We can see that there is a one-way propagation of radiance from the corner domains to the side domains and finally to the primary domain. This interaction is one-way due to the periodic BCs used in the solution of the auxiliary problems and ensures that the system of RTEs is solvable.

**3.1.2 Radiative Transfer Solutions**

With this basic setup, we can now describe the solution procedure of the system of RTEs. The solution to the RTEs is the first step to modelling specific instrument observables, i.e., the radiance at a particular pixel on a sensor. This forms the essence of the forward model that connects the unknown state to the measurements in AT3D. First, the corner RTEs must be solved to provide BCs to the side problems, and then these side problems must be solved to provide the BCs for the primary 3D problem.


At this point, we go into some detail below about the solution procedure, as the concepts introduced here are necessary for describing the approximate Jacobian calculation that is actually used in AT3D. In particular, in SHDOM the radiance field is decomposed into the direct solar radiance $\boldsymbol{I}_\odot(r, \boldsymbol{\Omega})$ and the diffuse, scattered radiance field $\boldsymbol{I}_{\text{d}}(r, \boldsymbol{\Omega})$:

$$I(r, \boldsymbol{\Omega}) = \boldsymbol{I}_\odot(r, \boldsymbol{\Omega}) + \boldsymbol{I}_{\text{d}}(r, \boldsymbol{\Omega}). \tag{21}$$

This is done so that the angular singularity of the solar source can be treated more accurately. We must also separate the incoming boundary source at the horizontal sides into its direct $g_\odot^{\text{SIDE}}(r, \boldsymbol{\Omega})$ and diffuse components $g_{\text{d}}^{\text{SIDE}}(r, \boldsymbol{\Omega})$:

$$g^{\text{SIDE}}(r, \boldsymbol{\Omega}) = g_\odot^{\text{SIDE}}(r, \boldsymbol{\Omega}) + g_{\text{d}}^{\text{SIDE}}(r, \boldsymbol{\Omega}). \tag{22}$$

The direct component of the boundary source at the horizontal sides is due to the propagation of the solar beam through an auxiliary RT domain to the incoming boundary of the domain under consideration. The direct radiance field is simply the

propagation of the boundary solar and direct sources into the domain, attenuated by the transmission along their optical path of propagation. On the other hand, specification of the diffuse radiation requires the solution of multiply-scattering RT problems whose source vectors are no longer angular singularities. To solve for the direct radiance field, we define the streaming operator $\mathcal{T}$ which propagates boundary radiances into the domain with the appropriate attenuation. This streaming operator maps $\Gamma_n^-$ to $D_n \times \mathbb{S}^2 \oplus \Gamma_n^\pm$, and is defined as





$\quad \mathcal{T}[I(r',\Omega')](r,\Omega) = \delta\left(\frac{r'-r}{\|r'-r\|} - \Omega'\right)\delta(\Omega - \Omega')I(r',\Omega')\,\text{T}(r,r'),$ (23)

where we have made use of the transmission between two points:

$$\text{T}(r,r') = \exp\left(-\int_0^{\|r'-r\|}\sigma\left(r' - l\frac{r'-r}{\|r'-r\|}\right)dl\right).$$ (24)

The streaming operator provides the direct solar radiance solution, namely,

$$I_\odot(r,\Omega) = \mathcal{T}\left[g_\odot^{\text{SIDE}}(r,\Omega) + g_\odot(r,\Omega)\right].$$ (25)

$\quad$ The RTE for the diffuse radiance then takes the following form

$$\mathcal{L}[I_{\text{d}}(r,\Omega)] = f_{\text{d}}(r,\Omega) = \sigma(r)\omega(r)\int_{\mathbb{S}^2}Z(r,\Omega,\Omega')\,I_\odot(r,\Omega)\text{d}\Omega' + \sigma(r)\big(1 - \omega(r)\big)B(r),$$ (26)

where $B(r)$ is an optional isotropic unpolarized black-body emission vector. The BCs of this equation are straightforward modifications to Eq. (15):

$$I_{\text{d}}(r,\Omega)|_{\Gamma^-} - \mathcal{R}[I_{\text{d}}(r,\Omega)|_{\Gamma^+}] = g_{\text{d}}(r,\Omega)|_{\Gamma^-} = \mathcal{R}[I_\odot(r,\Omega)|_{\Gamma^+}] + g^{\text{TOP}}(r,\Omega) + g^{\text{BOT}}(r,\Omega) + g_{\text{d}}^{\text{SIDE}}(r,\Omega)$$

$\qquad$ (27)

where we have defined the diffuse boundary source vectors $g_{\text{d}}^{\text{X}}(r,\Omega)$ for top (X = TOP), bottom (X = OT), and side (X = S) boundaries. This system of RTEs is solved using a solution operator $\mathcal{U}$, which is an abstract representation of the SHDOM solver, such as described in Martin et al. (2014):

$$I_{\text{d}}(r,\Omega) = \mathcal{U}\begin{bmatrix}f_{\text{d}}\\g_{\text{d}}\end{bmatrix}.$$ (28)

$\quad$ In SHDOM, all of the RTE problems are solved jointly, so that the BCs of the primary 3D problem evolve over the iterative solution procedure along with the solution for the primary diffuse radiance field.

### 3.1.3 Observable Evaluation

Now that we have outlined the solution procedure for the system of RTEs, the final step to the evaluation of the forward model is the sampling of the radiance fields by the sensors. With this final step, we will have described how observables (elements

$\quad$ of the forward model) are modelled in AT3D. We will then be able to evaluate the cost function that measures misfit between our modelled state and the measurements we are using in a given tomography problem. We will then also be able to present, in Section 3.2, the derivatives of these observables with respect to elements of the state vector which form the Jacobian matrix. These are used in AT3D to perform the tomographic retrieval.

$\quad$ The sampling operation to calculate observables can be expressed as inner products between a sensor response function and the radiance fields. Let us define the inner products on each domain in terms of test fields $v$ and $w$:





$$\langle \boldsymbol{v}, \boldsymbol{w} \rangle_{D_n \times \mathbb{S}^2} = \int_{D_n} \int_{\mathbb{S}^2} \boldsymbol{v}(\boldsymbol{r}, \boldsymbol{\Omega}) \cdot \boldsymbol{w}(\boldsymbol{r}, \boldsymbol{\Omega}) \mathrm{d}S_{\boldsymbol{\Omega}} \mathrm{d}V_r; \tag{29}$$

$$\langle \boldsymbol{v}, \boldsymbol{w} \rangle_{\Gamma_n^-} = \int_{D_n} \int_{\Omega^-} \boldsymbol{v}(\boldsymbol{r}, \boldsymbol{\Omega}) \cdot \boldsymbol{w}(\boldsymbol{r}, \boldsymbol{\Omega}) \mathrm{d}S_{\boldsymbol{\Omega}} \mathrm{d}V_r; \tag{30}$$

$$\langle \boldsymbol{v}, \boldsymbol{w} \rangle_{\Gamma_n^+} = \int_{D_n} \int_{\Omega^+} \boldsymbol{v}(\boldsymbol{r}, \boldsymbol{\Omega}) \cdot \boldsymbol{w}(\boldsymbol{r}, \boldsymbol{\Omega}) \mathrm{d}S_{\boldsymbol{\Omega}} \mathrm{d}V_r. \tag{31}$$

For fields that are defined over all on both the boundaries and interior of each domain, we also define a composite inner product over the union of the domains $D_n \times \mathbb{S}^2 \oplus \Gamma_n^{\pm}$, which is simply the sum of the right-hand sides of Eqs. (29), (30), and (31).

As an example, an observable could be a radiance exiting a domain, which could be modelled as an inner product between an, as yet unspecified, sensor response function $\boldsymbol{P}_i(\boldsymbol{r}, \boldsymbol{\Omega})$ and the radiance field, $\langle \boldsymbol{P}_i(\boldsymbol{r}, \boldsymbol{\Omega}), \boldsymbol{I}(\boldsymbol{r}, \boldsymbol{\Omega}) \rangle_{\Gamma_n^+}$. There are several
complications with this simplistic picture due to both the coupling between the different RT domains and the numerical representation of the radiance field in SHDOM. To address these issues completely, we proceed by first defining the particular sensor response functions used in AT3D. Then we describe how to evaluate radiances at particular positions within each domain with SHDOM. Finally, we describe how these components combine to provide a radiance representative of the coupled system.


The sensor response functions $\boldsymbol{P}_i(\boldsymbol{r}, \boldsymbol{\Omega})$ in the original SHDOM software (Evans, 1998) take the form of an idealized, singular sampling at position $\boldsymbol{r}_i$ and angle $\boldsymbol{\Omega}_i$ with polarization analyser $\boldsymbol{O}_i$, which is a vector that weights the contribution of the different Stokes components to each observable. For example, $\boldsymbol{O}_i = [1, 0, 0, 0]$ for an intensity measurement. In AT3D, we have generalized the sensor response function to a weighted sum over $k$ singular samplings, which we refer to as sub-pixel
rays. Each sub-pixel ray has their own position within the domain, $\boldsymbol{r}_{ik} \in D$, and angle $\boldsymbol{\Omega}_{ik}$ with weights $w_k$ that sum to unity. In this way, we can more accurately model the field of view of sensors with resolution much coarser than the resolution of the RT grid. This addition in AT3D, enables the straightforward modelling of more realistic imagers, unlike in the SHDOM software. The sensor response function is then

$$\boldsymbol{P}_i(\boldsymbol{r}, \boldsymbol{\Omega}) = \boldsymbol{O}_i \sum_k w_k \delta(\boldsymbol{r} - \boldsymbol{r}_{ik}) \delta(\boldsymbol{\Omega} - \boldsymbol{\Omega}_{ik}), \tag{32}$$

where the weights are determined by a quadrature scheme over the sensor's detectors, e.g., a 2D Gauss-Legendre quadrature over each pixel. Given the linearity of the inner product, there is an easy generalization from a single quadrature point to a sum over several, as in Eq. (32), so we simply consider sampling by just one sub-pixel ray in the following descriptions. This removes the need for summation over the $k$ sub-pixel rays but we keep the $k$ index as a subscript so that sub-pixel quantities are clearly differentiated from pixel quantities in the following discussion.


The form of Eq. (32) indicates that we only need to be able to evaluate radiances at a set of singular positions $\boldsymbol{r}_{ik}$ and angles $\boldsymbol{\Omega}_{ik}$ to evaluate the inner product. This is not as simple as it appears as, while we already have the diffuse radiance solution



from SHDOM defined at every position and direction in Eq. (28), it is insufficiently accurate for radiances at particular positions and angles. This is because the diffuse radiance is represented in SHDOM on an angularly smooth basis of spherical

harmonics that is appropriate for fluxes, but not for highly anisotropic radiances.

To sample accurate radiances at position $r_{ik}$ and angle $\Omega_{ik}$ from the RT solution, the formal solution of the RTE is used instead. To formalize this procedure, we define some quantities and operators. Firstly, there is the effective volume source and the effective boundary source of the RTE, which are defined, respectively, as:

$$\hat{f}_{\mathrm{d}}(r, \Omega) = \sigma(r)\omega(r) \int_{\mathbb{S}^2} Z(r, \Omega, \Omega') I_{\mathrm{d}}(r, \Omega') \mathrm{d}\Omega' + f_{\mathrm{d}}(r, \Omega) \text{ and} \tag{33}$$

$$\hat{g}_{\mathrm{d}}(r, \Omega)|_{\Gamma^-} = \mathcal{R}[I_{\mathrm{d}}(r, \Omega)|_{\Gamma^+}] + g_{\mathrm{d}}(r, \Omega)|_{\Gamma^-}. \tag{34}$$

The effective volume source is also commonly known as the source function of the RTE problem (e.g., Evans 1998), though we have adopted our more precise nomenclature to avoid ambiguity with other sources such as boundary sources. The two effective sources are the arguments of the formal solution of the RTE, which is stated below in Eq. (36). The effective volume

source is represented on a basis of spherical harmonics in SHDOM, which is quite accurate as the radiance fields have been angularly smoothed by convolution with the phase matrix and are therefore far less affected by truncation error.

The final definition required for the radiance calculation is that of the volume streaming operator, which integrates the effective volume source along a characteristic. It maps from $D_n \times \mathbb{S}^2$ to $D_n \times \mathbb{S}^2 \oplus \Gamma_n^+$ and is defined in terms of the test field $v$ as

$$\mathcal{S}\left[v(r', \Omega)|_{D_n \times \mathbb{S}^2}\right]_{D_n \times \mathbb{S}^2 \oplus \Gamma^+}(r, \Omega) = \int_0^{\|r'-r\|} v(r - l\Omega, \Omega)T(r - l\Omega, r)\mathrm{d}l. \tag{35}$$

The radiance at a given position $r_{ik}$ and angle $\Omega_{ik}$ is then given by

$$\langle \delta(\Omega - \Omega_{ik})\delta(r - r_{ik}), I(r, \Omega)\rangle_{D_n \times \mathbb{S}^2 \oplus \Gamma_n^\pm} = I_{\odot}(r_{ik}, \Omega_{ik}) + \mathcal{T}[\hat{g}_{\mathrm{d}}(r, \Omega)|_{\Gamma^-}](r_{ik}, \Omega_{ik}) + \mathcal{S}[\hat{f}_{\mathrm{d}}|_{D_n \times \mathbb{S}^2}](r_{ik}, \Omega_{ik}). \tag{36}$$

In our formulation, the direct component of the radiance $I_{\odot}$, which is singular in direction, is assumed to never be observed, and so we actually neglect the first term on the right-hand side of Eq. (36). This is not a strong limitation for the back or side

scattering observation geometries of Earth-viewing remote sensing instruments deployed on airborne and satellite platforms. We note that SHDOM employs the Truncated Multiple Scattering (TMS) correction for forward scattering media which, while accurate in backscattering directions (except exact backscatter), can have significant inaccuracies near the solar direction (Nakajima and Tanaka, 1988). Therefore– further extension of this retrieval method to include measurements of the direct solar radiance should also include improvements to the SHDOM solver itself.


The evaluation of an element of the forward model involves contributions from these streaming operations from each domain along the line of sight of the sensor. Not all domains contribute similarly. This is because we would like to treat the coupled



system as one cohesive approximation to the horizontally infinite atmosphere in the evaluation of the forward model by applying the formal solution of the RTE across all unified domains, ignoring all horizontal boundaries. This means that all

domains that intersect the line of sight will contribute their effective volume source to the observable. Expressing this precisely takes some complexity, as we wish to express the radiance as a sum over inner products over each domain so that we can easily express differentiation of the observables with respect to the state vector following Martin et al. (2014).

To begin, let us consider the example shown in Fig. 4 before introducing the general mathematical description, which follows

in Eq. 41. The mathematical expression specific to this example is shown on Fig. 4. We want to calculate a radiance at the position $r_{ik}$ and direction $\Omega_{ik}$ of a sub-pixel ray. This sub-pixel ray has a line of sight that points in the opposite direction to the propagation of the radiance that is sampled by the ray, $-\Omega_{ik}$. In Fig. 4, this line of sight is presented as the grey shading. We can see that the line of sight of the sub-pixel ray overlaps the three domains; $D_3$, $D_1$, and $D_5$, and so they will all contribute to the radiance calculation.

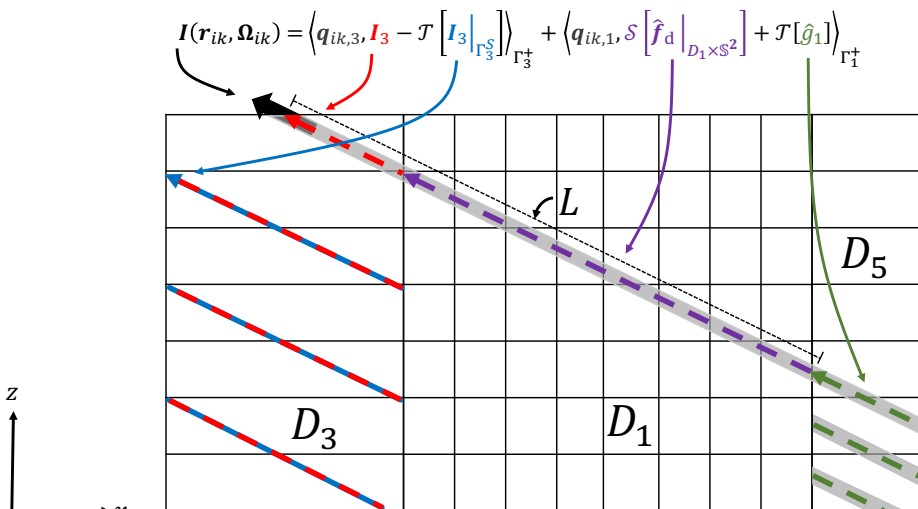


**Figure 4: A side view of the system of RTE domains, illustrating the differences in how radiances are calculated during the solution of the RTEs in each domain and during the evaluation of the forward model. Consider the calculation of the radiance at the position $r_{ik}$ and angle $\Omega_{ik}$ at the upper boundary of $D_3$ denoted by the large black arrow. During the solution of the RTE in $D_3$, the radiance is calculated by integration along the red characteristic that follows the periodic BCs at the edge of $D_3$. On the other hand, when the**

**forward model is evaluated, the radiance is calculated through integration along the characteristic denoted by gray shading which passes through $D_3$ into $D_1$ and $D_5$. The correspondence between the mathematical expressions for the evaluation of the forward model (Eq. 41) in this case and the graphical illustration are shown. See the main text for additional details.**

The line of sight first intersects $D_3$. All we actually want from $D_3$ is the contribution to the radiance from the effective volume source in the region of overlap between the red-dashed line and the grey shading. Let us call this contribution the "overlap

contribution". If we were to just evaluate the radiance at position $r_{ik}$ and direction $\Omega_{ik}$ using only the RT solution of $D_3$ using Eq. 36 we would be following the periodic boundary conditions of $D_3$ with the integration domain of the streaming operator



denoted by the red arrow on Fig. 4. Let us call this radiance the "$D_3$ radiance". To isolate the overlap contribution, we must additionally calculate the periodic boundary radiance which is in blue in Fig. 4. This boundary radiance, attenuated along the line of sight to $\boldsymbol{r}_{ik}$, must then be subtracted from the "$D_3$ radiance" to give the overlap contribution, which is a portion of the observable.


We then need to add the contributions to the observable from $D_1$ by evaluating Eq. 36 in that domain, which is shown in purple in Fig. 4. The line of sight intersects the horizontal boundary of $D_1$. The evaluation of the boundary term in Eq. 36, which is the second term on the right-hand side, requires a contribution from $D_5$. Following the open boundary conditions (e.g., Eq. 18), the incoming radiance at the horizontal boundaries of $D_1$ is simply the radiance at the boundary of $D_5$, calculated using Eq. 36. This calculation requires integration along the green chord.


To write this mathematically, we first need to isolate the incoming set on the horizontal sides, which is a subset of the incoming set defined in Eq. (11):

$$\Gamma_n^S = \{(\boldsymbol{r}, \boldsymbol{\Omega}) \in \Gamma_n^-: 0 < z < L_z\}. \tag{37}$$


We then need to define an indicator function for the domains for which we need to calculate direct radiance contributions. For the example above the domains that directly contribute are $D_1$ and $D_3$. $D_5$ does not directly contribute, rather it contributes through the open boundary condition of $D_1$. This distinction is important for linearization of the forward model. We use an indicator function to separate between these cases, which takes the form of a Heaviside function along the line of sight.


To define the indicator, we need to define a length denoted by $L$, which is the distance until the intersection of the line of sight with a boundary of a domain with a lower dimensionality of radiative transport than the current domain. This intersection marks the point where the domains contribute indirectly through the open boundary conditions. Recall that the corner domains are 1D, the side domains are 2D and the primary domain is 3D. For the example in Fig. 4, we would find the intersection at the boundary between $D_1$ and $D_5$. The corresponding length $L$ is illustrated in Fig. 4. We can then define the indicator function as

$$H_L(x) = \begin{cases} 1, & 0 \leq x < L \\ 0, & \text{otherwise} \end{cases}. \tag{38}$$

Note that the indicator function is not inclusive. Now, we can define the volume response functions $\boldsymbol{p}_{ik,n}(\boldsymbol{r}, \boldsymbol{\Omega})$ and the attenuated boundary response function $\boldsymbol{q}_{ik,n}(\boldsymbol{r}, \boldsymbol{\Omega})$ for the sub-pixel ray for each of the $n$ domains.

$\quad \boldsymbol{p}_{ik,n}(\boldsymbol{r}, \boldsymbol{\Omega}) = \mathbf{O}_i \delta(\boldsymbol{r}_{ik} - \boldsymbol{r}) \delta(\boldsymbol{\Omega} - \boldsymbol{\Omega}_{ik}) \quad \text{in } D_n \times \mathbb{S}^2 \text{ and} \hfill (39)$

$\quad \boldsymbol{q}_{ik,n}(\boldsymbol{r}, \boldsymbol{\Omega}) = H_L(\|\boldsymbol{r}_{ik} - \boldsymbol{r}\|) \mathbf{O}_i \delta\left(\frac{\boldsymbol{r}_{ik} - \boldsymbol{r}}{\|\boldsymbol{r}_{ik} - \boldsymbol{r}\|} - \boldsymbol{\Omega}_i\right) \delta(\boldsymbol{\Omega} - \boldsymbol{\Omega}_{ik}) \mathrm{T}(\boldsymbol{r}, \boldsymbol{r}_{ik}) \quad \text{on } \boldsymbol{\Gamma}_n^+. \hfill (40)$





These two response functions will sample the radiance within each domain if the sensor is internal to a domain and also at the boundaries of all of the domains along the line of sight between the sensor and the primary domain, weighted by the transmission to the sensor. In the example in Fig. 4, $\boldsymbol{q}_{ik,1}$, and $\boldsymbol{q}_{ik,3}$ are non-zero while $\boldsymbol{q}_{ik,5}$ and all other attenuated boundary
response functions are zero. Additionally, all $\boldsymbol{p}_{ik,n}$ are zero in the example in Fig. 4 as $\boldsymbol{r}_{ik}$ is not interior to any domain.

The forward model is then expressed as

$$F_i(\boldsymbol{x}) = \langle \boldsymbol{p}_{ik,1}, \boldsymbol{I}_{1,\mathrm{d}} \rangle_{D_1 \times \mathbb{S}^2} + \langle \boldsymbol{q}_{ik,1}, \boldsymbol{I}_{1,\mathrm{d}} \rangle_{\Gamma_1^+} + \sum_{n=2}^{9} \left( \langle \boldsymbol{p}_{ik,n}, \ \boldsymbol{I}_{n,\mathrm{d}} - \mathcal{T}[\boldsymbol{I}_{n,\mathrm{d}}|_{\Gamma_n^S}] \rangle_{D_n \times \mathbb{S}^2} + \langle \boldsymbol{q}_{ik,n}, \boldsymbol{I}_{n,\mathrm{d}} - \mathcal{T}[\boldsymbol{I}_{n,\mathrm{d}}|_{\Gamma_n^S}] \rangle_{\Gamma_n^+} \right). \ (41)$$

The inner products over the internal sets will be non-zero only if the sensor is internal to that domain, and will therefore be
non-zero only for one domain. The inner products over the boundaries will be non-zero only when the line of sight of the ray intersects the outgoing boundary of a domain. The first two terms on the right-hand-side of Eq. (41) describe the sampling of the radiance field in the primary domain of 3D RT ($D_1$), which follows the formulation of Martin et al. (2014).

The second set of inner products shows the contributions of the auxiliary domains. We have removed the component from the
radiance field that originates from the periodic horizontal BCs. The resulting expressions are still differentiable, though the existence of an adjoint formulation has not been proven. We do not consider the adjoint formulation of this system here. This forward model is valid for all radiances except those that are measured exactly in the horizontal plane, where it is undefined just like for periodic horizontal BCs. We now have a full description of the forward model $F(\boldsymbol{a})$ used in AT3D to connect radiometric observables to the state of the atmosphere.


The derivatives of those inner products with respect to the state vector can be expressed using tangent-linear or forward-adjoint principles, as long as care is taken to address the sensitivity of this RT solution to changes in the incoming boundary radiance. This extension to the optimization of the BCs is not discussed in Martin et al. (2014), but is relatively straightforward and will be described below.

**3.2 Linearization of the Forward Model**

We can now calculate derivatives of the forward model with respect to a component of the state vector $a_j$, which form the essence of the tomographic retrieval employed in AT3D. These derivatives tell us how to optimally adjust the state vector to better match the measurements. We compute these derivatives for the first set of inner products in Eq. (41) over the primary 3D domain following Martin et al. (2014). Let $F_{i,n}(\boldsymbol{a})$ refer to the contribution of the $n^{\mathrm{th}}$ RTE domain to the forward model
output. For $n = 1$, we have

$$\frac{\partial F_{i,1}(\boldsymbol{a})}{\partial a_j} = \left\langle \boldsymbol{p}_i, \frac{\partial \boldsymbol{I}_{1,\mathrm{d}}}{\partial a_j} \right\rangle_{D_1 \times \mathbb{S}^2} + \left\langle \boldsymbol{q}_i, \frac{\partial \boldsymbol{I}_{1,\mathrm{d}}}{\partial a_j} \right\rangle_{\Gamma_+^1}. \tag{42}$$



These inner products are evaluated by solving a modified RTE problem for the derivatives of the diffuse radiance. The following expression holds for an arbitrary RTE domain, and are formed by differentiating the RTE (Eq. 26) and regrouping terms to find the volume source vector of the modified RTE for the derivatives of the radiance field, $\Delta f_j$:

$$\mathcal{L}\left[\frac{\partial I_{\mathrm{d}}(r,\Omega)}{\partial a_j}\right] = -I_{\mathrm{d}}\frac{\partial\sigma}{\partial a_j} + \left(\frac{\partial\sigma}{\partial a_j}\omega + \frac{\partial\omega}{\partial a_j}\sigma\right)\int_{\mathbb{S}^2} Z\, I_{\mathrm{d}}\mathrm{d}\Omega' + \sigma\omega\int_{\mathbb{S}^2}\frac{\partial Z}{\partial a_j}I_{\mathrm{d}}\mathrm{d}\Omega' + I_{\odot}(r,\Omega)\left[\frac{\partial\sigma}{\partial a_j}\omega Z + \frac{\partial\omega}{\partial a_j}\sigma Z + \frac{\partial Z}{\partial a_j}\sigma\omega\right] +$$

$$\left(\frac{\partial\sigma}{\partial a_j}(1-\omega) - \frac{\partial\omega}{\partial a_j}\sigma\right)B + \sigma(1-\omega)\frac{\partial B}{\partial a_j} + \frac{\partial I_{\odot}(r,\Omega)}{\partial a_j}\omega\sigma Z = \Delta f_j \tag{43}$$

We also differentiate the BCs of the RTE (Eq. 27) to obtain the associated boundary source vector of the modified RTE, $\Delta g_j$:

$$\left.\frac{\partial I_{\mathrm{d}}}{\partial a_j}\right|_{\Gamma_-} - \mathcal{R}\left[\left.\frac{\partial I_{\mathrm{d}}}{\partial a_j}\right|_{\Gamma_+}\right] = \mathcal{R}\left[\left.\frac{\partial I_{\odot}}{\partial a_j}\right|_{\Gamma_+}\right] + \frac{\partial\mathcal{R}}{\partial a_j}[I_{\odot} + I_{\mathrm{d}}] + \frac{\partial g^{\mathrm{TOP}}(r,\Omega)}{\partial a_j} + \frac{\partial g^{\mathrm{BOT}}(r,\Omega)}{\partial a_j} + \frac{\partial g_{\mathrm{d}}^{\mathrm{SIDE}}(r,\Omega)}{\partial a_j} = \Delta g_j. \tag{44}$$

The derivative of the reflection operator (Eq. 16) is

$$\frac{\partial\mathcal{R}}{\partial a_j}[I] = \int_{\Omega+}|n\cdot\Omega'|\frac{\partial R(r,\Omega,\Omega')}{\partial a_j}I(r,\Omega')\mathrm{dS}_{\Omega'}. \tag{45}$$

The solution to these modified systems is then

$$\frac{\partial I_{\mathrm{d}}(r,\Omega)}{\partial a_j} = \mathcal{U}\begin{bmatrix}\Delta f_j \\ \Delta g_j\end{bmatrix} \tag{46}$$

Equation (46) constitutes a tangent linear model to SHDOM (Eq. 28). To evaluate $\Delta f_j$, we just need the solution of the forward RTE problem and the optical property derivatives. We also have to calculate the derivatives of the direct solar beam, that is,

$$\frac{\partial I_{\odot}(r,\Omega)}{\partial a_j} = \frac{\partial\mathcal{T}}{\partial a_j}\left[g_{\odot}^{\mathrm{S}}(r,\Omega) + g_{\odot}(r,\Omega)\right] + \mathcal{T}\left[\frac{\partial g_{\odot}^{\mathrm{SIDE}}(r,\Omega)}{\partial a_j}\right]. \tag{47}$$

The derivative of the streaming operator (Eq. 23) simply follows from the derivative of the transmission function (Eq. 24):

$$\frac{\partial\mathcal{T}}{\partial a_j}[I(r',\Omega')](r,\Omega) = -\mathcal{T}[I(r',\Omega')](r,\Omega)\int_0^{\|r'-r\|}\frac{\partial\sigma}{\partial a_j}\left(r - l\frac{r'-r}{\|r'-r\|}\right)dl'. \tag{48}$$

The remaining terms to be specified are the derivatives of the boundary source vectors on the domain top, $\frac{\partial g^{\mathrm{TOP}}(r,\Omega)}{\partial a_j}$, bottom, $\frac{\partial g^{\mathrm{BOT}}(r,\Omega)}{\partial a_j}$, , and sides, $\frac{\partial g_{\mathrm{d}}^{\mathrm{SIDE}}(r,\Omega)}{\partial a_j}$ and $\frac{\partial g_{\odot}^{\mathrm{SIDE}}(r,\Omega)}{\partial a_j}$. The first two terms are determined by local analytic relationships with the state, so that their derivatives can be readily calculated. For instance, the surface emission source on the right-hand side of Eq. (15) depends on temperature, and the surface BRDF in Eq. (16) on the left-hand side may also be parameterized by elements of the state vector. The second two terms involve the side boundary sources, which are non-zero only for the open boundaries, and involve coupling between the different RTE solutions. The direct component, $\frac{\partial g_{\odot}^{\mathrm{SIDE}}(r,\Omega)}{\partial a_j}$, is just the direct radiance derivative, $\frac{\partial I_{\odot}(r,\Omega)}{\partial a_j}$, on the adjoining boundary, and can therefore be calculated through recursive application of Eq.



(46) across each domain until the direct solar source is only the domain top, where $\frac{\partial g_\odot^{\text{SIDE}}(r,\Omega)}{\partial a_j} = 0$. We do not need to differentiate the domain-top solar source, $g_\odot(r,\Omega)$, which is fixed and known. The derivatives of the diffuse boundary source are more complex to calculate, requiring their own RTE solutions. Specifically, the derivatives of the diffuse boundary source on the 3D primary domain $D_1$ requires RTE solutions on the side domains (e.g., $D_3$). The derivatives of the diffuse boundary

source on the side domains requires RTE solutions on the corner domains (e.g., $D_2$ and $D_4$). We can then evaluate

$$\frac{\partial g_{1,\text{d}}^{\text{SIDE}}(r,\Omega)}{\partial a_j} = \left\langle \delta(x), \frac{\partial I_{3,\text{d}}}{\partial a_j} \right\rangle_{\Gamma_+^3} + \left\langle \delta(y), \frac{\partial I_{5,\text{d}}}{\partial a_j} \right\rangle_{\Gamma_+^5} + \left\langle \delta(x - L_x), \frac{\partial I_{7,\text{d}}}{\partial a_j} \right\rangle_{\Gamma_+^7} + \left\langle \delta(y - L_y), \frac{\partial I_{9,\text{d}}}{\partial a_j} \right\rangle_{\Gamma_+^9} \tag{49}$$

For each of the side domains, we then have, for example,

$$\frac{\partial g_{3,\text{d}}^{\text{SIDE}}(r,\Omega)}{\partial a_j} = \left\langle \delta(y - L_y), \frac{\partial I_{2,\text{d}}}{\partial a_j} \right\rangle_{\Gamma_+^2} + \left\langle \delta(y), \frac{\partial I_{4,\text{d}}}{\partial a_j} \right\rangle_{\Gamma_+^4}. \tag{50}$$

We can now evaluate all of the derivatives of the forward model:

$$\frac{\partial F_i(x)}{\partial a_j} = \left\langle p_{i,1}, \frac{\partial I_{1,\text{d}}}{\partial a_j} \right\rangle_{D_1 \times \mathbb{S}^2} + \left\langle q_{i,1}, \frac{\partial I_{1,\text{d}}}{\partial a_j} \right\rangle_{\Gamma_1^+} + \sum_{n=2}^9 \left( \left\langle p_{i,n}, \frac{\partial I_{n,\text{d}}}{\partial a_j} - \mathcal{T}\left[ \frac{\partial I_{n,\text{d}}}{\partial a_j} |_{\Gamma_n^S} \right] - \frac{\partial \mathcal{T}}{\partial x_j} [I_{n,\text{d}} |_{\Gamma_n^S}] \right\rangle_{D_n \times \mathbb{S}^2} + \left\langle q_{i,n}, \frac{\partial I_{n,\text{d}}}{\partial a_j} - \right. \right.$$

$$\left. \left. \mathcal{T}[\frac{\partial I_{n,\text{d}}}{\partial a_j} |_{\Gamma_n^S}] - \frac{\partial \mathcal{T}}{\partial a_j} [I_{n,\text{d}} |_{\Gamma_n^S}] \right\rangle_{\Gamma_n^+} \right). \tag{51}$$

This is an exact treatment of the derivatives of the forward model, and is what would be numerically approximated by performing finite differencing of the forward model. This is not the approach directly employed in AT3D

### 3.3 Approximate Jacobian calculation

In AT3D, we evaluate approximate derivatives of the forward model instead of exactly evaluating Eq. (51). This is done for reasons of algorithmic simplicity and computational efficiency, but naturally can have consequences for the accuracy of the retrieval. This is quite common in optimization problems (Ye et al., 1999; Eppstein et al., 2003; Dwight and Brezillon, 2006), as convergence is the key criterion to measure success of an optimization based retrieval, rather than a high-accuracy solution to any particular linearized problem. The approximate derivatives use approximate solutions to the RTEs for the radiance

derivatives. Specifically, the approximation to the derivatives uses a no-scattering assumption in the solution of the tangent linear model and resulting evaluation of the radiance fields following Levis et al. (2020). The no-scattering assumption is equivalent to a zeroth order approximation to a successive-order-of-scattering solution to Eq. 46. It does not involves setting the single scattering albedo to zero. The result is that the multiple-scattering tangent-linear model in Eq. (46) does not need to be evaluated, which saves the computational expense of evaluating an additional 3D RT model at each iteration to calculate

derivatives. The approximate radiance derivatives require only the evaluation of the formal, integral solutions which are just line integrations with the same geometry as the forward model (see Fig. 4). This gives us an easy way to adjust the unknown state to better match the measurements and, from there, perform the tomographic retrieval. This key approximation is the essence of AT3D, and makes it computationally suitable for solving practical tomography problems.





Let us formalize the approximate derivative calculation. We define the effective volume source and the effective boundary source of the radiance derivative RTE analogously to in the forward solution (Eq. 33 and 34) as,

$$\Delta \hat{f}_j(r,\Omega) = \sigma(r)\omega(r)\int_{\mathbb{S}^2} Z(r,\Omega,\Omega')\frac{\partial I_d}{\partial a_j}(r,\Omega')d\Omega' + \Delta f_j(r,\Omega) \text{ and} \tag{52}$$

$$\Delta \hat{g}_j(r,\Omega)|_{\Gamma^-} = \mathcal{R}\left[\frac{\partial I_d}{\partial a_j}(r,\Omega)|_{\Gamma^+}\right] + \Delta g_j(r,\Omega)|_{\Gamma^-}. \tag{53}$$

The inner products in Eq. (51) may then be evaluated following the same rule for singular sampling as in the forward model
(Eq. 36):

$$\langle\delta(\Omega - \Omega_{ik})\delta(r - r_{ik}), \frac{\partial I_{n,d}}{\partial a_j}\rangle_{D_n\times\mathbb{S}^2\oplus\Gamma_n^\pm} = \mathcal{T}\left[\Delta\hat{g}_d(r,\Omega)|_{\Gamma_n^-}\right](r_{ik},\Omega_{ik}) + \mathcal{S}\left[\Delta\hat{f}_j|_{D_n\times\mathbb{S}^2}\right](r_{ik},\Omega_{ik}). \tag{54}$$

In AT3D, Eq. (54) is approximated by invoking the no-scattering assumption, so the effective volume and boundary sources contain no recursive dependence on $\frac{\partial I_d}{\partial a_j}$. In particular, we approximate the effective volume source for the radiance derivatives as

$$\Delta\hat{f}_j|_{D_n\times\mathbb{S}^2} \approx \Delta f_j(r,\Omega). \tag{55}$$

We also neglect the first term on the right-hand side of Eq. (53), which is the reflection of diffusely scattered radiance derivatives at the boundary. These two no-scattering approximations are applied for a given domain for the evaluation of integrals of the form of Eq. (54). However, there is also an effect through the BCs that further approximates $\Delta g_j(r,\Omega)|_{\Gamma^-}$, due to the application of the no-scattering approximation to all domains. The diffuse horizontal boundary sources for the
radiance derivatives in Eq. (44), $\frac{\partial g_{n,d}^{SIDE}(r,\Omega)}{\partial a_j}$, are calculated by applying the same singular radiance derivative sampling expressed in Eq. (54) on the other domains. For a side domain (e.g., $D_3$), the horizontal diffuse boundary radiance comes from only corner domains (e.g., $D_2$ and $D_4$), which themselves do not have open horizontal BCs. As such, the diffuse horizontal boundary sources for radiance derivatives in Eq. (50) are approximated as:

$$\frac{\partial g_{3,d}^{SIDE}(r,\Omega)}{\partial a_j} = \left\langle\delta(y-L_y),\frac{\partial I_{2,d}}{\partial a_j}\right\rangle_{\Gamma_+^2} + \left\langle\delta(y),\frac{\partial I_{4,d}}{\partial a_j}\right\rangle_{\Gamma_+^4} \approx \frac{\partial g_{3,d}^{SIDE,0}(r,\Omega)}{\partial a_j} = \left\langle\delta(y-L_y),\mathcal{T}\left[\Delta g_j|_{\Gamma^-}\right](r,\Omega) + \right.$$

$$\left. \mathcal{S}\left[\Delta f_j|_{D_2\times\mathbb{S}^2}\right](r,\Omega)\right\rangle_{\Gamma_+^2} + \left\langle\delta(y),\mathcal{T}\left[\Delta g_j|_{\Gamma^-}\right](r,\Omega) + \mathcal{S}\left[\Delta f_j|_{D_4\times\mathbb{S}^2}\right](r,\Omega)\right\rangle_{\Gamma_+^4}. \tag{56}$$

Note that, when compared to Eq. (54), we have made the approximation that $\Delta\hat{g}_j(r,\Omega)|_{\Gamma^-} \approx \Delta g_j(r,\Omega)|_{\Gamma^-}$ in Eq. (56) in addition to Eq. (55). The simpler form of the boundary term is because the corner domains have no horizontal sides, so $\frac{\partial g_{n,d}^{SIDE}(r,\Omega)}{\partial a_j} = 0$, without requiring further approximation. For the side domains (e.g., $D_2$ and $D_4$) and primary domain ($D_1$), the approximation of $\frac{\partial g_{n,d}^{SIDE}(r,\Omega)}{\partial a_j}$ with $\frac{\partial g_{n,d}^{SIDE,0}(r,\Omega)}{\partial a_j}$ leads to the approximation of the effective boundary source as





$\quad \Delta \widehat{g}_{n,j}(r, \Omega)|_{\Gamma_n^-} \approx \Delta \widehat{g}_{n,j}^0(r, \Omega)|_{\Gamma_n^-} = \mathcal{R}\left[\frac{\partial I_\odot}{\partial a_j}\Big|_{\Gamma_+^n}\right] + \frac{\partial \mathcal{R}}{\partial a_j}\left[I_{n,\odot} + I_{n,d}\right] + \frac{\partial g_n^{TOP}(r,\Omega)}{\partial a_j} + \frac{\partial g_n^{BOT}(r,\Omega)}{\partial a_j} + \frac{\partial g_n^{SIDE,0}(r,\Omega)}{\partial a_j}.$ (57)

With this, the approximate radiance derivatives in any domain at position $r_{ik}$ and direction $\Omega_{ik}$ are calculated as

$\langle \delta(\Omega - \Omega_{ik})\delta(r - r_{ik}), \frac{\partial I_{n,d}}{\partial a_j}\rangle_{D_n \times \mathbb{S}^2 \oplus \Gamma_{\tilde{n}}^\pm} \approx \mathcal{T}[\Delta\widehat{g}_{n,j}^0(r,\Omega)|_{\Gamma_n^-}](r_{ik}, \Omega_{ik}) + \mathcal{S}[\Delta f_j|_{D_n \times \mathbb{S}^2}](r_{ik}, \Omega_{ik}),$ (58)

with the appropriate recursive calculation of $\Delta\widehat{g}_{n,j}^0$. Numerical implementation of these integrals in Eq. (58) is described in Appendix C. The computational cost of this calculation relative to the evaluation of radiances (Eq. 41) is presented in Appendix

F.

As this series of approximations follows from a no-scattering assumption in the tangent-linear model, the error in the resulting radiance derivatives, and therefore forward model derivatives, will depend on the relative contribution of higher orders of scatter to the radiance derivative at the positions and angles sampled by the sensor. When the single scattering albedo is near

unity, these contributions will be most significant. They will also be relatively large when the optical path between the source and sensor is large as radiance reaching the sensor will necessarily have undergone many scattering events. The approximation is clearly appropriate for emission problems without scattering or in the single scattering limit, where it is also exact. For highly scattering solar transport problems, this approximation requires some justification. The approximation of the volume derivative source term in Eq. (55) is the same as described in Levis et al. (2020). The treatment of the surface derivative source

term here is an extension from Levis et al. (2020), as $\Delta\widehat{g}_j$ was assumed to vanish in that formulation. We note that the treatment of the volume derivative source term is identical to the formulation of Zawada et al. (2017), where it is appropriate in the quasi-single scattering context of limb scattering.

In this section, we have described the tomographic retrieval methodology, including a full description of the forward model

and its approximate linearization, which is utilized in AT3D for computationally efficient solutions to the retrieval problem. In the following section, we present a linear analysis of the conditioning of the inverse radiative transfer problem and other properties of the exact Jacobian matrix. This provides us with a more detailed framework for understanding the limitations of the cloud tomography method and how it will generalize to the full range of scattering regimes in the atmosphere. We will then apply this framework in Section 5 to understand in more detail the quantitative behaviour of the approximate linearization

described here.

**4 Linearized Inverse Radiative Transfer**

Our analysis in this section is focused on describing the information content about the spatial variability of optical properties contained within multi-angle measurements, and therefore is focused on the inversion of the RT within a linearized context. We begin this analysis by first describing the conditioning of the exact Jacobian matrix, measured by its condition number



(Eq. 6). Studying the exact Jacobian matrix provides a point of reference for understanding the effects of using approximate Jacobian matrix. Through the comparison, we can identify which limitations of tomography are physical and which are a result of using an approximate Jacobian. This section contains some results that have wider implications for tomographic retrievals and also some which are specific to the SHDOM solver. The implications of these results are discussed in Section 4.3.

The structure of the inverse problem and the nature of the Jacobian approximation used in AT3D can most easily be conceptualized if we use the equivalent adjoint formulation of the inner products, for reasons that we make clear below. This formulation will provide the basis for our presentation of the qualitative theory of inverse RT. Here we give only a brief summary of the adjoint formulation of the Jacobian calculations. More details on the formulation of the adjoint problem can be found in Martin et al. (2014) for 3D vector RT, or in similar formulations for plane-parallel media (Hasekamp and Landgraf,

2005). In the adjoint formulation, the inner products in Eq. 51 are expressed in terms of an adjoint radiance field whose sources are now the sensor response functions, $\boldsymbol{p}_i$ and $\boldsymbol{q}_i$ (instead of the Sun). The adjoint RT problem is solved using an adjoint solution operator $\mathcal{U}^*$, which is adjoint to the forward solution operator $\mathcal{U}$. In practice, the adjoint RTE is not directly solved. Instead, a pseudo-forward RTE problem is solved whose source is the direction-reversed and polarization-flipped sensor response functions $\boldsymbol{p}_i^\dagger$ and $\boldsymbol{q}_i^\dagger$:

$$\boldsymbol{p}_i^\dagger = \boldsymbol{Q}^\dagger \boldsymbol{p}_i(\boldsymbol{r}, -\boldsymbol{\Omega}) \text{ and} \tag{59}$$

$$\boldsymbol{q}_i^\dagger = \boldsymbol{Q}^\dagger \boldsymbol{q}_i(\boldsymbol{r}, -\boldsymbol{\Omega}), \tag{60}$$

where

$$\boldsymbol{Q}^\dagger = \begin{pmatrix} 1 & & & \\ & 1 & & \\ & & 1 & \\ & & & -1 \end{pmatrix}. \tag{61}$$

The pseudo-forward problem is solved by the forward solution operator $\mathcal{U}$ so it can, in principle, be solved by a forward model

like SHDOM as long as its implementation supports sufficiently general volume and boundary source vectors. In practice, this pseudo-forward problem is much more numerically challenging due to the spatio-angular singularity of the sources $\boldsymbol{p}_i^\dagger$ and $\boldsymbol{q}_i^\dagger$ in 3D (Doicu and Efremenko, 2019; Martin and Hasekamp, 2018). The adjoint radiance field can then be found by direction reversal and polarization flipping of the pseudo-forward radiance $\boldsymbol{I}^\dagger(\boldsymbol{r}, \boldsymbol{\Omega})$.

$$\boldsymbol{I}^\dagger(\boldsymbol{r}, \boldsymbol{\Omega}) = \mathcal{U} \begin{bmatrix} \boldsymbol{p}_i^\dagger \\ \boldsymbol{q}_i^\dagger \end{bmatrix} \tag{62}$$

We can then express the inner products for the forward model derivatives as

$$\langle \boldsymbol{p}_i, \tfrac{\partial I_d}{\partial a_j} \rangle_{D_n \times \mathbb{S}^2} + \langle \boldsymbol{q}_{i,1}, \tfrac{\partial I_d}{\partial a_j} \rangle_{\Gamma_n^+} = \langle \boldsymbol{Q}^\dagger \boldsymbol{I}_i^\dagger(\boldsymbol{r}, -\boldsymbol{\Omega}), \Delta \boldsymbol{f}_j \rangle_{D_n \times \mathbb{S}^2} + \langle \boldsymbol{Q}^\dagger \boldsymbol{I}_i^\dagger(\boldsymbol{r}, -\boldsymbol{\Omega}), \Delta \boldsymbol{g}_j \rangle_{\Gamma_n^+} \tag{63}$$





Given the singular form of the sensor response functions, the RTE problem for the pseudo-forward radiance is a pencil-beam illumination problem. This is a well-studied RT problem (Doicu et al., 2020; Liemert and Kienle, 2013; Martelli et al., 2016), as its solution is the Green's function for 3D RT. The approximate Jacobian in the 3D RT domain is equivalent to approximating the pseudo-forward solution by its direct-beam solution

$$\langle \boldsymbol{p}_i, \mathcal{T}[\Delta \boldsymbol{g}_j^0(\boldsymbol{r},\boldsymbol{\Omega})|_{\Gamma_1^-}](\boldsymbol{r},\boldsymbol{\Omega}) + \mathcal{S}[\Delta \boldsymbol{f}_j|_{D_1 \times \mathbb{S}^2}]\rangle_{D_1 \times \mathbb{S}^2} + \langle \boldsymbol{q}_i, \mathcal{T}[\Delta \boldsymbol{g}_j^0(\boldsymbol{r},\boldsymbol{\Omega})|_{\Gamma_1^-}](\boldsymbol{r},\boldsymbol{\Omega}) + \mathcal{S}[\Delta \boldsymbol{f}_j|_{D_1 \times \mathbb{S}^2}]\rangle_{\Gamma_1^+} =$$

$$\langle \boldsymbol{Q}^\dagger \mathcal{T}[\boldsymbol{p}_i^\dagger](\boldsymbol{r},-\boldsymbol{\Omega}), \Delta \boldsymbol{f}_j\rangle_{D_1 \times \mathbb{S}^2} + \langle \boldsymbol{Q}^\dagger \mathcal{T}[\boldsymbol{q}_i^\dagger](\boldsymbol{r},-\boldsymbol{\Omega}), \Delta \boldsymbol{g}_j^0\rangle_{\Gamma_1^+}. \tag{64}$$

With the forward-adjoint formulation described above, each of the columns of the Jacobian matrix corresponds to a different observing geometry, and therefore to a different "pencil-beam" pseudo-forward RTE solution. We can use this to express the independence of columns in the Jacobian matrix in terms of the independence of the pseudo-forward radiance fields, weighted by the radiance derivative source vectors $\Delta \boldsymbol{f}_j$ and $\Delta \boldsymbol{g}_j$. In this way we link the properties of the Jacobian matrix, such as its condition number, which controls the difficulty of the retrieval problem, to the optical properties of the medium using arguments from RT theory. To be clear, we use the linearized framework to classify the difficulty of the iterative optimization process based on the current state vector. The linearization around the ground truth does provide some information on the difficulty of the inverse problem but we must bear in mind that, given the non-convex nature of the inversion, the iterative retrieval may encounter difficult cost function structure far from the ground truth, depending on the optimization trajectory. This linearized framework provides us with some hypotheses about the behaviour of AT3D's tomographic retrieval that can be used to explain the behaviour of fully nonlinear retrievals in terms of physical principles.

**4.1 Numerically evaluating the Jacobian matrix**

We support the physical arguments presented in this section with quantitative evidence from numerical experiments. The quantitative evidence is produced through the numerical calculation of reference Jacobian matrices using a 2-point central difference around a wide range of media or base states. These reference Jacobian matrices are also used to quantitatively evaluate the approximate Jacobian calculation in Section 5. The numerical experiments that we use are as follows.

Each Jacobian matrix is calculated around a reference configuration of the state vector, referred to as a base state. Each base state is a 3D gaussian extinction field embedded in a uniform extinction field evaluated on a grid with 50 m resolution and 21 grid points in each direction, hence primary domain dimensions $L_x = L_y = L_z = 1$ km. Open horizontal BCs are used. This configuration is chosen as a simple analogue of an isolated cloud with a sufficiently high resolution that we can study how the elements of Jacobian matrix vary with position within the cloud. The extinction field at each grid point is given by

$$\sigma(x,y,z) = \sigma_{\text{bg}} + A \exp\left(\frac{(x-x_0)^2 + (y-y_0)^2 + (z-z_0)^2}{r^2}\right), \tag{65}$$


where $\sigma_{\mathrm{bg}} = 0.1$ km$^{-1}$ is a uniform background extinction, $(x_0, y_0, z_0) = (L_x, L_y, L_z)/2$ is the centre of the computational domain, and $r = 0.2$ km (i.e., 4 grid cells $= L_z/5$). The constant $A$ (in km$^{-1}$) is varied so that the maximum vertical optical path of the gaussian is 0.1, 5.0, 40.0 and 100.0. When we include the uniform background, which contributes an optical depth of $\sigma_{\mathrm{bg}}L_z$, the maximum vertical optical paths $\tau_{max} = \int_0^{L_z} \sigma(x_0, y_0, z)dz$, is 0.1, 5.0, 40.0 and 100.0 of the medium are 0.2,

5.1, 40.1 and 100.1. But, in our following results we label our base states by the maximum optical path of the gaussian.

Each base state has a spatially uniform single scattering albedo and phase function and a uniform Lambertian surface albedo. The single scattering albedos of the base states are 0.9, 0.99 and 1.0, and the surface albedos are 0.0, 0.2, and 0.7. Three different combinations of phase function and angular resolution are tested. The first uses an isotropic phase function with 16

zenith discrete ordinate bins and 32 azimuthal discrete ordinate bins. The second uses a Mie phase function equivalent to a gamma droplet size distribution with effective radius $r_e = 10$ $\mu m$ and effective variance $v_e = 10$ evaluated at a wavelength of 0.86 $\mu$m at the same angular resolution. The third configuration also uses the Mie phase function but uses a reduced angular resolution of only 2 zenith discrete ordinates bins and 4 azimuthal discrete ordinate bins. The different combinations of these parameters give a total of 108 different base states.


For each base state, we numerically evaluate the Jacobian matrix. We choose the observational sampling to consist of 33 different imaging sensors described by perspective projections that image the domain simultaneously from nadir and also 32 combinations of four zenith angles [75.0°, 60.0°, 45.6°, 26.1°] and eight relative azimuthal angles [0°, ±45°, ±90°, ±135°, 180°]. Each imaging sensor has a field of view of 6° and 26 by 26 pixels and points at the centre of the domain from a distance

of 10 km. The cosine of the solar zenith angle is set to 0.3 ($\theta_{\mathrm{sun}}$) for all simulations. These observations thus capture both the diffuse radiance escaping the shadowed side of the cloud as well as the backscattering radiance from its illuminated side. We choose the state vector to be the extinction at every 2nd grid point in each dimension to reduce computational expense in the finite differencing calculations. We also exclude the grid points at the domain boundaries that parameterize the open BCs from the state vector. The state vector is therefore of length 1100 (11 by 11 by 10). It is important to remember that ill-posedness

(or instability, or ill-conditioning) in an inverse problem is dependent on the choice of representation for the retrieved quantity. Here, we tackle a fully 3D representation with equi-spaced grid points throughout the cloud. We use this setup to demonstrate the issues that arise from a generalization of remote sensing to 3D that would not arise in, for example, a retrieval of only optical depth using 3D RT (e.g., Marchand and Ackerman, 2004).

Increments to the state vector for numerical evaluation of the Jacobian matrix are set as

$\Delta a_j = sgn(a_j) \max(0.01|a_j|, 0.01).$ (66)



This decision, and other numerical considerations of the finite differencing are described and justified in Appendix E. We also describe our procedure for accelerating the finite differencing calculations, which is based on a method discussed in Evans (1998) for accelerating multi-spectral SHDOM solutions. This method can lead to an acceleration of up to ~100 times for finite differencing calculations with optically thick base states.

## 4.2 Results

For each of the base states, we evaluate the condition number (Eq. 6) of the reference Jacobian, which is shown in Fig. 5. We see increasing condition numbers with larger optical thickness, and with reduced phase function anisotropy and higher angular resolution. Larger condition numbers indicate increasing instability of the linearized inverse problem. There is little variation of condition number with single scattering albedo over this range and almost no dependence on surface albedo. The condition number ranges from well-conditioned $\kappa(\mathbf{K}) \sim 10^1$ to very ill-conditioned $\kappa(\mathbf{K}) \gg 10^5$. The maximum values reached $\kappa(\mathbf{K}) \sim 10^{16}$, which is extremely ill-posed, likely due to noise from the finite differencing. This transition towards ill-posedness is not a deficiency of the observing geometries, which can occur and have been documented elsewhere (Holodovsky et al., 2016), or resolution. The problem setting employed here has hemispherical observations, so all of the cloud is well-observed. This is confirmed by the low condition number in the optically thin setting. Instead, the cause of the larger condition numbers is traceable to the nature of the radiation transport itself, i.e., it is largely inherited from the continuous RTE problem (Bal and Jollivet, 2008; Chen et al., 2018; Zhao and Zhong, 2019). As we increase the order of scattering, we increase the spatio-angular smoothness of the radiation field and it therefore loses information about the spatial detail of the extinction field. This smoothing operation of the scattering leads to increasing ill-posedness under inversion, as quantified by the condition number.





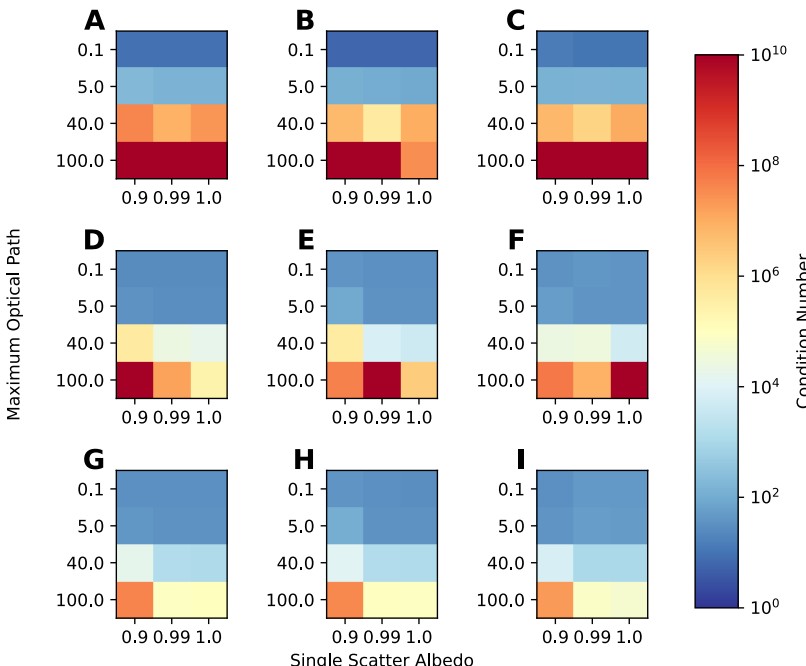

**Figure 5: The condition number (see text for details) of the finite differenced Jacobian for 3D gaussian clouds with different combinations of single scattering albedo and maximum optical path. Row A, B, C has isotropic phase functions. Row D, E, F has Mie phase functions and Row G, H, I have Mie phase functions with reduced angular resolution (see text for details). Column A, D, G has black surface. Columns B, E, H have Lambertian surface albedos of 0.2 while columns C, F, I have Lambertian surface albedos of 0.7. The crimson squares exceeding the color scale reach $10^{16}$.**

Since the condition number of the Jacobian is expected to affect the accuracy and convergence rate of the tomographic retrievals (Section 3), it is important to understand the physical and numerical principles that lead to this behaviour. We can explain the results in Fig. 5 using the forward-adjoint framework by considering both the structure of the reference RTE solutions used to calculate $\Delta f_j$ and the pseudo-forward RTE solutions, which together form the Jacobian elements. To characterize the reference and pseudo-forward RTE solutions, we will make use of the Knudsen Number (Kn), which is the ratio of the mean free path (Davis and Marshak, 2004) to the domain length scale. This quantity can be used to define the ballistic (Kn ≫ 1) and diffusion (Kn ≪ 1) regimes of transport. In a homogeneous medium, Kn is simply the reciprocal of the optical depth but tends to be larger in a heterogeneous medium as transmission becomes more efficient (Davis and Marshak, 2004). Our explanations in the following paragraphs are simply descriptions of the qualitative behaviour of scattering, supported by rigorous mathematical or numerical results where appropriate.

The Knudsen number depends on the definition of the system. In any partly cloudy system we will have Kn ~ 1, due to the dominance of ballistic trajectories from the Sun to the surface. On the other hand, within individual clouds, we may have Kn





$\ll 1$. We consider individual clouds as the system under consideration and the mean-free-path used in the definition of the Kn is the domain average rather than at any particular position (Davis and Marshak, 2004). Kn is a classification of the optical properties, not of the transport, so it is common to both the reference and the pseudo-forward RTE, and can be used to describe both. $Kn \ll 1$ does not guarantee that the diffusion approximation for radiative transport holds everywhere in the system, but rather that there is an interior region where it holds to order Kn in the appropriately non-dimensionalized RTE problem (Chen et al., 2018).

In the ballistic limit ($Kn \gg 1$), both the reference RT problem, which controls the radiance derivative source vector $\Delta \boldsymbol{f}_j$, and the pseudo-forward problems, which samples $\Delta \boldsymbol{f}_j$ to calculate the elements of the Jacobian matrix in Eq. (60), are dominated by their direct beams. The scattering is limited to low order. This means that $\Delta \boldsymbol{f}_j$ does not have a strong spatial dependence as the optical paths are small enough that the exponential nature of transmission has not yet come into play. The pseudo-forward radiance, which has a pencil-beam source, has a highly singular spatial distribution, that is largely restricted to the line of sight of each sensor pixel. Because of this, the overlapping spatial support between $\Delta \boldsymbol{f}_j$ and the pseudo-forward radiance is highly localized in space. This means that the pseudo-forward radiances corresponding to different sensor pixels are highly independent and so are the columns of the Jacobian matrix. As a result, the measurements can resolve spatial variability at the pixel scale. This leads to the low condition numbers in the optically thinnest base states in Fig. 5 as measurement resolution and the grid resolution are similar.

The pseudo-forward radiances and $\Delta \boldsymbol{f}_j$ are also highly anisotropic. The angular variation of the measurements therefore contains information about both the 3D variation of the phase matrix and also the 3D variation of the scattering coefficient. In the continuous inverse transport problem, this high degree of spatio-angular singularity in the pseudo-forward radiances in the ballistic limit enables the unique and stable inversion of both quantities simultaneously (Bal, 2009; Bal and Jollivet, 2008). The downside of the ballistic regime is that the scattering coefficient and phase matrix are not separable, and even high angular moments of the phase matrix must be retrieved to avoid errors in the retrieval of the scattering coefficient.

On the other hand, in the limit of $Kn \ll 1$, the radiation transport degenerates to a diffusion problem (Chen et al., 2018; Davis and Marshak, 2001). The radiance derivative sources $\Delta \boldsymbol{f}_j$ now span several orders of magnitude with large values optically close to the sun and much smaller values optically far from the sun. In a homogeneous medium, the behaviour in the diffuse limit is an exponential decay of the actinic flux as function of the diffusion length from the source (Davis and Marshak, 2001). This leads to a scale mismatch in the magnitude of the Jacobian elements between the illuminated and shadowed sides of a cloud, as illustrated in Fig. 6. There is a corresponding trend towards decreasing anisotropy of $\Delta \boldsymbol{f}_j$ with increasing optical distance from the sun.





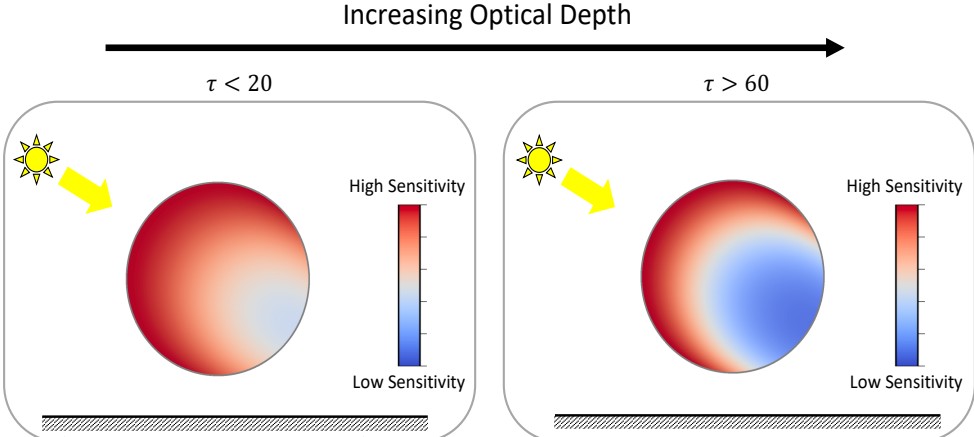

**Figure 6: A conceptual diagram illustrating the decrease in magnitude of the Jacobian elements in the solar direction through a cross section of a homogeneous spherical cloud. The mismatch in the magnitude of the Jacobian elements between the illuminated and shadowed sides increases as the optical depth increases.**

910  An example of the pseudo-forward radiance solution for one of the optically thick base states examined here is shown in Fig. 7. For the pseudo-forward radiance fields, near the pencil-beam sources, the direct beam still dominates. With increasing optical depth from the source, the diffuse pseudo-forward radiance, which has undergone many more scattering events, begins to dominate. The diffuse radiance field is angularly smoothed, through repeated convolution with the phase matrix, and spatially smoothed through the streaming of the angularly smooth source fields. In a homogeneous medium, this manifests

915  itself as an exponential decay of the angularly-averaged intensity with distance from the source, which also causes a scale mismatch in the Jacobian elements, though this time between regions optically close and optically far from the sensors. The increasing spatial smoothness of the pseudo-forward radiance with optical depth indicates that the measurements become sensitive to only increasingly wide averages of the optical properties with increasing optical distance from the sensor. This causes ill-conditioning under inversion (Chen et al., 2018; Zhao and Zhong, 2019). The spatial smoothing effect of the wide

920  pseudo-forward radiance is the three-dimensional or depth-resolved manifestation of the 2D radiative smoothing effect that has been widely studied, based on consideration of a plane-parallel, homogeneous base state (Marshak et al., 1995; Davis et al., 1997).





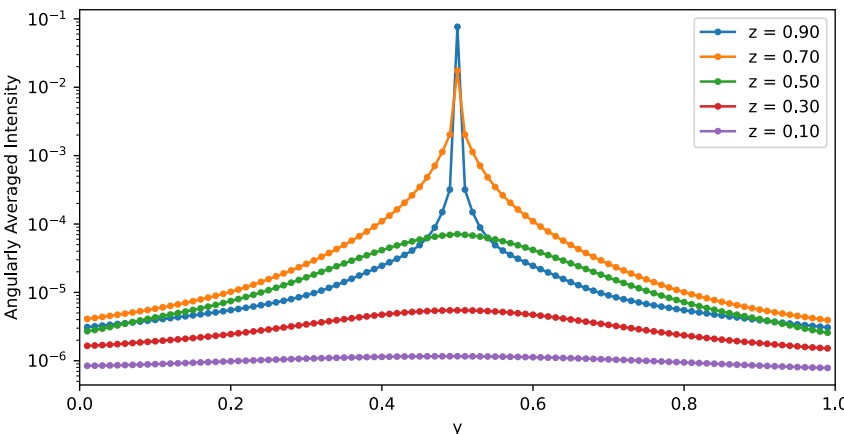

**Figure 7: Cross sections of the angularly-averaged intensity (actinic flux) for a pencil beam problem. The medium is the gaussian extinction field with the maximal optical depth of 40.0, single scattering albedo of 1.0 and the Mie phase function with full angular accuracy (see text for details). This simulation is done with SHDOM using an increased 101 points in each dimension (10 m resolution) to resolve the volumetric radiance field. The pencil beam source is located at the top of the centre of the domain (x=0.5, y=0.5, z=1.0) and pointed at nadir. Each coloured line shows a single transect of the angularly-averaged intensity at a different altitude.**

The exponential decay of the pseudo-forward radiance is a feature that is unique to the tomographic problem of retrieving three-dimensional or depth-resolved spatial variability. This feature has been recognized in other tomographic applications utilizing diffuse light (Tian et al., 2010; Niu et al., 2010). This feature means that the sensitivity of measurements to changes in optical properties is rapidly lost with increasing optical distance from the sensor. While the pseudo-forward radiances corresponding to adjacent sensor pixels remain quite independent in the region optically close to the sensors where the radiance fields are localized, they become indistinguishable in the region optically far from the sensors due to smoothing and their decay to zero. The condition number measures this worst-case loss of independence (Chen et al., 2018), which occurs in the region which is optically far from the sensors and also from the Sun (through $\Delta f_j$). As the optical dimension of the medium increases, this worst-case loss of independence is exacerbated, causing the rapid growth of the condition number (Figure 5). The exponential decay to zero of optical depth sensitivity causes a scale mismatch in the Jacobian elements between regions optically close and optically far from the sensors, similar to that which occurs in the forward problem, as illustrated in Fig. 8.





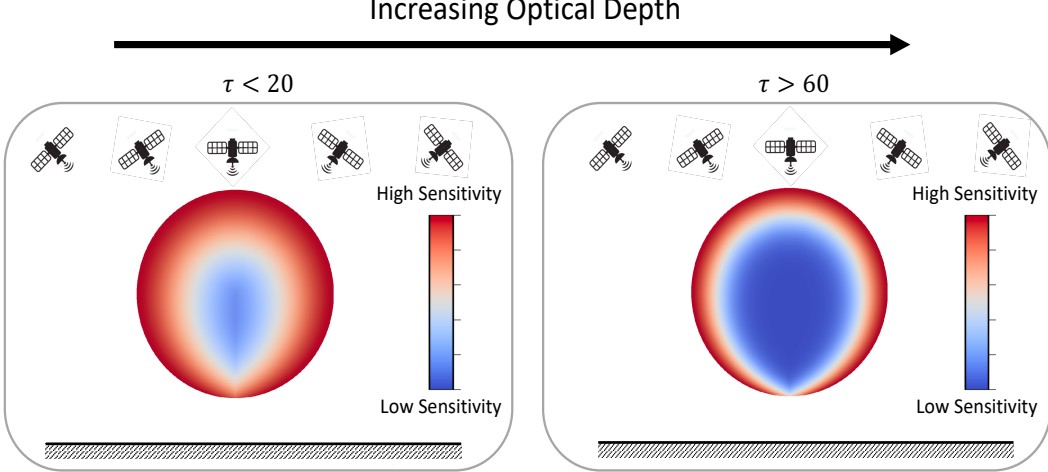

**Figure 8: A conceptual diagram illustrating the decrease in magnitude of the Jacobian elements with distance from the sensors through a transect through a homogeneous spherical cloud. The mismatch in the magnitude of the Jacobian elements between the outer edges and the interior of the cloud increases as the optical dimension of the cloud increases.**

The two scale mismatches related to the sensors and the Sun that develop in the Jacobian matrix as the medium becomes optically thick are a significant source of instability in the Jacobian as measurements will be orders of magnitude more sensitive to changes in optical properties in regions close to the sun and sensors than those further away. This can be seen quantitatively by binning the absolute magnitude of elements of the reference Jacobian matrices by the solar "delta-M" transmission to each grid point and the delta-M transmission of the minimum optical path from all sensors to each grid point (Figure 9). This latter

quantity is referred to as the minimum sensor transmission. The delta-M transmission is derived from the path integral of the delta-M scaled extinction (Wiscombe, 1977), where the truncation fraction is set by the phase function and the angular resolution of the SHDOM solver. This latter metric is a measure of overall optical distance from the sensors to each point in the cloud.

Optical path has also been used similarly to define a region of the cloud in which measurements are no longer sensitive to rearrangements of the small-scale features of the extinction field, known as the "veiled core" of the cloud (Forster et al., 2020). Forster et al. used a threshold of at least $\tau > 5$ along the line-of-sight of all sensors intersecting each volume element to define the volumetric extend of the veiled core of the cloud. This optical depth threshold is roughly equivalent to Delta-M scaled minimum sensor transmissions less than 0.1 for the simulations employed here using the Mie phase function and full angular

accuracy. Given that the set of sensors covers the upper hemisphere, the minimum sensor transmission is loosely equivalent to minimum transmission to the edge of the domain. Figure 9 shows the increasing scale mismatch in derivative magnitudes between the optical interior and exterior as the total optical depth of the medium increases and the Knudsen Number decreases. Figure 9's panels D, H, and L show that derivatives of radiances with respect to changes of extinction within the veiled core can be almost two orders of magnitude smaller than those with respect to extinction changes in the exterior.




Note that it is the mismatch of scales between sensitivity to the interior and exterior that is important here, not the absolute smallness of the pseudo-forward radiance in the veiled core of the cloud (up to numerical precision). If the state vector were restricted to describing a region that only included similar transmissions from all sensors, then the problem of ill-conditioning would be much reduced. Such a partitioning is not generally available. The interior regions must be optically thin enough that

the non-linearity of the transmission causing the scale mismatch is small. In an optically thick cloud, this would require extremely detailed knowledge about the extinction field in the edge regions of the cloud, which is not generally available. For moderately opaque clouds, lidar may provide valuable information to constrain the outer portions of the cloud and thereby mitigate this issue.

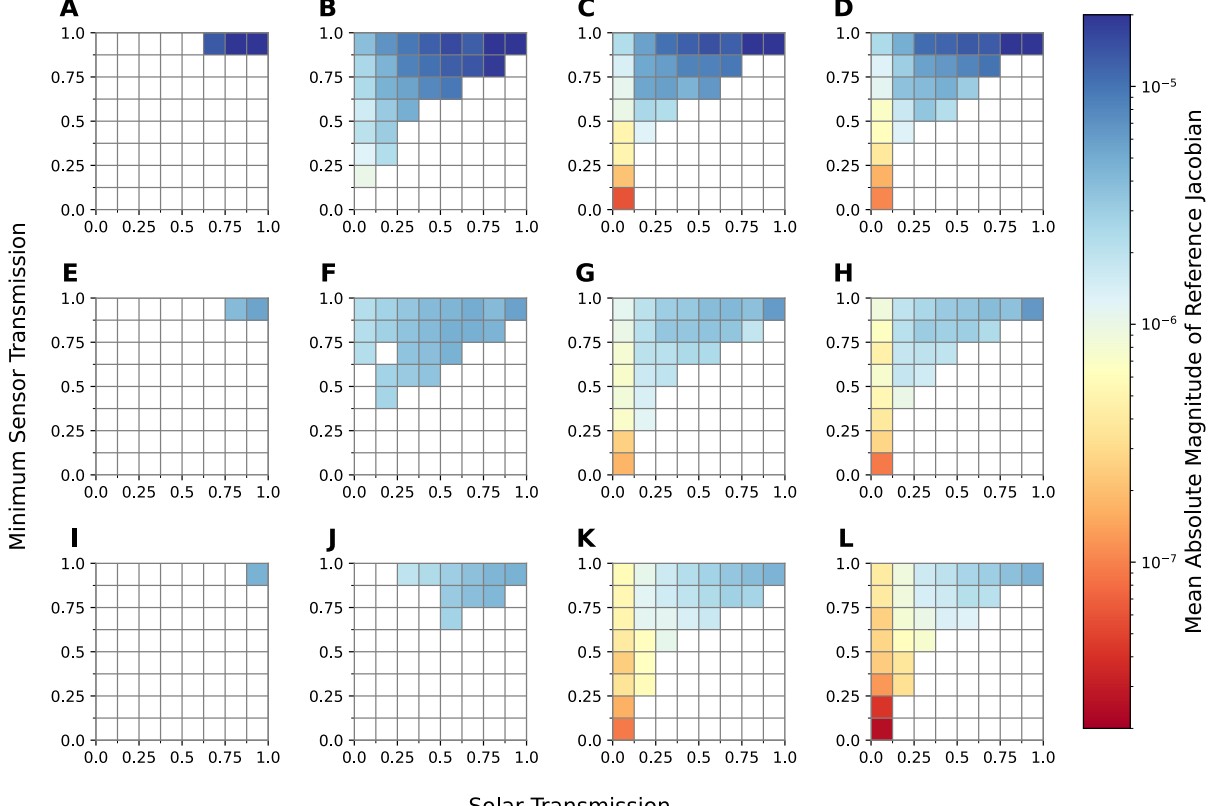

**Figure 9: The mean absolute magnitude of finite difference Jacobian for 3D gaussian clouds in each bin of solar and minimum sensor delta-M transmission. Each column corresponds to a base state with maximum cloud optical depth from 0.1, 5.0, 40.0, 100.0, increasing from left to right. Each row corresponds to a different combination of phase function and angular accuracy with isotropic phase function, Mie with full angular accuracy, and Mie with reduced angular accuracy in descending order. All base states have a black surface with conservative scattering.**


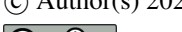



The arguments presented so far explain the dependence of the difficulty of the inverse problem (measured by condition number of the Jacobian matrix) on the extinction field. They also explain the sensitivity of the condition number to low order single scattering properties such as the asymmetry parameter and single scattering albedo, which modulates the diffusion lengths and transport mean free paths that control the spatial smoothing and exponential decay in the diffuse limit (Davis et al., 2021)They

do not, however, explain the strong sensitivity of the condition number or spatial structure of the Jacobian matrix to the angular resolution of the SHDOM solver present in Figure 5 and Figure 9, respectively. We now consider this behaviour.

In the diffusion limit, Davis et al. (2021) investigate the role of the asymmetry parameter on controlling the spatial smoothing and exponential decay of the forward and pseudo-forward problems using random walk theory. We can see from our results showing the sensitivity of the Jacobian to angular resolution in Fig. 5 and Fig. 9 that, in addition to what is discussed in Davis

et al. (2021), there is also a role for the higher order moments of the phase function beyond the asymmetry parameter, and that these interact with the spatial moments of the pseudo-forward radiance. In particular, we know that in a medium with a large forward scattering peak, more radiance will stay angularly close to the direct beam after several scattering events even if the average direction of propagation is lost rapidly due to backscattering. We can hypothesize that even if the asymmetry parameter remains unchanged, a larger forward peak will skew the pseudo-forward radiance near to the direct beam, increasing its

localization property and thereby reducing ill-conditioning.

This hypothesis is borne out in the approximate numerical framework of SHDOM, where the forward scattering peak of a phase function is treated by the delta-M approximation. The larger the forward scattering peak and the lower the angular resolution of the SHDOM solver, the more of the phase function is angularly unresolved by the model and is lumped together

into the direct transmission. As such, a larger forward scattering peak or lower angular resolution act to transform the medium to one with a higher effective Knudsen Number, and thereby improving its conditioning. For the Mie phase function considered here, the use of low angular accuracy causes an almost halving of the extinction compared to the high angular accuracy, which is itself also roughly a halving of the true extinction. The change in the condition number with a change in angular resolution

of the SHDOM solver is substantial, and indicates that the stability of the inverse problem depends on the discretization of the system.

### 4.3 Discussion

### 4.3.1 Physical Implications

In Section 4, we have documented the behaviour of a linearized tomography problem. A number of these results have general

implications that are not specific to the SHDOM model or the use of the approximate Jacobian described here. We now take the time to consider the wider implications of these results. We have shown that the condition number of the inverse problem largely depends on the Knudsen Number or optical size of the medium, as supported by theory. We should therefore expect





the convergence rate of an iterative retrieval to decrease in optically thicker media, as discussed in Section 3. As such, the tomographic retrieval of optically thicker media is expected to be computationally more expensive due to both RTE solutions becoming more expensive in optically thicker media and the need for more iterations of optimization to achieve a user-specified level of accuracy. High levels of retrieval accuracy may not be obtainable in optically thick media due to the extreme ill-conditioning, possibly causing slower convergence than the stopping condition of an optimization procedure while still far from the optimal solution.

We have also shown the existence of substantial spatial variability of the linear sensitivity of radiances to changes in the extinction field in optically thick clouds. In particular, linear sensitivity decreases exponentially with optical depth from the sun and from the sensors, likely causing slow convergence of the extinction field in these regions. This feature was also noted in Levis et al., (2015). We have shown here that it is, at least in part, a result of a physical limitation not just the approximations used within their paper. The fastest way to decrease the misfit with the measurements will be to change the extinction field optically close to the sensors and also the sun. If iterative retrievals of optically thick clouds are unconstrained apart from measurements, then the retrieved extinction field may approach a local minimum with little change from the initialization in the cloud centre and on the shadowed side of the cloud.  This behaviour may introduce a solar zenith angle dependence to the retrieval error, despite the use of 3D radiative transfer.

The issues apparent in optically thick clouds appear to substantially limit the applicability of the method, but we must bear in mind that, in terms of both number and area, a large portion of trade cumulus cover comes from small clouds (Zhao and Di Girolamo, 2007). Many trade cumulus tend to be smaller than 800 m in geometric depth (Chazette et al., 2020; Guillaume et al., 2018) and average adiabatic fractions for these clouds can be significantly less than unity (Eytan et al., 2022). Most of these clouds will have maximum optical depths less than 40, which suggests that prior information or regularization will not be essential for ensuring high fidelity retrievals of these clouds. We have not so far considered the feasibility of tomography in other, more stratiform, cloud types. We explore the sensitivity to such differences in Section 6, using the more computationally efficient approximate Jacobian calculation.

### 4.3.2 Numerical Implications

We have also identified an important sensitivity of the ill-conditioning of the retrieval to the numerical discretization of the method. Of course, ill-conditioning is always sensitivity to discretization choices. For example, if we were to only retrieve a single unknown per column with parameterized vertical variability, then the condition number of the corresponding Jacobian matrices in the optically thickest cases considered here would be of order 10, the same as the optically thinnest tomography problems. This distinction is not unique to our study.



The important behaviour that we have documented here for the first time is the importance of the angular discretization of the forward model for determining the conditioning of the model, rather than standard properties such as the number and spacing of measurements or the spatial resolution. The sensitivity of ill-conditioning to angular discretization arises from the presence of strongly peaked phase functions and the use of the delta-M approximation, which reduces the effective optical depth of the medium to account for strong forward scattering that is unresolved by the angular discretization.


The large decrease in condition number observed when decreasing angular accuracy suggests that using low angular resolution may be beneficial in inversions. The trade-off for forming a better-conditioned inverse problem in this way is that the forward model is a poorer approximation for reality and may in fact have significant biases (Evans, 1998). Even if a benefit in convergence rate is not apparent, these results indicate that retrieval results should not be expected to generalize to other

angular resolutions even when the angular resolution is high enough that the forward model has converged in accuracy. In particular, inversions may actually decrease in fidelity as angular resolution is increased. Additionally, results should not be expected to generalize between phase functions with substantially different forward scattering peaks such as Mie vs Henyey-Greenstein phase functions, even if they have the same asymmetry parameter.

The conditioning results presented here are likely only representative of other explicit RTE solvers like SHDOM utilizing the delta-M approximation. It is unclear how the results will generalize to other widely used methods of solving the RTE, such as Monte Carlo. Those state-of-the-art Monte Carlo solvers in atmospheric radiative transfer that do use the delta-M or phase function truncation approximations may have some similar dependence of their conditioning on the truncation fraction as documented here for SHDOM. However, modern implementations of phase function truncation tend to be scattering-order

dependent (Wang et al., 2017) to avoid strong bias and so may not express the behaviour demonstrated by SHDOM.

**5 Validating the Approximate Jacobian calculation**

With the theory introduced in the previous section in place, we can examine the consequences of the approximation to the Jacobian (Eq. 58) used in AT3D, and how that approximation exhibits itself as quantitative errors. We examine how well the approximate Jacobian reproduces the behaviour described in Section 4. As described in Section 3.3, we are approximating the

pseudo-forward radiance by its direct beam (Eq. 61). In this case, we can see that the entire diffuse pseudo-forward radiance profile (Fig. 7) will be neglected. This means that the approximate Jacobian does not represent the non-local sensitivity of measurements to changes in optical properties outside of their field of view other than through changes to the direct solar transmission. This may seem a rather extreme approximation, but we must bear in mind that the most stable information to extract is contained in the highly localized direct beam component (Bal and Jollivet, 2008). The success of the approximation

requires that the pixel-to-pixel smoothing only becomes important and significant when the medium is optically thick enough that there will be significant decay in the sensitivity with distance from the sensor. In this case, the loss of sensitivity to the



cloud interior from all pixels (in the linearized setting) is much more important than the neglect of the pixel-to-pixel smoothing. In this section, we test the extent to which this is true quantitatively.

From the theory developed in the previous section, we also expect a sensitivity of the approximate Jacobian to the dimensionality of the RT problem. In 3D, the pseudo-forward radiance field is highly singular in space (Figure 7) and anisotropic. On the other hand, as the dimensionality of the transport decreases, the pseudo-forward solution will become increasingly less singular. For example, in 1D, the pseudo-forward source is a plane illumination, just like the sun (Hasekamp and Landgraf, 2005). The symmetry of the source means that the pseudo-forward radiance does not disperse perpendicular to the collimated source but, instead, it substantially modifies the depth dependence of the pseudo-forward radiance field and reduces its anisotropy. This means that the direct beam approximation will perform worse and the performance of the approximation to the Jacobian calculation adopted here cannot be expected to generalize from 3D to 2D or, especially, 1D problems. This has an important implication for our extension of the approximate Jacobian to the linearization of the system of RTEs described in Section 3 to model a 3D domain embedded in a plane-parallel atmosphere. As such, we separate our

error analysis between the elements of the approximate Jacobian corresponding to the primary 3D domain $D_1$ (Figure 2) and the other elements controlling the boundary radiances through the auxiliary RTE problems.

Note that, in the following analysis, we quantitatively validate derivatives of radiances only with respect to volume extinction coefficient at different grid points rather than other optical properties such as the single scattering albedo or components of the

phase matrix. The approximation to the Jacobian only approximates the pseudo-forward radiance and therefore is common to derivatives of radiances with respect to all optical properties. However, each optical property interacts differently with the pseudo-forward radiance field. For example, the single scattering albedo interacts with all spherical harmonics of the radiance field, while the extinction coefficient interacts with all except the isotropic component. As such, errors in radiance derivatives with respect to single scattering albedo may be disproportionately affected by error in the pseudo-forward radiance that occur

optically far from the sensors compared to derivatives with respect to extinction. In contrast, the higher-order expansion coefficients of the phase matrix are only sensitive to high-order spherical harmonics of the radiance field and will therefore be disproportionately affected by errors in the pseudo-forward radiance that occur optically close to sources. As a result, Jacobian errors may be distributed slightly differently in space and observation angle as the angular structure of $\Delta \boldsymbol{f}_j$ changes with the nature of the optical (or microphysical) unknown. We focus solely on extinction to illustrate how errors in the approximate

pseudo-forward radiance propagate to the Jacobian as the retrieval of extinction is fundamental for retrievals of spatial structure.

To quantify the agreement between the finite differenced Jacobians defined in Section 4.1 and the equivalent approximate Jacobians ($\tilde{\mathbf{K}}$) we use the relative Frobenius error, which is an element-wise relative root-mean-square error (RMSE):



Relative Frobenius Error $= \frac{\|\widetilde{\mathbf{K}} - \mathbf{K}\|_F}{\|\mathbf{K}\|_F} = \frac{\sqrt{\Sigma_{ij}(\widetilde{\mathbf{K}}_{ij} - \mathbf{K}_{ij})^2}}{\sqrt{\Sigma_{ij}(\mathbf{K}_{ij})^2}}$                  (67)

Our validation approach of comparing the approximate Jacobian to the finite-differenced Jacobians measures the consistency of the forward model with the approximate Jacobian. The consistency between these two is what is important for the robustness of the local optimization method used in the tomographic retrieval implemented in AT3D. As such, the Relative Frobenius Error includes the effects of computational noise in the forward model, that will emerge in the finite-differenced derivatives.

When evaluating derivatives calculated using a forward-adjoint principle, the Relative Frobenius Error was around 0.04, reflecting uncertainties in the finite differencing and inconsistencies between the forward and adjoint models (Doicu et al., 2020). We expect a similar degree of accuracy in the optically thin limit for the approximate Jacobian. We performed extensive verification of the approximate Jacobian. We identified a number of instances where the computational noise in the finite-differenced derivatives is actually the dominant source of error, based on comparison against an analytic solution (Appendix

D). For that reason, it is important to keep in mind that this error metric is a measure of the consistency of the approximate Jacobian with the forward model, and that the stability of the forward model itself is also contributing.

### 5.1 Primary Domain

For the primary domain, we are analysing the accuracy of the approximate Jacobian across the same base states as in Section 4. We also define the state vector the same way, so we are not analysing derivatives with respect to the open BCs. We can see

in Fig. 10 that the error rapidly grows beyond a benchmark value of 0.02 as the clouds become optically thicker, scattering is closer to conservative, and surface albedos become larger. The small change in the error between a black surface and a Lambertian surface with an albedo of 0.2, in Fig. 10 show that there is little sensitivity of the error to the surface albedo when it is not too reflective. This indicates good suitability of the approximate Jacobian for media over oceanic or other dark surfaces. We also see greater agreement for more forward scattering media, especially at lower angular resolution. This is because the

approximate Jacobian treats the direct beam exactly, hence the more energy within the direct beam due to delta-M scaling, the more accurate the approximation. A lower angular accuracy therefore gives the benefit of more consistent derivatives with the forward model, with the downside that the forward model will have larger errors against reality. To better understand these systematic differences between the approximate and reference Jacobian matrices, we also calculate how the errors are distributed in space and angle, i.e., in state and measurement space.





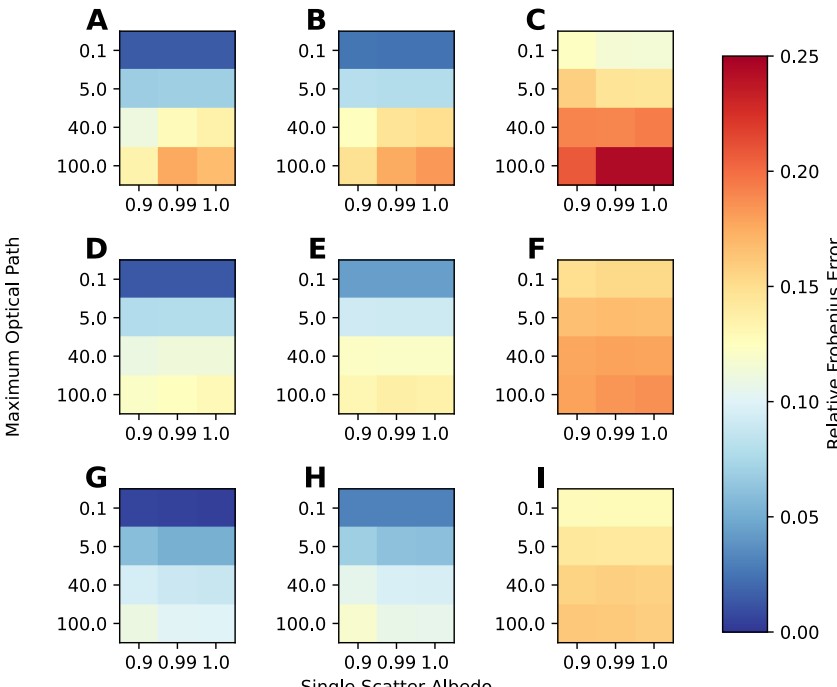


**Figure 10: The relative Frobenius error (Eq. 67) of the approximate Jacobian with respect to the finite-differenced Jacobian for the same 3D Gaussian clouds as used in Fig. 5. Row A, B, C has isotropic phase functions. Row D, E, F has Mie phase functions and Row G, H, I have Mie phase functions with reduced angular resolution (see text for details). Column A, D, G has black surface. Columns B, E, H have Lambertian surface albedos of 0.2 while columns C, F, I have Lambertian surface albedos of 0.7.5.1.3 Spatial Variation of Error**


We again categorize Jacobian elements according to the delta-M transmission from the Sun to each grid-point and the minimum delta-M transmission from the sensors to each grid-point. For Jacobian elements in each bin, we calculate the RMSE and normalize it by the root-mean-square magnitude of the entire reference Jacobian matrix calculated by finite-differencing. This indicates which grid points produce approximate Jacobian elements with errors large enough to significantly change the overall

direction of the gradient. Figure 11 shows that these errors are largest in the regions at the exterior of the cloud close to both the sensor and Sun. These Jacobian entries are also the largest in magnitude and have the largest higher order derivatives, due to the curvature of the transmission function, so are also expected to have the largest errors in the finite-differencing.

We also show the ratio of the mean absolute magnitude of the approximate Jacobian to the mean absolute magnitude of the

reference Jacobian in each bin (see Fig. 12). We see that the typical magnitude of the approximate Jacobian decays much quicker with optical depth from the Sun and sensors than the reference. This means that the approximate Jacobian has an enhanced scale mismatch in sensitivity to the properties of the cloud at the exterior and interior, thus exacerbating the ill-conditioning problem outlined in Section 4.2. Physically, this is due to the faster exponential decay of direct transmission than





diffuse radiance. We could then hypothesize that the condition number of the approximate Jacobian would be higher than that

of the reference in the optically thickest cases. This is not borne out in Fig. 13, which shows that condition numbers for the

approximate Jacobian is not appreciably larger than for the reference Jacobian (cf., Fig. 5), and is actually smaller in the

optically thick limit. Thus, the condition numbers in Fig. 5 for the reference Jacobian are likely larger in the optically thick

limit due to numerical noise in the derivative calculations. The patterns of error illustrated in Figures 11 & 12 are similar for

the non-zero surface albedos and non-conservative scattering except with overall larger errors (not shown).

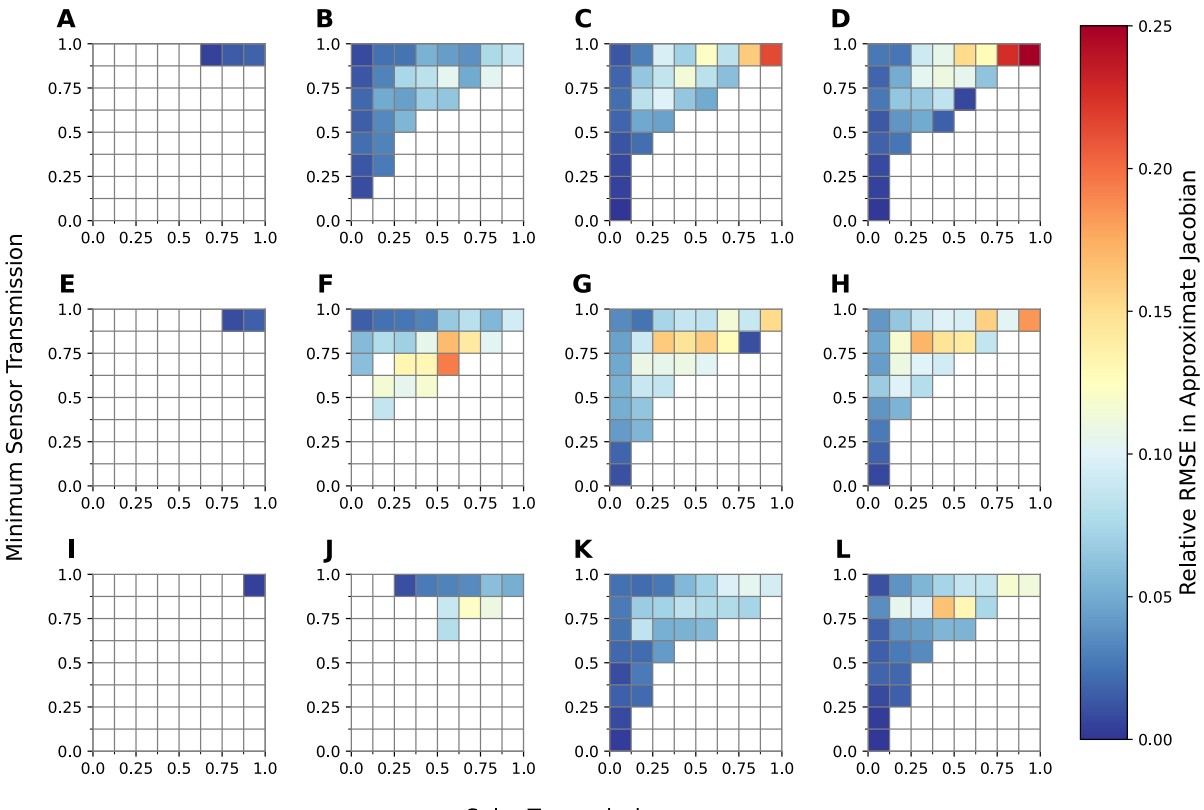


**Figure 11: The relative RMSE error in the approximate Jacobian for 3D gaussian clouds in each bin of solar and minimum sensor delta-M transmission (See Fig. 9 and associated text for definitions). The normalization of the RMSE in each bin is the RMS magnitude of the entire Jacobian matrix, cf. Eq. (67). As in Fig. 9, each column corresponds to a base state with maximum cloud optical depth from 0.1, 5.0, 40.0, 100.0, increasing from left to right. Each row corresponds to a base state with different phase**
**function and angular resolution, with isotropic phase function (A, B, C, D), Mie phase function with high angular resolution (E, F, G, H), and Mie phase function with low angular resolution (I, J, K, L).**

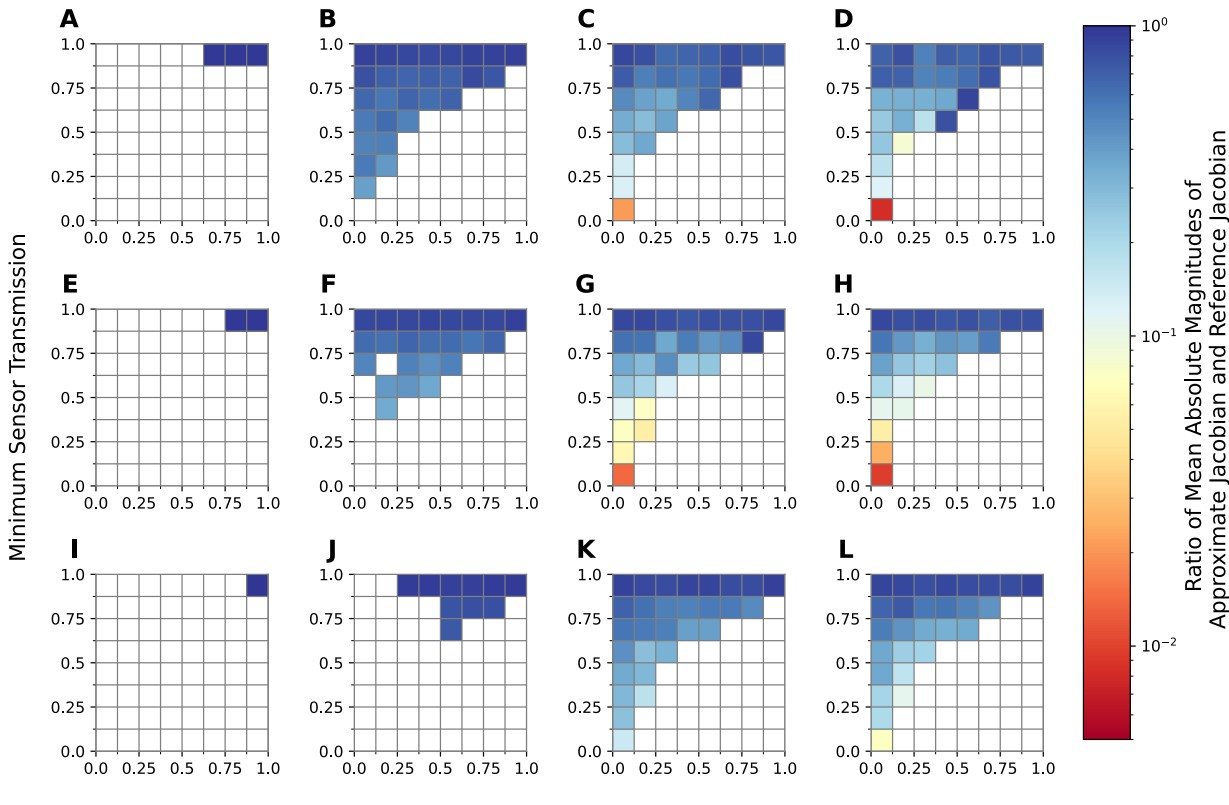

**Figure 12:** The ratio of the mean absolute magnitude of the approximate Jacobian to the mean absolute magnitude of the finite
difference Jacobian in each bin of solar and minimum sensor delta-M transmission for 3D gaussian clouds with conservative
scattering over a black surface. As in Figs. 9 and 11, As in Fig. 9, each column corresponds to a base state with maximum cloud
optical depth from 0.1, 5.0, 40.0, 100.0, increasing from left to right. Each row corresponds to a base state with different phase
function and angular resolution, with isotropic phase function (A, B, C, D), Mie phase function with high angular resolution (E, F,
G, H), and Mie phase function with low angular resolution (I, J, K, L).





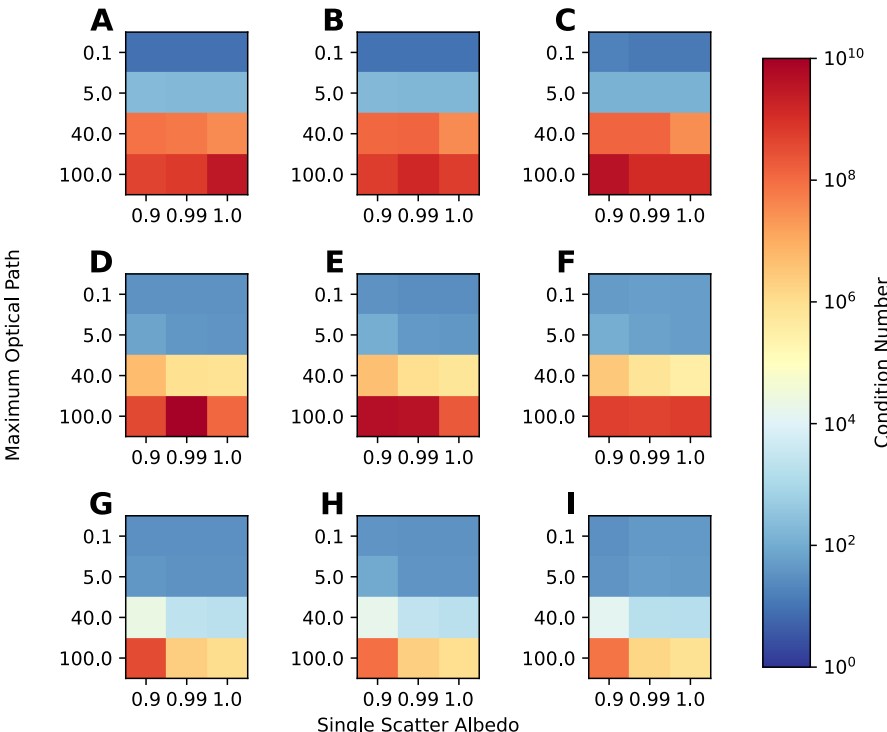

**Figure 13: The condition number (see text for details) of the approximate Jacobian for 3D gaussian clouds with different combinations of single scattering albedo and maximum optical path. Same as Fig. 5, but for the approximate Jacobian. Row A, B, C has isotropic phase functions. Row D, E, F has Mie phase functions and Row G, H, I have Mie phase functions with reduced angular resolution (see text for details). Column A, D, G has black surface. Columns B, E, H have Lambertian surface albedos of 0.2 while columns C, F, I have Lambertian surface albedos of 0.7.**

### 5.1.4 Angular variation of error

We also examined the sensitivity of the Jacobian errors to the set of scattering angles in the observations. In Fig. 14, we show the relative RMSE in the Jacobian elements corresponding to each viewing angle, grouped by scattering angle. The shape of the error depends on the phase function. In general, the larger errors occur in the backscattering directions, for the isotropic phase functions (Fig. 14 A, B, C, D). These observation geometries also include large sensitivity to grid-points that are optically close to the Sun, and therefore have the largest truncation errors, consistent with Fig. 11. These truncation errors may account for a substantial portion of the scattering angle dependence. When a Mie phase function is used (Fig. 14 I, J, K, L), the error is also angularly dependent with a minimum around the rainbow direction and a maximum at scattering angles of around 100°.

We performed further investigation of these angular patterns in the error using much simpler clouds with hyper angular observations. We defined cloud base states that consisted of just a single, optically thin, cloudy grid point and observations with 1° zenith angle spacing both along and across the solar plane. We confirmed that the dip in error observed around the





rainbow scattering angles in Fig. 14 is due to the Mie phase function as this feature does not appear when a featureless Henyey-Greenstein phase function is used with the same asymmetry parameter. There is further angle-dependence in the error that is not correlated with scattering angle, which we can attribute to multiple-scattering effects. We do not expect a consistent angular dependence for these features. For example, a peak in error at scattering angles of around 100° appears for our single grid-point clouds regardless of the choice of a Mie or Henyey-Greenstein phase function. Changing the solar zenith angle for our simplified single grid-point clouds revealed that this error is actually dependent on zenith, peaking near nadir. We should expect the angular dependence of the error to vary with the base state cloud structure and cannot easily extrapolate from our simplified cloud to the more complex base states quantified in Fig. 14. However, we have identified that there are angular features in the error that depend only on the phase function due to the increasing relevance of single-scattered radiation in the rainbow scattering angles. Further work with a wider diversity of cloud base states would be needed to reliably isolate multiple scattering signals in the error that are independent of phase function details.

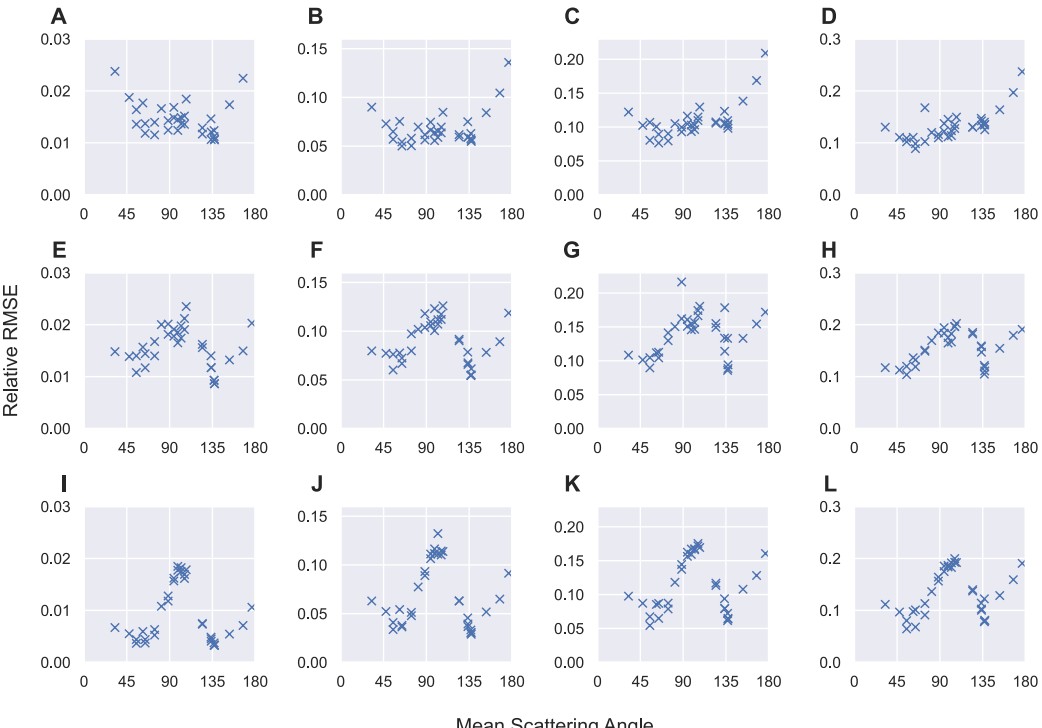

**Figure 14: The dependence of the relative RMSE in the approximate Jacobian for each image on the mean scattering angle of each image for cloud base states with conservative scattering and a black surface. As in Figs. 9, 11 and 12, each row corresponds to a base state with different phase function and angular resolution, with isotropic phase function (A, B, C, D), Mie phase function with high angular resolution (E, F, G, H), and Mie phase function with low angular resolution (I, J, K, L). Columns correspond to base states with maximum optical depths of 0.1 (A, E, I), 5 (B, F, J), 40 (C, G, K) and 100 (D, H, L).**



## 5.2 Auxiliary Domains

Here, we assess the accuracy of derivatives of radiances with respect to changes in the extinction fields in the auxiliary RTE domains that control the open horizontal BCs. We use the same set of base states and observations as in Sections 4.1 and 5.1. However, we focus only on the appropriateness of the approximate Jacobian for open horizontal BCs. We compute derivatives

with respect to extinction along the horizontal boundaries for every 4$^{th}$ point in the vertical and every 5$^{th}$ point in the horizontal. This set of base states have a low optical depth of 0.1 in the auxiliary domains and are therefore analogous to the case of an optically thin embedding medium, such as the cloud-free atmosphere.

These errors in the approximate Jacobian are displayed in Fig. 15 and show much larger overall errors than for the internal

extinction derivatives shown in Fig. 9 (note the difference in colour scale). The larger errors in the optically thin cases, when compared to Fig. 10, are driven by the poorer applicability of the Jacobian approximation to the 2D and 1D RT in the auxiliary domains. The much weaker dependence of the errors on the optical thickness of the primary domain of 3D RT than in Fig. 10 indicates that the neglect of multiple scattering within the primary domain is much less important than the errors due to the application of the Jacobian approximation to the lower-dimensional auxiliary domains. There is a much greater sensitivity to

the phase function and angular accuracy, with nearly a halving of the error when moving from isotropic phase function to Mie phase function, to the 2-stream Mie phase function.





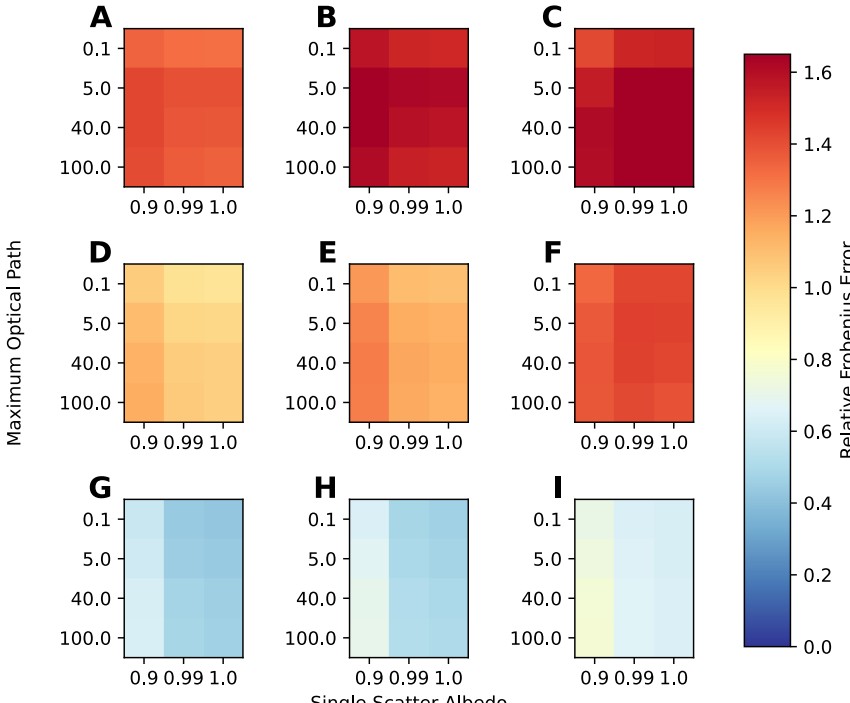

**Figure 15: Similar to Fig. 10, but for derivatives of radiances with respect to the extinction within the auxiliary domains that parameterize the open horizontal BCs. Note the difference in colour scale from Fig. 10. Row A, B, C has isotropic phase functions. Row D, E, F has Mie phase functions and Row G, H, I have Mie phase functions with reduced angular resolution (see text for details). Column A, D, G has black surface. Columns B, E, H have Lambertian surface albedos of 0.2 while columns C, F, I have Lambertian surface albedos of 0.7.**

We also examine another set of base states which are just plane-parallel cloud layers (Fig. 16). The domain and discretization for 3D RT is the same as in Section 4.1 and Section 5.1, and the horizontally infinite portion is modelled using the open horizontal BCs rather than periodic assumptions. The extinction is distributed homogeneously within the $(1 \text{ km})^3$ domain. This set of base states are analogous to stratiform clouds. We calculate derivatives only with respect to the domain boundary extinction to examine the appropriateness of the approximate Jacobian for these situations. We vary the optical depth of the plane-parallel layers across the same values used previously for the maximum optical depth in the 3D gaussian extinction fields. The errors in the approximate Jacobian increase much faster with optical depth than for the 3D Gaussian extinction





fields (Fig. 10) and are much larger. This indicates that the approximate Jacobian has no skill for BC optimization in conditions

of stratiform cloud.

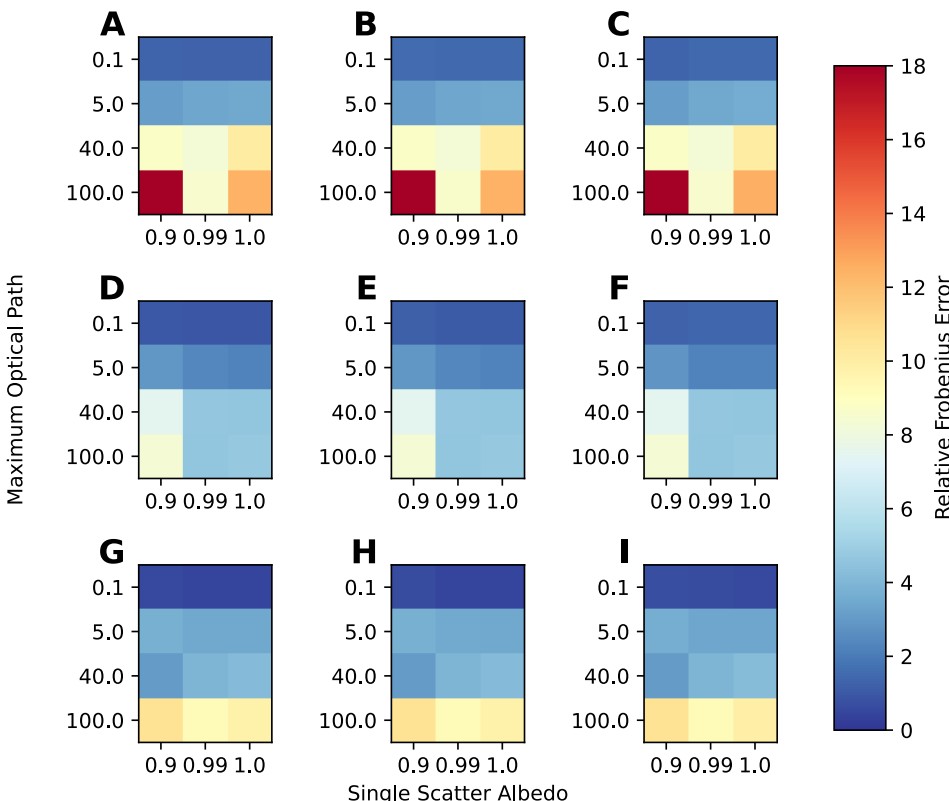

**Figure 16: Similar to Fig. 15, but using base states that are plane-parallel, horizontally homogeneous extinction fields. Note the difference in colour scale from Figs. 10 and 15. Row A, B, C has isotropic phase functions. Row D, E, F has Mie phase functions and**
**Row G, H, I have Mie phase functions with reduced angular resolution (see text for details). Column A, D, G has black surface. Columns B, E, H have Lambertian surface albedos of 0.2 while columns C, F, I have Lambertian surface albedos of 0.7.**

**5.3 Discussion**

Here, we discuss the implications of the errors in the approximate Jacobian for the iterative retrieval. The relative Jacobian

errors documented above bound the relative errors that can occur in the calculation of cost function gradients, which

additionally depend on the structure of the measurement residuals. The theoretical examination of the consequences of gradient

errors on the L-BFGS-B method have been only recently investigated (Shi et al., 2021), due to the interest in developing

stochastic variants for deep learning and other similar applications. The L-BFGS method uses finite-differencing to

approximate the Hessian. When there is noise in the gradients, the approximate Hessian can become corrupted. With bad

curvature information, typically only very small step sizes will be valid, or there may be a complete failure to select a valid

search direction. In this sense, both the inherent ill-conditioning of the inverse problem and the approximate Jacobian errors



should have a similar deleterious effect on retrieval performance. Moreover, it will not be possible to disentangle these two effects in nonlinear retrievals.

The typical solution for gradients with errors is to perform a step lengthening procedure (Shi et al., 2021). However, in systems
that are highly ill-conditioned, even without noise, such as the optically thick clouds discussed here, a step lengthening procedure may cause significant difficulty in the selection of an update vector satisfying the stabilizing Wolfe-Armijo line search conditions. We note that the errors in the gradients induced by the approximate Jacobian are deterministic, not stochastic noise, so there is an opportunity for them to be highly correlated from one base state to the next. If this occurred it would result in a certain amount of cancellation of the errors and better approximation of the Hessian of the cost function through the
L-BFGS method. We have not examined this quantitatively here due the computational expense of numerically calculating $2^{nd}$-order derivatives, but this should be kept in mind when considering differences in performance of the approximate Jacobian for tomographic retrievals when used with $1^{st}$-order optimization methods (e.g., gradient descent) vs. quasi-Newton methods such as L-BFGS-B.

For the auxiliary domains, overall errors are much larger, reaching relative Frobenius errors far in excess of 100% even for optically thin atmospheres (with isotropic phase functions). Errors are relatively independent of the scattering regime of the internal medium, but are very sensitive to the optical depth in the auxiliary domains. These results indicate that while the approximate Jacobian proposed in Levis et al. (2020) is appropriate for 3D media, it is much less so for lower-dimensional transport. This indicates that a retrieval of stratiform cloud properties using the approximate Jacobian and open horizontal BCs
is ill-advised. The ability to optimize BCs using AT3D may still be useful for retrieving a best-fitting cloud-free atmosphere jointly with the retrieval of a 3D cloud field, as the cloud-free atmosphere is optically thin.

## 6 The importance of heterogeneity for tomography

We can make use of the computational efficiency of the approximate Jacobian to explore the dependence of the condition number of the Jacobian on the spatial structure and optical thickness of the cloud field in more detail than in Figs. 5 and 13. In
particular, we illustrate the critical importance of the spatial structure of the extinction field for determining the feasibility of tomography. We contrast the behaviour of plane-parallel homogeneous clouds and 3D gaussian extinction fields, which are restricted to a $(1 \text{ km})^3$ domain under inversion. Note that the plane-parallel homogeneous cloud is still resolved in 3D and the radiation transport is in 3D. We are examining our ability to retrieve inhomogeneities embedded within the cloud. As such, the dimensionality effects described in Section 5 are not at play here. We use the same hemispheric set of observations and the
same grid geometry used in Sections 4.1 and 5.1. The clouds are conservatively scattering and use the Mie phase function with a black surface. We calculate the derivatives with respect to all internal grid points in each horizontal grid-point but only each





second grid point in the vertical. This is done to avoid the need to adopt special techniques for the Singular Value Decomposition (SVD) of large matrices to calculate the condition number.

We classify the plane-parallel clouds by their vertical optical depth and the 3D gaussian extinction fields by their vertical optical depth to the centre of the cloud (half their diameter). We refer to this as the "Optical Dimension". This classification puts the two types of extinction field on the same footing in terms of the minimum transmission from any sensor to the grid-point that is optically furthest from all sensors. They are therefore equivalent in terms of the presence of a "veiled core," as defined by Forster et al. (2020) and investigated in Section 4.2.

**6.1 Results**

In Fig. 17, we can clearly see the exponential growth of the condition number of the approximate Jacobian at larger optical dimensions consistent with theory, indicating exponential growth with inverse Knudsen Number (Zhao and Zhong, 2019). The plane-parallel clouds are notable in that the condition number increases at a faster rate. This shows how much larger the Knudsen Number (and hence mean-free-path) is in a heterogeneous cloud like the 3D gaussian extinction field and how much

more information about spatial detail is preserved, for a similar optical thickness. The importance of finite cloud edges for sensing cloud vertical structure from multi-angle radiances has also been demonstrated in nonlinear retrievals in a 2D setting (Martin and Hasekamp, 2018). The purely geometric part of this effect is partially reflected in the use of the optical radius, rather than diameter of the 3D gaussian extinction field, when comparing to the plane-parallel clouds. This reflects the ease with which oblique sensors can constrain the cloud base and edges when clouds have aspect ratios around 1, even when they

have large vertical optical depths.





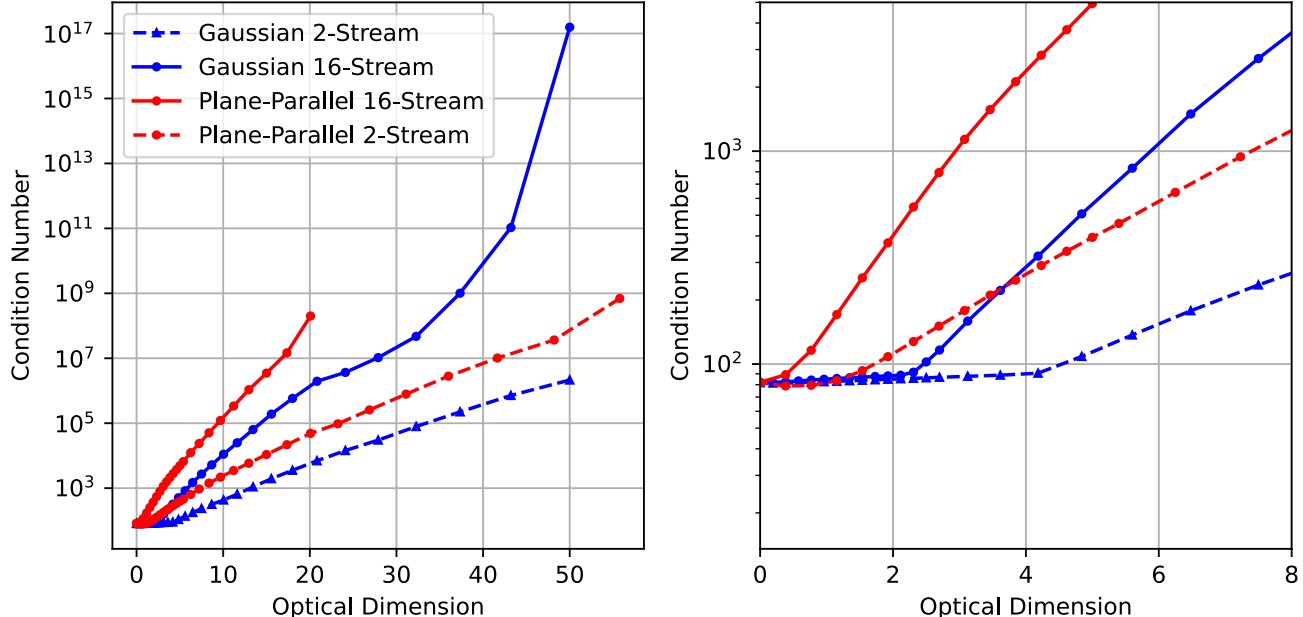

**Figure 17: The dependence of the condition number of the approximate Jacobian on the "Optical Dimension" which is the vertical optical thickness of plane-parallel homogeneous (red lines) and the optical radius of 3D gaussian (blue lines) extinction fields. The right panel is a zoomed version of the left one, highlighting the different behaviours at small optical depths. The plane-parallel 16-Stream curve stops at maximum optical thickness 20 since beyond this it becomes truly ill-posed (condition number tending to infinity).**

In the right panel of Fig. 17, we see that the condition number of the approximate Jacobians in the 3D gaussian clouds remains roughly invariant before the onset of exponential growth. This is likely due to the fact that the Knudsen Number remains large enough that a diffusion regime has not developed within the cloud. The reduced angular accuracy results indicate that the clouds are equivalent to those operating with almost halved extinction, as expected. This includes the transition point from slow-scaling to exponential scaling for the 3D gaussian extinction field.

**6.2 Discussion**

The results in Fig. 17 indicate that plane-parallel clouds are a lot more ill-posed under inversion than finite clouds, and will require stronger regularization of prior information. This fact highlights the fact that, from a fundamental perspective, highly heterogeneous cloud fields are actually much simpler targets for remote sensing than homogeneous, stratiform clouds as the vertical variations within the cloud can be inferred much more easily from passive imagery. Additionally, the results in Fig. 17 also indicate that tomographic retrievals of optically thin ($\tau \leq 3$) stratiform media (e.g., cirrus), should be as effective as for other cloud geometries with similar vertical optical depth. As these optically thin cirrus clouds are well conditioned, we should anticipate successful tomographic retrievals. This suggests that the proposed retrieval algorithm will be most effective for broken trade cumulus, thin cirrus and also possibly multi-layered combinations of them, along with their aerosol



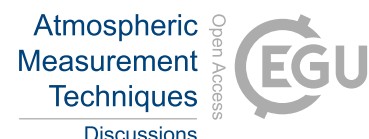

environment. Tomographic retrievals will therefore be highly effective in filling the observational gap in operational cloud and aerosol property retrievals using the IPA, which do not perform well in situations with broken cumulus.

It remains to be seen how effective tomography will be for very thick, cumuliform clouds and for moderately thick stratiform clouds, all of which are strongly ill-conditioned. It also remains to be seen what regularization schemes or prior information are required to improve retrievals in these conditions. There are also methods that have been proposed to mitigate the ill-conditioning of the inverse problem through the use of tailor-made preconditioning schemes (Niu et al., 2010; Tian et al., 2010).

## 7 Summary

In this study, we have introduced and validated an algorithm for retrieving the 3D volumetric properties of clouds using multi-angle, multi-pixel radiances and 3D radiative transfer. The retrieval utilizes an iterative, optimization-based solution to the generalized least-squares problem to find a best-fitting state vector parameterizing the atmosphere. The iterative retrieval is made computationally tractable through the use of an approximate Jacobian calculation introduced by Levis et al. (2015, 2017, 2020) that has been extended to accommodate open and periodic horizontal boundary conditions and an improved treatment
of non-black surfaces. We implemented this retrieval in a new software package, AT3D, which we have made publicly available.

We presented the basic physical principles of inverse radiative transfer from a linearized perspective. We identified that the iterative retrieval will tend to ill-posedness as the optical depth of the medium increases. This is due to the increasing smoothing
effects of multiple-scattering, which are ill-conditioned under inversion. This ill-conditioning of the inversion is also highly sensitive to the numerical treatment of the forward-scattering peak for highly peaked phase functions such as cloud droplets at solar wavelengths. In the SHDOM solver used in the retrieval algorithm, this manifests as a sensitivity to both the phase function and also the angular resolution used in the solver. When forward scattering peaks are strong and angular resolutions are low, then the ill-conditioning is mitigated.


Our linear analysis of the cloud tomography problem also indicates that the fastest reductions in the cost function will occur by modifying regions of the cloud optically close to the sun and to the sensors, where the magnitude of the elements of the Jacobian matrix are the largest. As the cloud becomes optically thick, the magnitude of the elements of the Jacobian matrix become exponentially larger in the regions close to the sun and sensors than those furthest away. This may cause a retrieval
using local optimization, such as AT3D described here, to converge to a local minimum when the target medium is optically thick if no other constraints are employed in the retrieval other than the multi-angle radiances.



We presented the derivation of the approximate Jacobian as an approximation to an adjoint radiative transfer problem and evaluated its accuracy. Errors in the elements of the approximate Jacobian matrix which contain derivatives of radiances with respect to the 3D volume extinction coefficient increase from 2% to 12% for media with cloud-like single scattering properties over surfaces with Lambertian albedos less than 0.2 as the maximum optical depths of the medium increase from 0.2 and 100. When the albedo of the Lambertian surface is 0.7 the errors are larger, reaching 20% for media with maximum optical depths of 100. Errors are smaller for media with phase functions with strong forward scattering peaks, especially when low angular resolution is utilized in the SHDOM solver.

The elements of the approximate Jacobian matrix that contain derivatives of radiances with respect to the volume extinction coefficients of the plane-parallel media that provide the open horizontal boundary conditions to the 3D radiative transfer problem are very inaccurate. Errors in these elements of the approximate Jacobian matrix exceed 50% unless the plane-parallel media are optically thin (~0.1). The larger errors in the elements of the Jacobian matrix that correspond to the horizontal boundary conditions than to those that correspond to the volumetric optical properties of the medium are due to the accuracy of the approximate calculation of the Jacobian matrix, which has lower accuracy in lower dimensional (2D and 1D) transport problems. These results indicate that AT3D will likely only be useful for jointly retrieving volumetric optical properties and horizontal boundary conditions when the horizontal boundary conditions correspond to optically thin plane-parallel media like the clear atmosphere.

The approximate Jacobian captures the key information content in the reference Jacobians and has a similar dependence of ill-conditioning on the optical depth of the medium as the reference. Our numerical tests using the approximate Jacobian also indicate that the retrieval problem becomes much more ill-posed, as measured by the condition number, when the target medium forms an infinite slab geometry than when it forms a finite geometry. This difference is due to the inability of the sensors to distinguish between rearrangements in the extinction field at the bottom of an optically thick plane-parallel layer and indicates that tomographic retrievals will be most beneficial in optically thinner stratiform clouds or broken fields of cumulus.

We therefore judge that the approximate Jacobian, and by extension the retrieval method currently available in AT3D for tomographic retrievals, to be suitable for retrievals in thin, cirriform clouds and trade cumulus over oceanic surfaces and their adjacent aerosols. The successful application of the retrieval to a broader variety of clouds and surface types is also possible but will likely require the incorporation of additional constraints. In Part 2 of this study, we will examine the implications of the ill-conditioning and errors in the approximate Jacobian in idealized tomographic retrievals of simulated clouds. In Part 2, we focus on the retrieval of 3D volume extinction coefficient using monochromatic radiance to address these fundamental concerns. The AT3D software is already able to examine a much wider variety of problems and can be used to explore microphysical retrievals using polarimetric (Levis et al. 2020) or multispectral measurements. We hope that in making this



software package publicly available we will encourage the development of this retrieval method and other "next-generation" remote sensing retrievals that utilize 3D radiative transfer modelling.

**Appendices**

1395 **Glossary**

| | | |
|---|---|---|
| $\boldsymbol{a}$ | State vector. | Eq. (1) |
| $\tilde{\boldsymbol{a}}$ | Retrieved state vector. | Eq. (2) |
| $\boldsymbol{a_0}$ | Initial value of state vector. | Figure 1. |
| $\Delta\boldsymbol{a}_m$ | Update to state vector at $m^{\text{th}}$ iteration of optimization. | Eq. (3) |
| 1400   $a_{tol}$ | Absolute error tolerance when testing for agreement between floats. | Eq. (B1) |
| $\boldsymbol{B}(\boldsymbol{r})$ | Polarized volume emission source. | Near Eq. (13) |
| $B_\lambda(T)$ | Planck black-body radiance function. | Near Eq. (13) |
| $\boldsymbol{b}$ | A vector of output from AT3D on which to test for accuracy. | Eq. (B1) |
| $\boldsymbol{c}$ | A vector of reference output used to evaluate AT3D | Eq. (B1) |
| 1405   $D$ | Spatial domain of RT. | Near Eq. (7) |
| $\partial D_n$ | The boundary of a spatial domain. | Eq. (11) |
| $D_1$ | Spatial domain of 3D RT. | Eq. (7) |
| $D_2 - D_9$ | Spatial domain of 2D and 1D RT (evens 1D, odds 2D). | Eq. (8) |
| $D_n \times \mathbb{S}^2$ | Internal set of radiative transfer (space and direction). | Figure 3. |
| 1410   $\delta$ | Dirac delta distribution. | Eq. (17) |
| $\boldsymbol{F}(\boldsymbol{a})$ | Forward model. | Eq. (1) |
| $F_{i,n}(\boldsymbol{x})$ | The contribution of the $n^{th}$ RT domain to the $i^{\text{th}}$ component of the forward model. | Eq. (42) |
| $\boldsymbol{F_0}$ | The flux along the solar beam. | Eq. (17) |
| 1415   $\boldsymbol{f}(\boldsymbol{r},\boldsymbol{\Omega})$ | The volume source of the RTE. | Eq. (13) |
| $\boldsymbol{f}_{\text{d}}(\boldsymbol{r},\boldsymbol{\Omega})$ | The volume source of the RTE for diffuse radiance. | Eq. (26) |
| $\hat{\boldsymbol{f}}_{\text{d}}(\boldsymbol{r},\boldsymbol{\Omega})$ | The effective volume source of the RTE for diffuse radiance. | Eq. (33) |
| $\boldsymbol{g}(\boldsymbol{r},\boldsymbol{\Omega})$ | The boundary source of the RTE. | Eq. (15) |
| $\boldsymbol{g}^{\text{BOT}}(\boldsymbol{r},\boldsymbol{\Omega})$ | Boundary source at domain bottom (thermal emission). | Eq. (27) |
| 1420   $\boldsymbol{g}^{\text{TOP}}(\boldsymbol{r},\boldsymbol{\Omega})$ | Boundary source at domain top (isotropic emission). | Eq. (27) |
| $\boldsymbol{g}_\odot(\boldsymbol{r},\boldsymbol{\Omega})$ | Boundary source at domain top due to collimated Solar illumination. | Eq. (17) |
| $\boldsymbol{g}^{\text{SIDE}}(\boldsymbol{r},\boldsymbol{\Omega})$ | Boundary source on domain sides. | Eq. (18) |



| | | |
|---|---|---|
| $\boldsymbol{g}_{\odot}^{\mathrm{SIDE}}(\boldsymbol{r},\boldsymbol{\Omega})$ | Boundary source on domain sides due to direct, un-scattered solar illumination. | Eq. (22) |
| $\boldsymbol{g}_{\mathrm{d}}^{\mathrm{SIDE}}(\boldsymbol{r},\boldsymbol{\Omega})$ | Boundary source on domain sides not due to direct, un-scattered solar illumination. | Eq. (22) |
| $\boldsymbol{g}_{\mathrm{d}}$ | The boundary source of the RTE for diffuse radiance. | Eq. (27) |
| $\hat{\boldsymbol{g}}_{\mathrm{d}}(\boldsymbol{r},\boldsymbol{\Omega})$ | The effective boundary source of the RTE for diffuse radiance. | Eq. (34) |
| $\Gamma_n^-$ | Incoming radiation at the boundary of the $n^{\mathrm{th}}$ RT domain. | Eq. (11) |
| $\Gamma_n^+$ | Outgoing radiation at the boundary of the $n^{\mathrm{th}}$ RT domain. | Eq. (11) |
| $\Gamma_n^{\mathrm{S}}$ | Incoming radiation at the horizontal sides of the $n^{\mathrm{th}}$ RT domain. | Eq. (37) |
| $H_L(x)$ | A Heaviside function with transition at length $L$. | Eq. (38) |
| $i$ | Index of forward model. | Eq. (5) |
| $\boldsymbol{I}_n(\boldsymbol{r},\boldsymbol{\Omega}) = [I,Q,U,V]^T$ | Stokes vector (polarized radiance field) of the $n^{\mathrm{th}}$ domain. | Eq. (12) |
| $\boldsymbol{I}_{\odot}(\boldsymbol{r},\boldsymbol{\Omega})$ | Direct, un-scattered solar radiance. | Eq. (21) |
| $\boldsymbol{I}_{\mathrm{d}}(\boldsymbol{r},\boldsymbol{\Omega})$ | Diffuse radiance. All radiance that isn't direct un-scattered solar radiance. | Eq. (21) |
| $j$ | Index of state vector elements. | Eq. (5) |
| $k$ | Index of sub-piexl rays in a sensor response function. | Eq. (32) |
| $\mathbf{K}$ | Jacobian matrix containing the derivatives of the forward model with respect to elements of the state vector. | Eq. (4) |
| $\kappa(\mathbf{K})$ | Condition Number (of Jacobian matrix). | Eq. (6) |
| $\boldsymbol{l}$ | Lower bounds on state vector. | Eq. (2) |
| $l$ | Path length | Eq. (24) |
| $l_e$ | Path length over a subgrid interval | Eq. (C2) |
| $L_x$ | Length of 3D RT domain in x-direction. | Eq. (7) |
| $L_y$ | Length of 3D RT domain in y-direction. | Eq. (7) |
| $L_z$ | Length of 3D RT domain in z-direction. | Eq. (7) |
| $\mathcal{L}[\cdot]$ | The transport operator of the RTE. | Eq. (12) |
| $m$ | Iteration number in L-BFGS-B optimization. | Eq. (2) |
| $M$ | The number of past L-BFGS-B iterations over which gradient information is stored to approximate the Hessian of the cost function. | Near Eq. (3) |
| $n$ | Domain index of RT domains. | Eq. (11) |
| $\boldsymbol{n}_n(\boldsymbol{r})$ | Unit vector outer normal to the boundary of the $n^{\mathrm{th}}$ domain. | Eq. (11) |
| $\boldsymbol{O}_i$ | The polarization analyser of the sensor response function of the $i^{\mathrm{th}}$ | Eq. (32) |



| | element of the forward model. | |
|---|---|---|
| $\boldsymbol{\Omega}$ | Direction of propagation. | Eq. (11) |
| $\boldsymbol{\Omega}_{\mathrm{sun}}$ | Direction of propagation of solar illumination. | Eq. (17) |
| $\boldsymbol{\Omega}_{ik}$ | Propagation direction of radiance that is sampled by the $k^{\mathrm{th}}$ sub-pixel ray of the sensor response function for the $i^{\mathrm{th}}$ element of the forward model. | Eq. (32) |
| $\omega(\boldsymbol{r})$ | Single scattering albedo. | Eq. (12) |
| $\boldsymbol{P}_i$ | Sensor response function of the $i^{\mathrm{th}}$ measurement. | Eq. (32) |
| $\boldsymbol{p}_{ik,n}$ | The volume response function of the $k^{\mathrm{th}}$ sub-pixel ray of the sensor response function for the $i^{\mathrm{th}}$ element of the forward model for the $n^{\mathrm{th}}$ RT domain. | Eq. (39) |
| $\boldsymbol{q}_{ik,n}$ | The attenuated boundary response function of the $k^{\mathrm{th}}$ sub-pixel ray of the sensor response function for the $i^{\mathrm{th}}$ element of the forward model for the nth RT domain. | Eq. (40) |
| $\boldsymbol{r} = (x, y, z)^T$ | Position vector. | Eq. (7) |
| $\boldsymbol{r}_{ik}$ | Position of the $k^{\mathrm{th}}$ sub-pixel ray for the sensor response function of the $i^{\mathrm{th}}$ measurement. | Eq. (32) |
| $r_{tol}$ | Relative error tolerance when testing for agreement between floats. | Eq. (B1) |
| $R(\boldsymbol{a})$ | Regularization term in cost function. | Eq. (1) |
| $\mathcal{R}$ | Reflection operator part of the boundary condition of the RTE. | Eq. (15) |
| $\mathbf{R}(\boldsymbol{r}, \boldsymbol{\Omega}, \boldsymbol{\Omega}')$ | Polarized Bidirectional Reflectance Distribution Function. | Eq. (16) |
| $\mathbf{S}_{\epsilon}$ | Error co-variance matrix. | Eq. (1) |
| $\mathbb{S}^2$ | Unit-sphere in 3D which is the domain of directions of RT. | Eq. (11) |
| $\mathrm{d}S_{\boldsymbol{\Omega}}$ | An element of surface parameterized by the propagation direction. | Eq. (29) |
| $\mathrm{d}V_{\boldsymbol{r}}$ | An element of volume parameterized by the position vector. | Eq. (29) |
| $\mathcal{S}[\cdot]$ | Volume streaming operator. | Eq. (35) |
| $\sigma(\boldsymbol{r})$ | Volume extinction coefficient. | Eq. (12) |
| $s_{largest}$ | Largest singular value of Jacobian matrix. | Eq. (6) |
| $s_{smallest}$ | Smallest singular value of Jacobian matrix. | Eq. (6) |
| $\mathcal{T}[\cdot]$ | Streaming operator. | Eq. (23) |
| $\mathrm{T}(\boldsymbol{r}, \boldsymbol{r}')$ | Transmission between two positions. | Eq. (24) |
| $\mathcal{U}$ | Solution operator of the RTE. | Eq. (28) |
| $\boldsymbol{u}$ | Upper bounds on state vector. | Eq. (2) |



| | | |
|---|---|---|
| $\boldsymbol{v}$ | Test field. | Eq. (29) |
| 1490 $\boldsymbol{w}$ | Test field. | Eq. (29) |
| $w_k$ | The weights of the sub-pixel rays forming the sensor response function. | Eq. (32) |
| $\chi^2$ | Cost function. | Eq. (1) |
| $\boldsymbol{y}$ | Measurement vector. | Eq. (1) |
| 1495 $\boldsymbol{Z}(\boldsymbol{r}, \boldsymbol{\Omega}, \boldsymbol{\Omega'})$ | Scattering phase matrix. | Eq. (12) |
| $\oplus$ | Set union operator. | Near Eq. (12) |

## Appendix A: Modifications to the SHDOM solver

The SHDOM solver in AT3D is derived from the original implementation of SHDOM in Fortran 77 and Fortran 90 (Evans 1998). We have made some modifications to the solver and optical property schemes to ensure the differentiability of radiances
with respect to all optical properties and microphysical properties. The SHDOM solver uses two different grids, one to represent the optical properties, known as the property grid, and one to solve the RT problem. The property grid is regular. The RT grid is based on a regular grid, but with optional local grid refinement that makes the grid irregular, and is referred to as the adaptive grid. Specific choices must be made to both prepare the optical properties on the property grid and to interpolate them onto the RT grid.


The original optical property generation scheme (PROPGEN) prepared the optical properties through external mixing of different participating particle species. This means that the total volume extinction coefficient and single scatter albedo are calculated by summing the volume extinction coefficients and the volume scattering coefficients. The calculation of the effective phase functions of the mixture would require calculating the weighted mean of the phase functions, where the weights
are the fractional contribution of each species to the total volume scattering coefficient. This scheme would typically produce a unique phase function for each point, which would cause a substantial memory burden in the SHDOM solution, especially if polarization is considered. This is also an issue when the adaptive grid is used in the SHDOM solution, as the number of grid-points can grow, with each one requiring a new unique phase function.

To alleviate this, the original SHDOM PROPGEN program limited the number of unique phase function mixtures by only adding a new phase function if a tolerance on the phase function accuracy is not met by any of the phase functions within the current set. This smaller set of unique phase functions is then stored in a Look-Up Table (LUT), with a corresponding pointer at each grid-point. This scheme is non-differentiable as a thresholding operation is used to select the phase functions.
When a new adaptive grid point requires new optical properties, the extinction and single scattering albedo at each point can
be calculated with linear interpolation. To avoid new unique phase functions, a less accurate nearest-neighbour interpolation



is used. Specifically, the new adaptive grid-point inherits the phase function of a property grid-point if either a) the two are co-located or b) the property grid-point is the one with the largest scattering coefficient amongst the property grid-points surrounding the new adaptive grid-point. This scheme is also not differentiable.

We have replaced the PROPGEN program and modified the SHDOM solver used in AT3D itself to accommodate a new scheme for representing phase functions that is differentiable. Specifically, we employ online mixing of phase functions when required during the SHDOM solution procedure, which is during the convolution of the phase function and the radiance field referred to as the source function computation. We note that the delta-M scaling occurs on the RT grid, consistent with the original SHDOM implementation. This operation is performed by the subroutine COMPUTE_SOURCE in the shdomsub1.f

file. In the online mixing, we store a limited LUT for each scattering species. We may adopt a nearest-neighbour or linear-interpolation from this LUT to define the phase function of each species at each grid-point. We then calculate the total phase function of all species at each grid-point by performing a weighted average over the phase function of each species. This latter part of the procedure is the "exact" part of the phase function calculation. The approximation can occur depending on the choice of phase functions within the LUT and the choice of interpolation rule. When the number of unique phase functions is

small, each entry in the LUT can be the species phase function. On the other hand, when the number of unique phase functions is large, we approximate the dependence of the phase function on the particle's microphysical properties using linear interpolation. This approximation occurs for each individual particle species, but the mixing is still exact. Both the mixing and linear interpolation from the LUT are differentiable. The downside of the interpolation from the LUT is that some error will be induced compared to using the exact Mie calculations for each phase function.


The final result of this, is that we have replaced the memory burden of storing phase functions with the computational burden or performing weighted sums of phase functions. The computational burden is relatively low because the number of phase function expansion coefficients that need to be mixed for the SHDOM solution is relatively small. To be precise, it is small compared to the total number of expansion coefficients needed to store a unique phase function. It is also small compared to

the number of spherical harmonics in the source function computation. As such, the computational cost of the source function computation and the solver itself is relatively invariant, as shown in Table A1. Given that most simulations will have at most cloud liquid, cloud ice, and one or two types of aerosol, the computational expense of performing these simulations is only around 10% larger for typical angular accuracies. This is relatively large because the source computation is actually called 3 times during each solution procedure to reduce the memory expense of the SHDOM solver. Multi-species SHDOM solutions

are much more expensive for lower angular accuracies, but the solutions are extremely fast in these cases, resulting in minimal changes in wall time.

**Table A1: The ratio of the computational expense of the SHDOM solution procedure with the new source function computation for different number of particle species against the equivalent exact mixture, for varying angular resolution of the SHDOM solver. The media used in the SHDOM solutions are generated from uniform distributions of effective particle radius, extinction and single**
**scattering albedo. The medium has 832 radiative transfer grid points. Half of the particle species in each medium have cloud-sized**

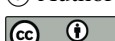



effective particle radii (10 to 30 microns), and the other half have aerosol-sized effective particle radii (0.3 – 0.5 microns). The timing ratios are averaged over 10 stochastic realizations for each configuration with the standard deviations of the ratios shown in brackets.

| Number of Spherical Harmonics | Number of Particle Species | | | |
|---|---|---|---|---|
| | 2 | 4 | 6 | 8 |
| 4 | 1.52 (0.07) | 2.16 (0.13) | 2.68 (0.22) | 3.42 (0.36) |
| 16 | 1.24 (0.15 | 1.57 (0.08) | 1.94 (0.15) | 2.26 (0.18) |
| 64 | 1.15 (0.11) | 1.39 (0.16) | 1.53 (0.16) | 1.66 (0.09) |
| 256 | 1.06 (0.06) | 1.11 (0.06) | 1.18 (0.05) | 1.28 (0.08) |
| 1024 | 1.02 (0.1) | 1.05 (0.07) | 1.07 (0.07) | 1.19 (0.06) |

## Appendix B: SHDOM Solver Verification

We have made substantial modifications to the implementation of the SHDOM solver in AT3D. These include the modifications to the representation of optical properties in the source function calculation described above in Appendix A, and also through the python wrapping of the SHDOM solution procedure. Due to these changes, it is critical to perform a thorough verification of the new software to ensure that bugs have not been introduced and the efficacy of the translation of the algorithm into code remains intact (Kanewala and Bieman, 2014). As such, we have performed a verification of the model to ensure consistency of the implementation of SHDOM in AT3D with the original SHDOM implementation and with several analytic benchmarks. The comparisons against the original SHDOM implementation were the most informative as they provided strict bounds on the behaviour of the model in complex cases, thereby allowing us to diagnose several errors, including one in the original SHDOM implementation. The details of all of the tests in AT3D, including the solver verifications described here can be found in AT3D's tests folder, including input files for the original SHDOM code.

In our quantitative verification of AT3D's SHDOM solver, we must compare arrays of floating-point numbers. Given our changes to the optical property and source function calculations in the solver, it is not possible to test for bit perfect reproduction of the original SHDOM solver. The comparison of outputs is also difficult due to the need to verify the solution when the





adaptive grid is used. The adaptive grid scheme utilizes a thresholding operation on a splitting criterion, which is a floating-point number, to decide when to refine the grid. This thresholding is very sensitive to even changes in the inputs or numerical operations at the level of the numerical precision of the single precision floats. The changes in the solution in the RT solution (e.g. fluxes and radiances) due to one additional refinement of the grid can be much larger (i.e., several percent) than changes in the inputs. As such, the adaptive grid scheme can amplify small differences in inputs into much larger differences in the

outputs and is numerically unstable. Care must be taken to manage this issue when performing the comparisons with original SHDOM. To test for the presence of significant differences between a vector of output from the AT3D solver, $\boldsymbol{b}$, and a vector of reference output (e.g. from the original SHDOM solver), $\boldsymbol{c}$, we test whether the absolute differences between the $N$ elements of the vectors exceed the sum of a prescribed relative error, $r_{tol}$, and absolute error $a_{tol}$. Specifically, we require

$$|b_i - c_i| \leq a_{tol} + r_{tol}|c_i| \quad \forall\, i \in 1, \dots, N, \tag{B1}$$

to hold for every element. For all tests, $r_{tol}$ is set to $10^{-5}$ while $a_{tol}$ is set according to the expected truncation error, which varies depending on the nature of the reference. When comparing between AT3D and SHDOM radiances we use output written to file with 5 significant figures (standard SHDOM output) and we require $a_{tol} = 10^{-5}$ as most outputs are of order unity. We also use custom untruncated output of the source function from SHDOM. SHDOM uses single-precision floats and so we expect agreement to within seven significant figures. As such, we judge the test to be successful in cases with untruncated

output if agreement is to within $a_{tol} < 10^{-6}$ though we test as strictly as we are able by reporting the smallest value of $a_{tol}$ for which each test passes. For example, we verified that the modified source function computation in AT3D due to the new treatment of the phase matrices passes the consistency test with $a_{tol} = 10^{-8}$.

We proceed with our description of the verification process in order of increasing complexity. We begin with the analytic

benchmarks, some comparisons of AT3D against SHDOM in plane-parallel atmospheres then a comparison against SHDOM in a complex 3D case.

**B1 Analytic and Numerical Benchmarks for 1D RT**

We compare AT3D's solver against analytic benchmarks to ensure the absolute accuracy of the solver in simple situations (Jones and Girolamo, 2018). The first such situation is a non-scattering homogeneously-absorbing atmosphere with a

Lambertian surface where radiances are verified with $a_{tol} = 10^{-8}$. We next consider an isothermal atmosphere with surface reflection to verify the thermal RTE solution. The nadir radiance is verified against an analytical solution to 0.005% for a Lambertian surface with an albedo of 0.5, a surface temperature of 300 K and an atmospheric temperature of 288 K with 50 optical paths ranging from 0.03 to 15. The accuracy of this result depends on the angular resolution for calculating the reflected radiation at the surface. 128 zenith discrete ordinate bins and 256 azimuthal discrete ordinate bins are used. The analytic

solution in this case is given by Eq. B9 in Jones & Di Girolamo (2018). We also verified a case with a non-scattering,





homogeneously-absorbing, and isothermal atmosphere with solar surface reflection and emission, which is a combination of two previous cases, to the same level of accuracy.

We compare RT solutions from AT3D and SHDOM for molecular (Rayleigh) scattering in plane-parallel atmospheres over all of the surface BRDFs available in SHDOM and AT3D with varying parameters (e.g. surface wind speed) to verify that the surface BRDFs, the Rayleigh scatter calculations and 1D RT are consistent between the two solvers. The details of the comparison can be found in AT3D code. We compare hemispheric downwelling fluxes, hemispheric upwelling fluxes and direct fluxes at the surface as well as TOA radiances for 20 angles spaced equally in cosine of zenith over the upwelling hemisphere. This is done through comparison with SHDOM output written to file, i.e. it is verified with $a_{tol} = 10^{-5}$.

## B3 Intercomparison against SHDOM

We use a 3D Radiative Transfer setup with three Stokes components to compare the RTE solution between SHDOM and AT3D for a cloud distributed with the AT3D code. This cloud was utilized in both Levis et al. (2015, 2020). Both AT3D and SHDOM use optical properties read from a file prepared by SHDOM's PROPGEN program to avoid amplification of errors due to the adaptive grid. In brief, the simulation uses 8 zenith discrete ordinate bins are used, 16 azimuthal discrete ordinate bins, a splitting accuracy of 0.1, a spherical harmonic accuracy of 0.01, and a solution accuracy of $10^{-4}$. The final output of the SHDOM solver is the spherical harmonic expansion of the angular convolution of the radiance field with the phase function, denoted as the polarized SOURCE vector, on which we perform the consistency test. When the adaptive grid matches between AT3D and the SHDOM reference, as in our example cloud, the solutions are consistent with $a_{tol} = 5 \times 10^{-7}$.

Larger differences in the SOURCE vector may occur when comparing SHDOM solutions across different machines and compilers when the adaptive grid is used as if the adaptive grid splitting occurs differently the SOURCE vectors will not have a one-to-one correspondence. The input files and scripts to reproduce the SHDOM benchmarks used in the test are distributed with AT3D along with the static SOURCE output from the original SHDOM. The static output may not be consistent with other machines and may need to be regenerated. Given the good agreement between the solvers for the SOURCE vector when the adaptive grid splitting is consistent, we judge that the AT3D implementation of the SHDOM solver is in good agreement with the original SHDOM implementation.

We also test the radiance calculation in the complex situation of 3D RT with the adaptive grid. We find that even in a simplified situation where the original source function computation is used, there is substantial disagreement in the calculated radiances, despite the good agreement of the source function computations. We identify discrepancies in the radiances that can, in rare cases, reach up to 1.5% error in the intensity, and as such are clearly noticeable even using the truncated output of SHDOM written to file as a reference. The root mean square errors are much smaller, being less than 0.004%, and are therefore are effectively undetectable in model intercomparisons. Given the good agreement of the SOURCE vectors, the differences in the





radiances between AT3D and the original SHDOM were traced to differences in the subroutines used to calculate the integrals
for the formal solution of the RTE (Eq. 36). The original SHDOM implementation uses the CALCULATE_RADIANCE
subroutine which is used in original SHDOM's Radiance Output mode. AT3D's radiance calculation builds off the
VISUALIZE_RADIANCE subroutine, which is used in original SHDOM's Visualization Output mode. The Visualization
output mode is more flexible in accommodating a unique viewing angle for each calculated radiance.

We do not have an absolute benchmark for the radiance calculation in such a complex situation with the adaptive grid that is
sufficiently precise. Instead, we determined the cause of the discrepancy through code analysis and judged that the AT3D
implementation is more physically correct based on the following considerations. The discrepancy only occurs when the
adaptive grid is used and is due to differing implementations of the interpolation of the source-extinction product onto the
characteristic of the radiance when moving between cells with different resolution in the adaptive grid. In SHDOM, the source-
extinction product at the face of the most recently exited cell is always used as the source-extinction product at the entry point
of the next cell. However, if going to a higher resolution cell, the source-extinction product at the entry point can be estimated
with higher accuracy using the higher resolution grid-points in the new cell. This latter procedure is used in AT3D, which we
judge to be more accurate. We note that the significant discrepancies are limited to rare cases. The errors will be largest when
there is a large source function difference between the source function at the higher and lower resolution grid-points. Therefore,
the errors will also be minimized when a low value of splitting accuracy is used. Our example used a relatively large splitting
accuracy of 0.1 and even then, the bug correction produced a very low root-mean-square change in the radiance field (0.004%).
As such, the correction of this bug will not substantially affect holistic benchmarks of radiances in 3D RT, such as reciprocity
(e.g. Di Girolamo 1999, 2002) or previous model intercomparisons as intermodel differences in radiances are much larger.

**Appendix C: Implementation of the approximate Jacobian matrix calculation**

Here, we describe how the inner products used to calculate the entries in the Jacobian matrix (Eq. 58) are numerically evaluated.
These details are relevant to the performance of AT3D, as there are several strategies that can be employed when performing
forward-adjoint based linearization of a model, depending on whether adjoints are formed from continuous, discrete or
numerical algorithms (Klose and Hielscher, 2002). First, we consider the evaluation of the integral involving the volume
streaming operator, the second term in Eq. (58). This integral is simply a line integral as it only needs to be evaluated at the
specific position and angle $(\boldsymbol{r}_{ik}, \boldsymbol{\Omega}_{ik})$ corresponding to a quadrature point in the sensor's response function:

$$\mathcal{S}\big[\Delta\hat{\boldsymbol{f}}_j\big|_{D_n \times \mathbb{S}^2}\big](\boldsymbol{r}_{ik}, \boldsymbol{\Omega}_{ik}) = \int_0^l \Delta\boldsymbol{f}_j(\boldsymbol{r}_{ik} - l'\boldsymbol{\Omega}_{ik}, \boldsymbol{\Omega}_{ik}) \exp\left(-\int_0^{l'} \sigma(\boldsymbol{r}_{ik} - l''\boldsymbol{\Omega}_{ik})dl''\right)dl', \tag{C1}$$

This integral is approximated by dividing it into $E$ subintervals with optical paths smaller than a specified tolerance (default
0.1) with endpoints $l_e$ and $l_{e+1}$:





$$S\big[\Delta\hat{f}_j\big|_{D_n\times\mathbb{S}^2}\big](\boldsymbol{r}_{ik},\boldsymbol{\Omega}_{ik}) = \sum_e \exp\left(-\int_0^{l_e}\sigma(l')dl'\right)\int_{l_e}^{l_{e+1}}\Delta\boldsymbol{f}_j(l')\exp\left(-\int_{l_e}^{l'}\sigma(l'')dl''\right)dl'. \tag{C2}$$

We use a formula for the integration over each subinterval that assumes that $\Delta\boldsymbol{f}_j$ and the extinction $\sigma$ vary linearly along the path. This formula is accurate to second order in the subinterval path length.

$$\int_{l_e}^{l_{e+1}}\Delta\boldsymbol{f}_j(l')\exp\left(-\int_{l_e}^{l'}\sigma(l'')dl''\right)dl' \approx \left(1-e^{-\frac{1}{2}(l_{e+1}-l_e)(\sigma(l_e)+\sigma(l_{e+1}))}\right)\frac{2}{\sigma(l_e)+\sigma(l_{e+1})}\left\{\frac{1}{2}\Big(\Delta\boldsymbol{f}_j(l_e)+\Delta\boldsymbol{f}_j(l_{e+1})\Big)+\right.$$
$$\left.\frac{l_{e+1}-l_e}{12}\Big(\sigma(l_{e+1})\Delta\boldsymbol{f}_j(l_n)-\sigma(l_e)\Delta\boldsymbol{f}_j(l_{n+1})\Big)\left(1-\frac{l_{e+1}-l_e}{20}[\sigma(l_e)-\sigma(l_{e+1})]\right)\right\}. \tag{C3}$$

This formula is approximate but consistent with the discretization of the forward model, with one exception. Namely, the first
term $\Delta\boldsymbol{f}_j$ involves the radiance, which doesn't actually vary linearly since the source-extinction product varies linearly in the forward model. To evaluate Eq. (C3), all that is needed is to evaluate $\Delta\boldsymbol{f}_j$ and the extinction at the end points of each sub-interval. We adopt a trilinear interpolation rule from the RT grid-points for the volume extinction coefficient and all terms in $\Delta\boldsymbol{f}_j$ apart from the radiance. For the radiance, we directly evaluate it at the endpoints of each sub-interval by evaluating the forward model.

The remaining complication is the evaluation of $\Delta\boldsymbol{f}_j$ in the required direction. If the delta-M approximation and TMS correction are not employed, then all angularly varying quantities, including the single-scattering terms in $\Delta\boldsymbol{f}_j$, are represented in a basis of (real) generalized spherical harmonics as in the forward model. On the other hand, if the delta-M approximation and TMS correction are employed, then the single scattering terms in $\Delta\boldsymbol{f}_j$ are computed using a LUT of phase functions dependent on scattering angle, again consistent with the forward model.

Secondly, we consider the boundary integral in the radiance derivative calculation, which is the first term in Eq. (58):

$$\mathcal{T}\big[\Delta\hat{g}_{n,j}^0(\boldsymbol{r},\boldsymbol{\Omega})\big|_{\Gamma_n^-}\big](\boldsymbol{r}_{ik},\boldsymbol{\Omega}_{ik}) = \mathcal{T}\left[\mathcal{R}\left[\frac{\partial\boldsymbol{I}_\odot}{\partial a_j}\Big|_{\Gamma_+^n}\right]+\frac{\partial\mathcal{R}}{\partial a_j}\big[\boldsymbol{I}_{n,\odot}+\boldsymbol{I}_{n,\mathrm{d}}\big]+\frac{\partial g_n^{\mathrm{TOP}}(\boldsymbol{r},\boldsymbol{\Omega})}{\partial a_j}+\frac{\partial g_n^{\mathrm{BOT}}(\boldsymbol{r},\boldsymbol{\Omega})}{\partial a_j}+\frac{\partial g_n^{\mathrm{SIDE},0}(\boldsymbol{r},\boldsymbol{\Omega})}{\partial a_j}\right] \tag{C4}$$

Currently, in AT3D, only the first term is relevant, as all other terms are neglected. Note that the horizontal boundary derivative term, $\frac{\partial g_n^{\mathrm{SIDE},0}(\boldsymbol{r},\boldsymbol{\Omega})}{\partial a_j}$, is made up of volume and boundary terms in other auxiliary domains so that the first term is the only true boundary term that is considered. The direct radiance derivative is calculated at every RT grid-point, including on the boundary.
The reflection operator $\mathcal{R}$ in Eq. (16) acts on these grid-points. The result is then bi-linearly interpolated along the boundary to the position where it needs to be evaluated, which is the intersection of the line of sight of the sensor with the boundary $(\boldsymbol{r}_{ik}-l\boldsymbol{\Omega}_{ik},\boldsymbol{\Omega}_{ik})$.





**Appendix D: Verification of the approximate Jacobian matrix calculation**

It is critical to perform a proper verification of the approximate calculation of the Jacobian matrix before we can draw scientific conclusions about the behaviour of the proposed algorithm with numerical experiments (Kanewala and Bieman, 2014). To verify the approximate Jacobian matrix, we use a combination of benchmarks calculated from simple scenarios with analytic solutions and finite-differencing benchmarks. All of the analytic benchmarks come from consideration of the non-scattering thermal emission or surface reflectance cases.

If we have an emissive surface and a homogeneously-absorbing isothermal atmosphere, then we can easily calculate the derivative of the measured radiance with respect to the homogeneous extinction in terms of the black body radiances of the atmosphere $B_A$ and the surface $B_S$, i.e.,

$$\frac{\partial I}{\partial \sigma}\Big|_{\sigma_0} = \frac{h}{\mu}e^{\sigma_0 h/\mu}(B_A - B_S), \tag{D1}$$

where $h$ is the geometric thickness of the layer and $\mu$ is the cosine of the viewing zenith angle. This test can be used to determine correct discrete implementation of the derivatives consistent with the formal, integral solution of the RTE used to calculate the radiances in SHDOM. We compare this analytic benchmark with the approximate Jacobian and a derivative estimated from a 2-point central difference. In this simple scenario, with $h = 0.4$ and $\mu = 1.0$, the relative error in the approximate Jacobian increases abruptly from $10^{-6}$ to $5 \times 10^{-3}$ when $\sigma_0 = 10.0$. This is due to the saturation of the error as many sub-intervals in the integration reach an optical path length of 0.1. Errors are substantially reduced if a smaller optical path length is used. Error rapidly balloons in optically thicker scenarios ($\sigma_0 = 100$) due to rounding error affecting the signal as the signal is attenuated below the precision of single-precision floats. The error in the 2-point central difference also grows as $\sigma_0$ increases but is also dependent on the step size. If a step size is chosen as the maximum of 1% or 0.01 then error is maintained below 6% for $\sigma_0 < 10$, where absolute error also peaks. This indicates that, despite the use of the approximate integration of the sub-intervals, the approximate Jacobian tends to be a bit more accurate than the finite-differencing with respect to the analytic calculation, for a case where the approximate Jacobian should be exact.

In the remaining tests, we can only utilize a finite-differencing reference as we invoke more complex randomly generated media (white noise) to test the correct application of the chain rule to the interpolation rules used for the optical properties. First, we test the calculation of the direct beam radiance derivatives which are verified with $a_{tol} = 10^{-5}$ in (B1) for both periodic and open BCs and for reflective Lambertian surfaces to ensure that the surface boundary term is correctly calculated. Second, derivatives of measured radiances with respect to the extinction, asymmetry parameter and single scatter albedo are verified in the optically thin limit and achieve respectively $a_{tol} = 9.2 \times 10^{-6}$, $a_{tol} = 1.3 \times 10^{-6}$, $a_{tol} = 5.3 \times 10^{-7}$, both with and without delta-M scaling.



## Appendix E: Calculating the Jacobian matrix by finite differencing

Here, we describe our procedure for calculating the Jacobian matrix using finite differencing to produce the results in Section 4 and the reference results against which the approximate Jacobian calculation is compared in Section 5. The step size for computing the derivatives is chosen to loosely balance truncation error and rounding error. For finite differencing, the solution accuracy and step size must be chosen so that computational noise does not dominate the derivative calculation. We choose the solution accuracy as $10^{-6}$. We cannot perform perfect optimization of the step size as the optimal step size for each element

of the Jacobian matrix varies with the measurement, not just the state, so it would require a separate derivative calculation for each element of the Jacobian matrix. Instead, we note from our verification of the approximate derivatives that the dominant error contribution appears to be due to rounding error. As such, we have chosen a relatively large step size that is still small enough that it does not unduly compromise accuracy for measurements that are optically close to the sensor and Sun, which have larger higher-order derivatives (Box, 2002) and therefore larger truncation error.


To perform the numerical differentiation using the central difference scheme, we must evaluate the forward model twice for each element of the state vector for a total of 2201 times for each base state, which results in a total of almost 240,000 3D RTE solutions for all base states. Though they are still each relatively small, we still accelerate these solutions to reduce computational expense by noting that the forward and backward perturbed RTE solutions will be very close to the base state

RTE solution. As such, we initialize the SHDOM solution of the perturbed RTE solutions with the RTE solution of the base state. This method is inspired by the acceleration method for multi-spectral SHDOM solutions described in Evans (1998). It rapidly accelerates convergence of the perturbed RTE solutions especially in the optically thickest, isotropically scattering cases where a reduction of ~100 in the number of SHDOM solution iterations is achieved. This finite differencing procedure described above is more computationally efficient than solving the tangent-linear model in Eq. (46) directly using SHDOM.

The tangent-linear model tends to take more iterations to converge than a forward problem (e.g., twice as much) and cannot be accelerated as the tangent-linear problem is very different from the base state RTE problem. The slower convergence of a tangent linear model is possibly due to the inappropriateness of the initial radiance field used in the SHDOM solution iterations, though we did not investigate this. The adaptive grid is used in these RTE solutions with splitting accuracy of 0.03. The adaptive grid from the base state RTE solution is used in the forward and backward perturbation solutions without further grid

splitting. This eliminates the adaptive grid as a source of computational noise in the finite differencing.

## Appendix F: Computational cost of the approximate linearization

We now compare the computational cost of the approximate linearization to that of the radiance calculation, specifically the line integrations described in Appendix D, with the forward model. The timing of the SHDOM solutions is therefore not considered, though it is documented elsewhere (Evans, 1998; Pincus and Evans, 2009). In particular, we compare the timing

of the subroutine LEVISAPPROX_GRADIENT (in shdomsub4.f) that evaluates both the radiances and the approximate





Jacobian to the subroutine RENDER (in shdomsub4.f), that only evaluates radiances. As such, we always expect the ratio of their timing to exceed unity.

The evaluation of the approximate gradient is expected to be much more computationally expensive than RENDER because

of the evaluation of the direct beam derivatives $\frac{\partial I_\odot(r, \Omega)}{\partial a_j}$ in $\Delta f_j$, which requires its own traversal of the grid along the solar direction from each grid-point. In our case, the implementation pre-computes the pointers to each grid-point along these solar paths and their relative contributions to the direct beam derivative. This pre-processing isn't included in our timings reported here as they contributed negligibly to the relative timing. In LEVISAPPROX_GRADIENT, operations must still be performed on each property grid-point along the solar direction. This results in an overall quadratic scaling of

LEVISAPPROX_GRADIENT with the number of property grid-points. Without this calculation, we expect a linear scaling with the number of adaptive RT grid-points for LEVISAPPROX_GRADIENT along with RENDER. To confirm this, we compute timings both with and without this calculation, referred to as "with exact single scatter" and "without exact single scatter" as this is the calculation that ensures accuracy of the approximate Jacobian in the single-scattering limit.

Our timing is based on a homogeneous, plane-parallel cloud. We compute the approximate Jacobian and radiances for a single pixel viewing at nadir repeated 10,000 times to ensure minimal influence of overhead on the calculated time-per-ray. The Sun is also at zenith. The cloud has a varying number of grid-points in the vertical, which controls computational cost as the line integrals are vertical, and angular resolution. Each cell is optically thin so that there is one sub-interval per cell and there is no adaptive grid splitting. This also affects the relative timing of the approximate Jacobian to the radiance calculation as the

quadratic cost of the exact single-scatter equation scales with the property grid (not the RT grid). As such, we are examining a worst-case scenario for the relative timing of the approximate Jacobian.

The computational cost as a function of the number spherical harmonics varies with the number of RT grid-points which can also vary relative to the property grid based on optical depth and grid splitting of the medium. Note that we compute only

extinction derivatives, which do not require computation of the additional angular integrals in $\Delta f_j$ beyond what is already computed for the radiance calculation. As such the scaling of the relative timing with the number of spherical harmonics (NSH), is likely underestimated compared to derivatives with respect to the phase function. In Fig. F1, we compare the timing ratio of LEVISAPPROX_GRADIENT to RENDER, per ray. Computations are performed on a 2.3 GHz Intel Core i5 and timings are computed using the python function time.process_time. The larger timing ratios for lower numbers of spherical

harmonics shown in Fig. F1 are due to the larger number of interpolation calculations required per sub-interval in the approximate Jacobian calculation than in the radiance calculation. The number of spherical harmonic calculations per grid-





point are common to both subroutines. As expected, the relative timing is independent of the grid size when the exact single-scatter calculation is not performed.

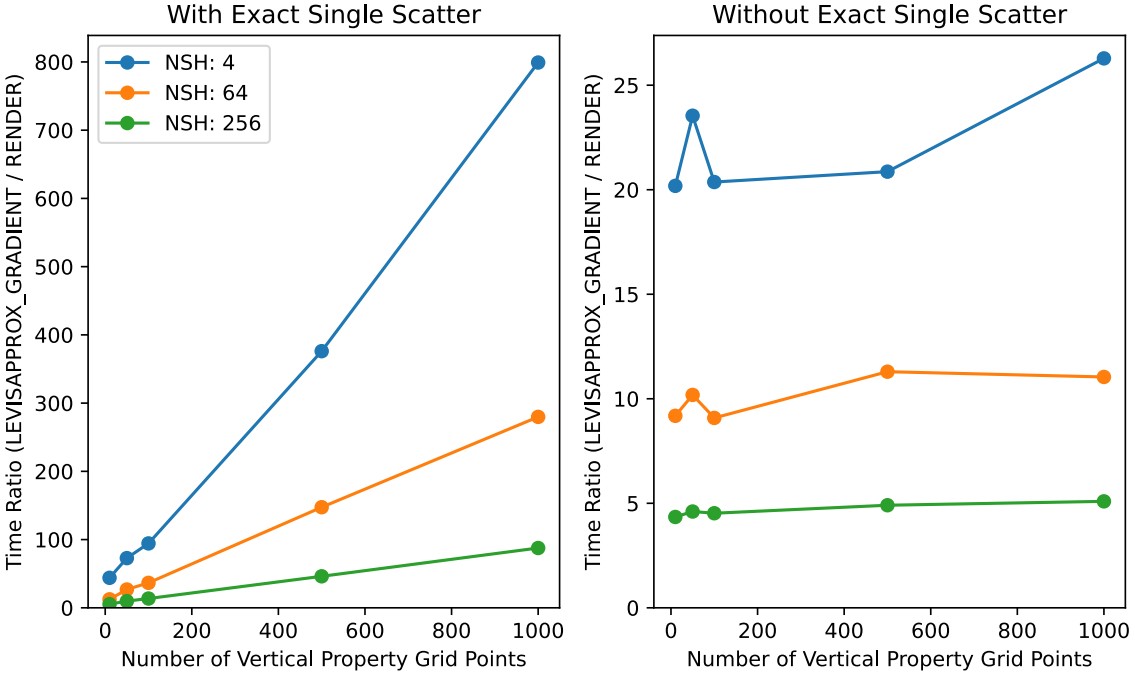

**Figure F1: Relative timings of the approximate Jacobian and radiance calculation (LEVISAPPROX_GRADIENT) and the radiance calculation only (RENDER) as a function of the number of grid-points for different numbers of spherical harmonics (NSH). The left panel shows the relative timing of the default approximate Jacobian calculation, which includes a term ensuring exact single scatter derivatives. The right panel shows the relative timing without that term. See text for details.**

## Code Availability

The software described and used in this paper called Atmospheric Tomography with 3D Radiative Transfer (AT3D). A static archive of the software is available at Loveridge et al. (2022). The most recent version is available from https://github.com/CloudTomography/AT3D. The original SHDOM code by Frank Evans is available from https://nit.coloradolinux.com/~evans/shdom.html.

## Author Contributions

JL performed the investigation and prepared the initial draft under the supervision of **LD**. **JL**, **LD**, **YY**, **AD**, and **AL** conceptualized the study. **JL**, **AL**, **VH** and **LF** developed the software. All authors contributed to the editing of the manuscript.



## Competing Interests

The authors declare that they have no conflict of interest.

## Acknowledgments

The authors would like to thank Frank Evans for making his SHDOM code publicly available.

Jesse Loveridge was supported by NASA's FINESST program under grant agreement 80NSSC20K1633. Aviad Levis is partially supported by the Zuckerman and Viterbi postdoctoral fellowships. This research was partially carried out at the Jet Propulsion Laboratory, California Institute of Technology, under a contract with the National Aeronautics and Space Administration (80NM0018D0004). AD was supported by the ROSES NRA Program Element TASNPP17-0165. Support
from the MISR project through the Jet Propulsion Laboratory of the California Institute of Technology under Contract 1474871 is gratefully acknowledged. LF was funded by the European Union's Framework Programme for Research and Innovation Horizon 2020 (2014–20) under the Marie Skłodowska-Curie Grant Agreement 754388 (LMUResearchFellows) and from LMUexcellent, funded by the Federal Ministry of Education and Research (BMBF) and the Free State of Bavaria under the Excellence Strategy of the German Federal Government and the Länder. Yoav Schechner is the Mark and Diane
Seiden Chair in Science at the Technion. He is a Landau Fellow - supported by the Taub Foundation. His work was conducted in the Ollendorff Minerva Center. Minerva is funded through the BMBF. This project has received funding from the European Union's Horizon 2020 research and innovation programme under grant agreement No 810370-ERC-CloudCT. The authors are grateful to the US-Israel Binational Science Foundation (BSF grant 2016325) for facilitating our international collaboration.

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
