# Peer review of "Retrieving 3D distributions of atmospheric particles using Atmospheric Tomography with 3D Radiative Transfer – Part 1: Model description and Jacobian calculation."

_Atmospheric Measurement Techniques, 2022_

## Referee Comment (RC1)

Review of the paper "Retrieving 3D distributions of atmospheric particles using Atmospheric Tomography with 3D Radiative Transfer – Part 1: Model description and Jacobian calculation" by Jesse Loveridge, Aviad Levis, Larry Di Girolamo, Vadim Holodovsky, Linda Forster, Anthony B. Davis, Yoav Y. Schechner.

Regarding this manuscript, I want to make some comments that are actually **not mandatory** for the authors. They can take into account my comments or not. If not, the article can be published in its **current form**. Over time, the authors have made an important contribution in this field so that now, they do not have to be hindered by technical details.

1. I do not understand why the side domains have periodic boundaries. A domain with these conditions is a domain that repeats itself infinitely in both directions. But, for example, $D_3$ is connected with $D_1$. In my opinion, a heterogeneous 3D domain embedded within a horizontally infinite medium can be modeled with periodic and open boundary conditions. In Fig. 8 of Evans' paper it is shown that for a Gaussian cloud, the flux computed with periodic boundaries for a 12-km domain agrees with that computed for open boundaries for a 3 km domain. Thus, in the case of periodic boundaries, the domain must be larger to diminish the contribution of the incoming radiance at the domain edges. That's all. However, because you need the domain $D_3$ to model the position of a sensor outside the cloud domain, I would consider that the domains $D_3$ and $D_1$ are connected by continuity, and I would impose open boundary condition on the left boundary of $D_3$. In other words, I would consider the computational domain $D_3 \cup D_1 \cup D_7$ as a whole, and impose open boundary conditions on the left boundary of $D_3$ and the right boundary of $D_7$.

2. In the case of satellite measurements, the main problem is the discretization of the domain above the clouds, which is usually meant up to 50-60 km. To connect the two domains (clear and cloudy), the idea used by Pincus and Evans for the parallelization of SHDOM can be used (the two domains communicate with each other through the boundary conditions).

3. I understand that the authors are fans of Martin. Indeed, Martin is an exceptional mathematician and his formalism is a mathematical delicacy. However, I think that it cannot be easily understood by a physicist or an engineer. For this reason, I want to suggest you to use a formalism that is more appropriate to Evans' implementation. By transforming Evans' implementation into a mathematical language, it is very easy to linearize the model's equations. In this way, different approximations for derivatives calculation, including the single-scattering approximation, become very clear.

---

## Author Comment (AC1)

Response to Reviewer #3

We are very grateful to the reviewers for taking the time to carefully review our manuscript. We respond to each reviewer's comments below. We have added line references to the positions in the tracked changes manuscript where we have made changes.

*Regarding this manuscript, I want to make some comments that are actually not mandatory for the authors. They can take into account my comments or not. If not, the article can be published in its current form. Over time, the authors have made an important contribution in this field so that now, they do not have to be hindered by technical details.*

We thank the reviewer for their helpful comments and their suggestions for future development of our method. We have addressed the comments below.

*I do not understand why the side domains have periodic boundaries. A domain with these conditions is a domain that repeats itself infinitely in both directions. But, for example, D3 is connected with D1. In my opinion, a heterogeneous 3D domain embedded within a horizontally infinite medium can be modeled with periodic and open boundary conditions. In Fig. 8 of Evans' paper it is shown that for a Gaussian cloud, the flux computed with periodic boundaries for a 12-km domain agrees with that computed for open boundaries for a 3 km domain. Thus, in the case of periodic boundaries, the domain must be larger to diminish the contribu- tion of the incoming radiance at the domain edges. That's all. However, because you need the domain D3 to model the position of a sensor outside the cloud domain, I would consider that the domains D3 and D1 are connected by continuity, and I would impose open boundary condition on the left boundary of D3. In other words, I would consider the computational domain D3 ∪ D1 ∪ D7 as a whole, and impose open boundary conditions on the left boundary of D3 and the right boundary of D7.*

This comment raised an important issue, which is that the physical reasoning for the formulation of the open boundary conditions is actually missing from our paper. We have added several sentences on this to the manuscript which we believe clarify this issue.

Line 443:

"The periodic boundary conditions in the auxiliary domains ensure that the RT solution in each auxiliary domain is independent of the 3D domain so that the system of RTs is solvable. This approximation neglects multiple scattering interactions between the heterogeneous medium ($D_1$) and the auxiliary domains. As a result, open horizontal boundary conditions are an approximate treatment of the RT solution for a heterogeneous medium embedded in a horizontally homogeneous medium. Features like cloud and surface adjacency effects cannot be modelled unless the domain of 3D radiative transfer is sufficiently large enough to resolve them."

The technical issue is that, while the radiance at the boundary between $D_3$ and $D_1$ is continuous in the directions incoming towards $D_1$, it is not continuous in the outgoing directions from $D_1$. If we make $D_1$ bigger, so that we resolve more of the plane-parallel medium with 3D radiative transfer, then this discontinuity will lessen, and the approximation of the RT with open horizontal boundary conditions will be more accurate. The example in Fig. 8 of Evans' paper is non-scattering as it is in the longwave and uses a black surface. In contrast to a scattering medium, the open horizontal boundary conditions provide an exact solution of the 3D RT for Evans' example even when the open horizontal boundary conditions are imposed right at cloud edge.

*In the case of satellite measurements, the main problem is the discretization of the domain above the clouds, which is usually meant up to 50-60 km. To connect the two domains (clear and cloudy), the idea used by Pincus and Evans for the parallelization of SHDOM can be used (the two domains communicate with each other through the boundary conditions).*

We agree that a treatment of the atmosphere is important either through explicitly resolving the full vertical column and making use of additional computational resources or by making use of approximate atmospheric correction methods given the weak atmospheric scattering at wavelengths typical for cloud remote sensing. We do intend to implement MPI-based parallelization for our method, which would be very valuable for multi-layered cloud scenes. The computational tradeoffs between the different approaches are a forward modelling issue beyond the scope of this paper and we opt not to discuss these ideas within the manuscript.

*I understand that the authors are fans of Martin. Indeed, Martin is an exceptional mathematician and his formalism is a mathematical delicacy. However, I think that it cannot be easily understood by a physicist or an engineer. For this reason, I want to suggest you to use a formalism that is more appropriate to Evans' implementation. By transforming Evans' implementation into a mathematical language, it is very easy to linearize the model's equations. In this way, different approximations for derivatives calculation, including the single-scattering approximation, become very clear.*

We agree that the mathematical formalism is quite formal and dense, but it has several benefits. The first benefit is that we are able to link the formalism of Martin et al. (2014) with the work of Levis et al. (2015, 2020) making the physical consequences of the approximate Jacobian calculation clearer. Secondly, Evans (1998) has only a very short description of the open boundary conditions and does not mention the influence of the boundary condition on the radiance calculation that occurs in the SHDOM code. If open boundary conditions may be more widely used for remote sensing retrievals using 3D radiative transfer, then it is worth describing this aspect of the model in detail. Thirdly, it is not possible to explain the much-reduced accuracy of the approximate Jacobian for the optical properties at the domain edge when using open boundary conditions without invoking the concept of a coupled system of RTEs. The description of the coupling is most clear when a precise, formal treatment is used rather than invoking high level analogies like in Evans (1998) (e.g. the concept of "independent scans") that are ambiguous in their meaning.

Because of these benefits, we have opted not to change the formalism within the manuscript. Instead, we have added a reference to other more pragmatic descriptions and clarified the relationship between our formalism and the details of implementation:

Line 381: "For a more pragmatic description of the essence of the approximate Jacobian calculation that does not include the treatment of the boundary conditions presented here, readers may refer to Levis et al. (2020). Our treatment focuses on the continuous problem rather than the details of numerical implementation such as delta-M scaling of the optical properties except where conceptually necessary. Pertinent details on the numerical implementation related to the delta-M scaling and TMS correction in SHDOM can be found elsewhere (Evans, 1998; Doicu and Efremenko, 2019) and in Appendices A and C."

---

## Author Comment (AC2)

Response to Reviewer #1

We are very grateful to the reviewers for taking the time to carefully review our manuscript. We respond to each reviewer's comments below. We have added line references to the positions in the tracked changes manuscript where we have made changes.

*This is a well written and informative paper that evaluates the capabilities and limitations of passive multi-angle observations for the 3D reconstruction of extinction fields.*

We thank the reviewer for their kind words and constructive comments which have improved the paper.

*The authors make good use of the formalism introduced by Martin et al. (2014), but fail to differentiate their own approach from that presented therein. Martin et al. (2014) uses the adjoint formalism to calculate the gradient of a cost function and never expresses the Jacobian in this manner. The author's use of the adjoint formulation to construct the Jacobian is therefore different to that of Martin and is interesting in its own right and that difference should perhaps be highlighted.*

We have clarified the provenance of ideas regarding the Jacobian calculation in the Introduction. (Line 141 and Line 151).

Martin et al. (2014) did in fact cover the formalism for calculating the entries of the Jacobian matrix. As they state in their paper, they didn't focus on it further due to the huge expense of computing the entire Jacobian matrix using adjoint (or forward) RTE. Doicu and Efremenko (2019) also covered the implementation of the appropriate forward-adjoint calculations in SHDOM. The approximate linearization of Levis et al. (2015, 2017, 2020) is the advance that gives computationally efficient access to the Jacobian matrix, though approximate.

Our contribution to Martin et al.'s formalism has been to extend it to include a treatment of open boundary conditions (as stated on Line 181 and again in Section 3.1).

In terms of further contributions distinct from Martin et al. (2014), we have derived the approximate Jacobian within the formalism of Martin et al. (2014) and validated it, and documented novel physical understanding about the cloud tomography problem gained from analyzing both the finite-difference and approximate Jacobians. We believe these contributions are already made clear in the paper so we do not make further modifications.

*It would probably benefit the paper if before Eq. (63) the authors note that it is a calculation of the Jacobian and include the dF/da term in the expression of the equation to remind the reader that is what it is and that it comes from Eq.(42) (we're 20 equations further in so a reminder is helpful). Also, given the later discussion of how areas that have strong illumination and "observation" are more sensitive/well constrained it would be worth noting after Eq.(43) how the "volume source vector of the modified RTE for the derivatives of the radiance field" depends*

*strongly on the strength of the direct and diffuse radiance. This can then be emphasized after Eq.(63) where is it is apparent that it is the combination of the strength of the adjoint sources and the illumination (via the volume source) that determine which areas of the cloud will contribute significantly to the Jacobian. The identification of this fact within the formalism of the adjoint calculation of the Jacobian is helpful in explaining the results that are described later in terms of delta-M transmission from solar and viewing directions.*

Equation 63 has been modified as suggested, and we have added short sentences to this effect:

Line 676:

"The spatio-angular structure of the volume source of this modified RTE ($\Delta \boldsymbol{f}_j$) is controlled by the radiance field which varies with the direction of solar illumination, for example."

Line 811:

"The spatial variations of the forward model derivatives are controlled by the optical distance to the sensor, through $\boldsymbol{I}_i^{\dagger}$, and also by optical distance to the solar source, through $\Delta \boldsymbol{f}_j$."

*The authors clearly demonstrate the issues associated with tomography for optically very thick clouds and plane parallel/stratiform clouds. They also note cloud types for which the condition and stability of the tomographic reconstruction should be good are extremely common. It would therefore be helpful if a little more care was given to discussing the performance of the Jacobian approximation for opacities associated with those cloud types, particularly for physically thicker cumulus clouds where one might expect the approximation to cause problems.*

We have added the following text to Section 5.3 (Line 1291) where the possible effect of errors in the approximate Jacobian on the optimization are discussed. New text is bolded.

"When there is noise in the gradients, the approximate Hessian can become corrupted. With bad curvature information, typically only very small step sizes will be valid, or there may be a complete failure to select a valid search direction. **This would result in early termination of the optimization, possibly far from a local optimum and will become more significant as the errors in the approximate Jacobian increase, i.e. as clouds become optically thicker.** In this sense, both the inherent ill-conditioning of the inverse problem and the approximate Jacobian errors should have a similar deleterious effect on retrieval performance. Moreover, it will not be possible to disentangle these two effects in nonlinear retrievals **without comparison against a reference method that uses an unapproximated method to linearize the forward model, whether it be a forward or forward-adjoint method. As such, we cannot make quantitative statements about what the consequences of using the approximated Jacobian without performing the retrieval with a known ground truth. The combined effects of the approximate Jacobian and ill-conditioning on retrieval accuracy can be examined in idealized circumstances where other sources of uncertainty in the retrieval are minimized. We perform such simulations in Part 2 of this study."**

*While it is asserted that tomography should work for optically thin stratiform clouds such as cirrus clouds given the Jacobian's condition number, Figure 17 showing the 16 stream stratiform results gives cause to doubt that is the case. Since this study has not actually done tomographic retrievals of cirrus clouds assertions regarding capabilities to apply tomographic retrievals to such clouds should be given appropriate caveats.*

We have clarified this by making changes at Line 1366 – Line 1374 in Section 6.2 where Figure 17 is discussed.

We have highlighted in Section 6.2 that the optically thin stratiform clouds occupy a similar range of condition number to that occupied by the typical optical depth range of trade cumulus and so, by this single metric alone, the tomographic retrievals of cirrus will be equally plausible to those of shallow cumulus. We have additionally clarified that the condition number alone is not sufficient evidence for the success of the tomographic retrieval and further investigation of tomography of cirrus is warranted.

Accordingly, the beginning of the last paragraph of the Section 7 (Line 1431) has been softened:

"We judge that the … retrieval method … is **most** suitable for retrievals in thin, cirriform clouds and trade cumulus over oceanic surfaces and their adjacent aerosols."

*The delta-M method and the Nakajima and Tanaka correction are mentioned in different places. It would probably be best to describe what is actually done in SHDOM when that code is described at the beginning and note that whenever you say delta-M you are referring to the treatment in the SHDOM code.*

We have added a brief description of the use of the delta-M and TMS correction where the algorithm of SHDOM is first introduced at Line 221 and made slight modifications to references to them at Lines 580 and Line 986. We have added a brief clarification of the relationship between the math and the numerical implementation at the beginning of Section 3.1 (Line 381) but we do not discuss the details of what is done in SHDOM except when absolutely necessary to avoid being further bogged down with details that are already described in Evans (1998) or Doicu and Efremenko (2019).

*Line 55: "at the full resolution of the sensors." would be more realistically "that makes complete use of the full resolution of the sensors."*

Modified as suggested.

*Line 60: "efficiently" should probably be "effectively"*

Fixed.

*Line 79: "there is no way to identify" is an exaggeration. Aerosol above clouds, cirrus above liquid cloud, multi-layered clouds are all things that have been done with IPA and passive remote sensing.*

Line 81: We agree that the wording is misleading as is. We have clarified this to refer to retrievals of the geometric variability of microphysics "within a cloud layer".

*Line 154: Could not find previous definition of LES*

Line 161: We have defined LES there.

*Line 285: Should note that logarithmic and other transformations of data are often used to help stabilize fitting.*

Noted at the end of the associated paragraph (Line 306).

*Line 331: "controlled by the singular value spectrum of K." Should caveat by noting that it is particularly the case for diagonal, or block diagonal error covariance matrices. Very strongly correlated measurements can significantly modify the spectrum of the Fisher information matrix compared with that of the Jacobian matrix.*

We have included a statement to this effect (Line 345):

"… largely controlled by the singular-value spectrum of **K**. **Strong off-diagonal error covariances in the measurements can also affect the spectrum of the Fisher information matrix through $S_\epsilon^{-1}$, though multi-angle imagers typically have block-diagonal error covariances that limit this effect.**"

*Line 417: do not bold face 1.*

Fixed.

*Line 430: Is this notation correct? The first term should probably not have a second (r,omega) parenthesis.*

Line 451: Fixed. The angular variable for the input to the reflection operator should have be $\mathbf{\Omega'}$ to distinguish between input and output angular variables. We keep the second set of $(\mathbf{r, \Omega})$ as it clarifies that the output of the reflection operator still varies with $\mathbf{\Omega}$ despite integration.

*Line 481: TOP, BOT and SIDE*

Line 510: Fixed.

*Line 816: Effective variance is probably not 10. Give correct value. 0.1?*

Line 856: Corrected to 0.1.

*Line 825: Why is theta_sun here in parenthesis? You have given the value of its cosine. Delete or put (theta_sun=xx°).*

Line 860: Fixed by deleting theta_sun.

*Line 882: Reference to Eq. (60) is given. It's wrong. I think it's Eq. (63), but please fix.*

Line 917: Fixed.

*Line 958: "extend" should be "extent"*

Fixed.